# Editing Metabolism, Sex, and Microbiome: How Can We Help Poplar Resist Pathogens?

**DOI:** 10.3390/ijms25021308

**Published:** 2024-01-21

**Authors:** Maxim A. Kovalev, Natalya S. Gladysh, Alina S. Bogdanova, Nadezhda L. Bolsheva, Mikhail I. Popchenko, Anna V. Kudryavtseva

**Affiliations:** 1Engelhardt Institute of Molecular Biology, Russian Academy of Sciences, Vavilov Str., 32, 119991 Moscow, Russia; kovalev_maksim_2002@mail.ru (M.A.K.); natalyagladish@gmail.com (N.S.G.); alina.bogdashka@yandex.ru (A.S.B.); nlbolsheva@mail.ru (N.L.B.); popchenko_m@mail.ru (M.I.P.); 2Department of Biology, Lomonosov Moscow State University, 119234 Moscow, Russia; 3Institute of Agrobiotechnology, Russian State Agrarian University—Moscow Timiryazev Agricultural Academy, 127434 Moscow, Russia; 4Center for Precision Genome Editing and Genetic Technologies for Biomedicine, Engelhardt Institute of Molecular Biology, Russian Academy of Sciences, Vavilov Str., 32, 119991 Moscow, Russia

**Keywords:** *Populus*, plant pathogens, plant immunity, plant microbiome, plant sex

## Abstract

Poplar (*Populus*) is a genus of woody plants of great economic value. Due to the growing economic importance of poplar, there is a need to ensure its stable growth by increasing its resistance to pathogens. Genetic engineering can create organisms with improved traits faster than traditional methods, and with the development of CRISPR/Cas-based genome editing systems, scientists have a new highly effective tool for creating valuable genotypes. In this review, we summarize the latest research data on poplar diseases, the biology of their pathogens and how these plants resist pathogens. In the final section, we propose to plant male or mixed poplar populations; consider the genes of the MLO group, transcription factors of the WRKY and MYB families and defensive proteins BbChit1, LJAMP2, MsrA2 and PtDef as the most promising targets for genetic engineering; and also pay attention to the possibility of microbiome engineering.

## 1. Introduction

*Populus* is a relatively young and evolutionarily successful genus in the family Salicaceae, widely distributed in the Northern Hemisphere, especially in temperate and boreal regions [1]. It is not only an important part of forest ecosystems but also of great economic importance [2,3]. One reason for this is that poplar has a high biomass accumulation rate [4] and is, therefore, used for the production of paper and wood products [5,6]. It is also increasingly used in green energy as it can be utilized to produce biofuels [7,8,9,10] and CO2 absorption to combat climate change [11,12]. In a number of cities around the world, it is used for urban landscaping [13,14], and its high resistance to heavy-metal pollution allows for it to be used for the phytoremediation of soils [15,16,17]. Secondary metabolites found in poplars may have medicinal value [18,19]. Since the genome of the woody plant *Populus trichocarpa* was first decoded in 2006 [20] and molecular genetic studies on this topic have become widespread, poplars have become model woody plants [21,22,23]. And since the interest in a green economy is intensively growing in societies, the applied use of scientific results obtained with this tree will be wide.

However, like any other plants, including those of cultural importance, poplars are susceptible to various diseases caused by fungal, bacterial and viral pathogens that serve as natural regulators of ecological systems [24,25]. However, this fact imposes limitations on the practical use of poplars, as damage to leaves, wood and other parts of the trees contributes to reduced growth rate and increased mortality. Since agrochemicals used as plant protection products are toxic and have negative impacts on the environment, including human and water resources [26,27,28], it is necessary to consider alternative ways of controlling phytopathologies that have a less negative impact on the environment. The development of pathogen-resistant transgenic plants seems to be the best option, as this approach minimizes the use of additional chemical compounds, which is in line with the concept of a green economy. To be able to create resistant poplars using genetic engineering methods, it is necessary to understand the physiological and molecular mechanisms that provide a response to infection with pathogens or immunity of poplars to them.

This review summarizes the currently known pathogens affecting poplar, and it specifies and clarifies their taxonomic affiliation. Further, the immune system of poplar is discussed in detail. Receptors perceiving the invasion of infectious agents, as well as the main hormonal and signaling transmission pathways triggering defense reactions, are described. The role of secondary metabolites, transcription factors and microRNAs regulating defense mechanisms is discussed. The influence of external agents, such as endophytes and phytophages, on the disease resistance of poplars is further analyzed. In the final section, we summarize the evidence for the use of genetic engineering, genome editing and other approaches used to improve poplar disease resistance and based on available research and theoretical knowledge, suggest apparently reasonable strategies for using CRISPR/Cas-based genome editing systems to improve poplar lines.

## 2. A Brief History of the Genus *Populus* and a List of Poplar Pathogens

The family Salicaceae includes ~50 genera and ~1000 species, of which ~330–500 and ~22–45 belong to the genera *Salix* (willow) and *Populus*, respectively. The family itself originated ~92 mya, probably in what is now Southeast Asia, and most of its primitive representatives now inhabit this region. One of the most significant events in the evolution of Salicaceae was the so-called Salicoid Whole-Genome Duplication (Salicoid WGD), which occurred ~60 mya and affected ~92% of the genome. Among the representative genera of the family, *Bennettiodendron*, *Idesia*, *Carrierea*, *Poliothyrsis*, *Itoa*, *Salix* and *Populus* passed through it. Moreover, *Salix* and *Populus* took advantage of the appearance of a large number of new genes in the genome and spread throughout the Northern Hemisphere, mainly in boreal regions, thus becoming the most biologically successful genera of the family [29,30]. 

According to modern data, the genus *Populus* is divided into four sections: *Abaso*, *Turanga*, *Populus* and ATL, the latter consisting of the traditional sections *Tacamahaca*, *Leucoides* and *Aigeiros*. Representatives of the most primitive section, *Abaso*, such as *P. mexicana*, have a narrow distribution range in Mexico and Southern United States, while plants of the section *Turanga*, which includes the desert species *P. euphratica*, are distributed in Central Asia and some parts of Africa. In contrast, representatives of younger groups—*Populus* and ATL—can be found practically throughout Eurasia and North America. Moreover, boreal species, which constitute the majority in the genus, have been more successful, advanced and diverse than tropical species such as *P. qiongdaoensis* from Hainan Island, China, and *P. ilicifolia* from East Africa [31].

However, after spreading over much of the landmass of the Northern Hemisphere, poplars not only became one of the most successful genera of woody plants, but also had to learn how to deal with a large number of pathogens from a variety of taxa.

Poplars have been planted and cultivated by humans since ancient times. White poplar wood was used in the 11th-13th centuries to create wood sculptures in China [32]. In the city of Iasi, Romania, a group of 15 white poplars grows, aged 233–371 years, which, unfortunately, are significantly damaged by fungal infections [33]. According to the information of the International Poplar Commission of the FAO from 1951, conscious experiments that began at the end of the nineteenth century to obtain new forms of poplars led to the emergence of disease-resistant lines [34]. Given the historical context and interest in poplar, which can be traced back through many centuries, it can be assumed that as a result of both spontaneous hybridizations and by directed breeding, modern representatives of the genus *Populus* are much more stable than ancestral forms. In the second half of the century, poplar research was carried out almost all over the world, including cultivation programs of various species [35]. Considering, in addition, the direct evolution of poplars due to the constant presence of pathogens, more genes were able to evolve in the genomes [36], which ultimately led to increased resistance to a number of pathogens. In general, mankind, using traditional breeding methods, could select those forms of poplars as starting material that turned out to be less damaged by phytopathogens. However, pathogens tend to overcome poplar resistance in a relatively short time, damaging the poplar again and, thus, reducing its value. In this regard, the creation of new stable poplar lines is a task that will not lose its relevance.

Pathogens are diverse in the biological strategy they exploit. One of the simplest classifications is their division into biotrophs, hemibiotrophs and necrotrophs. In brief, biotrophs feed on living plant tissues, whereas necrotrophs grow on dead parts of the plant. Hemibiotrophs use an intermediate approach: they first nourish on living tissues, and then, after they die off, they continue feeding on the dead material. Therefore, a defensive strategy that includes cell death is effective against biotrophs but will, in contrast, benefit necrotrophic pathogens. Therefore, it is crucial to have multiple signaling systems to properly orchestrate the immune response. It is generally accepted that salicylic acid (SA) and the reactions it induces are more fit to combat biotrophs, while jasmonic acid (JA) and ethylene (ET) are more suitable against necrotrophs, so we consider their signaling in more detail below [37,38].

Fungal pathogens mostly belong to the phyla Ascomycota and Basidiomycota (Figure 1). 

Ascomycota pathogens belong to the subphyla Taphrinomycotina (among which are *Taphrinia johansonii* and *T. rhizophora*, which cause the deformation of poplar fruitlets [39]) and Pezizomycotina. The latter are represented by the classes Eurotiomycetes (*Aspergillus* sp.), Leotiomycetes (*Marssonina brunnea*, *M. balsamiferae*, *Drepanopeziza populi-albae*, *D. populorum*, *D. tremulae*, *Botrytis cinerea*, *Septotis populiperda* (=*Septotinia populiperda*) and *Sclerotinia sclerotiorum* [40]), Sordariomycetes (*Entoleuca mammata*, *Tubercularia vulgaris*, *Cytospora* spp., *Colletotrichum gloeosporioides*, *Pestalotiopsis microspora*, *P. populi-nigrae*, *Cryphonectria parasitica*, *Nigrospora oryzae*, *Plectosphaerella populi* and *Fusarium* spp.: *F. solani*, *F. oxysporum* and *F. graminearum*) and Dothideomycetes (*Septoria* spp., *Cladosporium* sp., *Dothichiza populea*, *Neodothiora populina*, *Elsinoe australis*, *Venturia* spp., *Fusicladium* spp., *Alternaria alternata*, *Hormiscium* sp., *Botryosphaeria dothidea* and *Dothiorella gregaria*). *A. alternata* is the most typical pathogen of powdery mildew in poplar, which can also be caused by representatives of the genera *Aspergillus*, *Cladosporium* and *Hormiscium*. Species from the genera *Marssonina* and *Drepanopeziza* cause marsonioses, a brown leaf spot, and *Pestalotiopsis* spp. induces the formation of black spots on the leaf surface [41]. *Tubercularia vulgaris*, *Cytospora chrysosperma* and *Dothichiza populea* contribute to leaf necrosis; *S. populiperda* is a causal agent of leaf blotch [42,43]. Approximately 15 *Septoria* species, including *S. musiva* (=*Sphaerulina musiva*), *S. populi*, *S. populicola* and *S. tremulae* have been reported as the causative agents of septoriosis, a white-spot disease, and cankers [44]. *Venturia populina*, *V. inopina*, *V. radiosum*, *Fusicladium tremulae* and *F. radiosum*, which belong to the same family, can cause shoot blight and shepherd’s crooks [45], *C. parasitica* infection also results in blight [46], and *N. oryzae*, which was recently reported to infect poplars in China, is a causative agent of leaf blight [47]. Whereas *Hypoxylon mammatum* and *N. populina* affect wood, causing canker [48,49], *B. dothidea* [50] and *P. populi* [51] also do. *C. chrysosperma*, *C. notastroma* and *C. nivea* are also capable of causing cankers, and they can also colonize *N. populina* cankers [52]. *E. australis* and *C. gloeosporioides* are responsible for the development of anthracnose disease [53,54], while *D. gregaria* infection results in bark necrosis [55]. *Fusarium* spp. causes root rot and vascular wilt [56]. *B. cinerea* causes gray rot, or gray mold [57], while *S. sclerotinium* causes white mold [58]. Thus, ascomycete pathogens cause cankers and lesions on fruits and leaves. 

Basidiomycota include pathogens from the subphylums Pucciniomycotina and Agaricomycotina. The first are about 25 rust fungal species from the genus *Melampsora*, such as *M. larici-populina*, *M. larici-tremulae*, *M. medusae* and *M. × columbiana*, and they are biotrophic pathogens causing poplar leaf rusts [59]. Agaricomycotina are represented by necrotrophic macromycetes, often with a wide host range, feeding on wood and usually causing white, brown and yellow heartwood rots. These are members of the orders Agaricales (*Pholiota adiposa* [60], Hymenochaetales (*Inonotus hispidus* [61], *Phellinus tremulae* [62], *P. igniarius* [63] and Polyporales (*Laetiporus sulphureus* [64,65], *Polyporus squamosus* [66], *Fomes fomentarius*, *F. inzegnae* [67], *Climacodon septentrionalis*, *Spongipellis litschaueri*, *S. spumens* [68] and Cantharellales (*Rhizoctonia solani*—a causative agent of root rot and dampling off [69]).
Figure 1List of poplar fungal pathogens and the diseases they cause. For the reader’s convenience, their evolutionary and systematic relationships are depicted in the form of a manually constructed cladogram using external data from NCBI Taxonomy and articles [70,71,72,73,74,75]. The majority of poplar pathogens are fungi from the phyla Ascomycota and Basidiomycota. Among the Basidiomycota representatives, the majority are macromycetes that cause wood rot, as well as rust fungi from the genus *Melampsora*. Ascomycota pathogens are more diverse, causing powdery mildews, leaf spots, blights, necroses, rots, cankers, etc. Among them, the most important are representatives of genera *Marssonina*, *Septoria*, *Dothiorella*, *Botryosphaeria*, *Botrytis* and Alternaria. To resolve difficulties with the dichotomies, information from NCBI Taxonomy was used, as well as data from articles [70,71,72,73,74,75]. Created with BioRender.com (accessed on 18 January 2024).
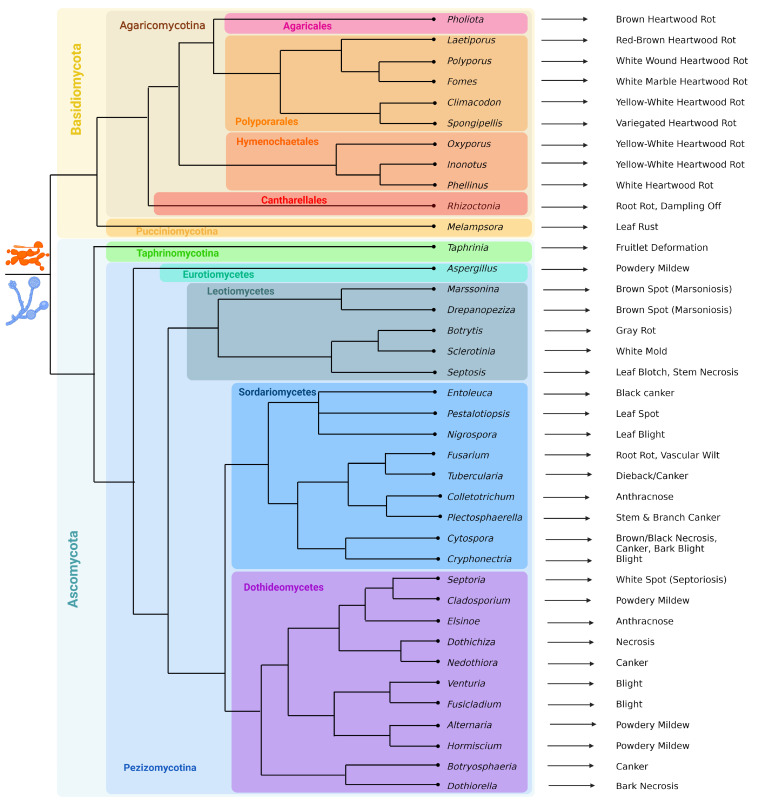


In addition to fungi, the generalist oomycete *Phytophthora cactorum*, which affects more than 200 different plants, is hemibiotrophic and causes rots, and it also effectively infected *P. trichocarpa* [76]. However, there are very few mentions of poplar diseases caused by oomycetes in the literature. Thus, they do not pose a serious threat to poplars.

Among the bacterial pathogens of poplar trees, *Pseudomonas syringae* f. *populi*, *P. cerasi* and *Xanthomonas populi* should be mentioned first of all [77,78,79]. They cause wilting, necrosis, rot, injury, tumors and cankers. *Lonsdalea populi* is a recently discovered species causing bark canker in poplars in different regions of Eurasia, and it is relative to *L. quercina* subsp. *populi* which also causes poplar diseases [80,81,82,83,84]. *Brenneria salicis* causes bacterial wilt [85]. All these bacteria belong to Gammaproteobacteria.

Poplars are also prone to viral diseases. Poplar mosaic virus (PopMV) from the genus *Carlavirus*, which is a (+)ssRNA virus, is one of the most studied viral pathogens, which is also the case in Europe [86,87], and a recent study described a new (−)ssRNA Emaravirus from the family Fimoviridae, transmitted by gall mites and causing aspen leaf mosaic in Scandinavian countries [88]. In the middle Rocky Mountain region, (+)ssRNA tobacco necrosis virus (TNV-A) from the family Tombusviridae was found to be able to infect *P. tremuloides* [89]. One of the largest genera of (+)ssRNA viruses affecting plants is *Potyvirus* from the family Potyviridae [90], which was isolated from *P.* × *euramericana* and *P. tremuloides* [91]. Bean common mosaic virus (BCMV), which is also from the *Potyvirus* genus, has also recently been described as the poplar mosaic disease causative agent of *P. alba* var. *pyramidalis* in China [92]. *P. alba* infected with begomoviruses (ToLCuKeV and PaLCuV) and their associated satellites (Ageratum alphasatellite, Multan alphasatellite and betasatellite) have been detected in Pakistan. The plants did not show disease symptoms, which led to the conclusion that they may be asymptomatic carriers of begomovirus-betasatellite-alphasatellite complexes, contributing to disease in other plants. Begomoviruses belong to the family Geminiviridae, are ssDNA viruses and are capable of causing cotton leaf curl disease, tomato leaf curl disease, etc., being transmitted by arthropods [93,94]. Moreover, it is worth noting that carrying some viruses can be beneficial to poplar: for example, LdNPV (*Lymantria dispar* nuclear polyhedrosis virus), which is pathogenic to insects but not to plants, affects gypsy moths if they eat infected leaves.

All poplar non-fungal pathogens considered here are summarized in Figure 2.

We tried to assess which of the pathogens we listed above are “the main” for poplar, i.e., cause the most significant damage to agroforestry. Unfortunately, we did not find accurate or at least rough quantitative estimates of economic damage from most poplar pathogens, and this requires a separate meta-analysis in the future. Nevertheless, based on the evolution of TLPs, it is plausible that fungi were a major biotic stressor during the evolution of poplars, as we discuss further in Section 3.3 [97]. Probably the most serious damage to poplar plantations is caused by members of the genus *Melampsora* causing rust diseases [98]. Fungi from the genera *Marssonina*, *Septoria*, *Venturia* and *Hypoxylon*; the bacterium *Lonsdalea populi*; and mosaic-causing viruses also have a considerable number of incidences [99]. This is indirectly reflected in the relative number of articles with different pathogens: a significant part of the experimental studies discussed in Section 3 and Section 4 of our review were conducted using the pathogens *Melampsora larici-populina*, *Septoria musiva*, *Marssonina brunnea*, *Dothiorella gregaria*, *Botryosphaeria dothidea*, *Botrytis cinerea*, *Alternaria alternata* and *Lonsdalea populi*.

Thus, poplars face fungal, oomycete, bacterial and viral pathogens, with the disease landscape changing dynamically around the world.

## 3. The Immunity of Poplars

### 3.1. A Brief Overview of the Major Plant Receptors and Signaling Pathways

Like all living organisms, plants use their analog of the immune system to defend themselves against pathogens. Unlike animals, they do not have adaptive immunity, but their analog of innate immunity is extremely sophisticated and diverse. The two most significant mechanisms are PAMP-triggered immunity (PTI) and effector-triggered immunity (ETI) (Figure 3) [37,100].

PTI provides basic defense and is triggered by PAMPs—pathogen-associated molecular patterns, which include non-specific substances found in many microorganisms in general and pathogens in specific—flagellin of bacteria, chitin of fungi, etc. PRRs (pattern recognition receptors) are located in the plasmalemma of plant cells and function as receptors for PAMPs. PRRs are divided into two groups: whereas receptor-like proteins (RLPs) consist only of an extracellular leucine-rich repeat (LRR) and a transmembrane domain (TM), and their cytoplasmic site is very small; receptor-like kinases (RLKs) additionally have an intracellular kinase site. Many pathways are involved in further signal transduction, including MAPK and calcium-dependent protein kinases (CDKs). In animals, Toll-like receptors perform an analogous function, but their diversity is incomparably lower than that of plant PRRs [37,100].

A genome-wide study revealed that there are 82 LRR-RLP genes in the *P. trichocarpa* genome, of which 66 are organized into clusters within which their evolution occurred mainly by tandem duplications [101].

A group of PRRs known as lysin motif RLKs (LysM-RLKs) has been extensively characterized for poplars. They are able to perceive fungal chitin oligomers, bacterial peptidoglycan and lipo-chitooligosaccharides. *A. thaliana* has 5 genes from this group: *AtCERK1* and *AtLYK2-5*, whereas *P. trichocarpa* has a total of 10 homologs. Most are expressed at intermediate levels in the majority of tissues, with only the *AtCERK1*, *AtLYK4* and *AtLYK5* homologs capable of binding chitin and only the *AtCERK1* homologs having kinase activity. Meanwhile, PtCERK1-like1, PtCERK1-like2, PtLYK4-like2 and PtLYK5-like1 proteins activate calcium signaling, protein degradation and the MAPK cascade, and they ultimately converge on TFs from the MYB, WRKY and bZIP groups discussed below [102,103].

Already at the PTI level, genetic modifications that increase poplar resistance are possible. For example, the hybrid poplar *P. davidiana × P. bolleana* is unable to launch PTI in response to treatment with nlp24, which is a 24-amino-acid-conserved site of necrosis and ethylene-inducing peptide 1-like proteins (NLPs) which are widely distributed among bacterial and fungal phytopathogens—for example, *M. brunnea* and *E. australis* possess NLPs. But AtRLP23, which is a PRR from *A. thaliana*, is able to recognize nlp24s epitopes. Therefore, the transgenic poplar expressing *AtRLP23* gene acquired the ability to sense this particular PAMP and, thus, became more resistant to NLP-containing fungi than the WT plant [53].

However, both bacterial and eukaryotic pathogens have adapted to circumvent PTI by using effectors, proteins that can block PRR signaling. For example, *S. musiva*, which causes poplar leaf spot, produces candidate secreted effector proteins (SmCSEPs), some of which, when transgenically expressed in *Nicotiana benthamiana*, promote infection by the fungus *F. proliferatum*, inhibit chitin-induced ROS burst and callose deposition via blocking PTI in many compartments [104].

But effectors can activate ETI, a system that elicits faster and stronger responses than PTI, up to and including the launch of hypersensitivity reactions. The main ETI receptors are NBS-LRRs, also called NB-LRRs, cytoplasmic soluble proteins that share some domains with NOD-like receptors of animals. They consist of three domains: LRR, nucleotide-binding site (NBS) and either toll interleukin-like (TIR), coiled coil (CC) or BED finger (named so for the BEAF and DREF proteins of *Drosophila melanogaster*, in which it was first described), from C-terminus to N-terminus. They can bind effectors both directly and via accessory proteins and then trigger an immune response [37,100].

*P. trichocarpa* has 64 TIR-NB-LRRs (TNLs), 119 CC-NB-LRRs (CNLs) and 34 BED-NB-LRRs (BNLs) in its genome compared to 93, 51 and 0 in *A. thaliana*, respectively. Thus, poplar summarily has about twice as many NBS-LRR genes as *Arabidopsis* [105].

TNLs transmit the signal about the infection through the EDS1/PAD4 and EDS1/SAG101 complexes, while CNLs do it through the NDR1 protein; both pathways ultimately converge on SA, the biosynthesis of which is upregulated with all the resulting effects, which we discuss below [105].

As for the BNL family, it is not only virtually unique to poplars among dicotyledons (as only *Vitis vinifera* possesses a gene with a BED domain), while among monocotyledons, for example, rice has 8 BNLs, which have arisen independently of poplar ones and at least one of them confers resistance to *Xanthomonas* sp. pathogens, and BNL protects wheat from rust disease [106], its role in defense and molecular mechanisms of action are not fully understood: BED is a zinc finger DNA-binding domain that recognizes a short sequence of 8 bp, and, hence, it is not highly specific [105].

Poplars interact with a huge number of symbiotic microorganisms, and various molecular mechanisms are employed to prevent the plant from triggering an immune response against them. For example, *Laccaria bicolor*, a model ectomycorrhizal fungus, secretes the effector protein MiSSP7, which is directed to the nucleus and there interacts with the JAZ6 protein, thus inhibiting JA signaling and thereby mitigating the immune response to root colonization [107].

Salicylic acid (SA), or 2-hydroxybenzoic acid, is known as a hormone critical for the activation of plant immune responses and for the acquisition of systemic acquired resistance (SAR), a state of increased resilience gained after primary infection. SA biosynthesis in plants begins with chorismate, an intermediate of the shikimate pathway which is used by plants to produce aromatic amino acids and related compounds, and proceeds through two alternative routes: the ICS or PAL pathways. In the ICS pathway, chorismate is converted into isochorismate by the enzyme isochorismate synthase (ICS) and then into SA, while in the PAL pathway, phenylalanine ammonia-lyase (PAL) transforms phenylalanine into trans-cinnamic acid, which is turned into SA via benzoic acid. The contribution of ICS- and PAL-pathways to total SA production is plant-specific—for example, *A. thaliana* mainly utilizes the ICS pathway, while poplar relies mainly on the PAL pathway. SA derivatives include methyl salicylate (MeSA), which is volatile, hydroxylated forms and conjugates with sugars and amino acids—all together, they are salicylates. SA reception is carried out via the Nonexpressor of Pathogenesis-Related proteins (NPRs), and TGA TFs serve as the final mediators of SA signaling as they modulate the expression of many resistance genes, primarily PRs. NPR1 is able to bind and activate TGAs, but in the absence of SA, it exists mostly in an oligomeric state in the cytosol, and the few molecules in the nucleus are bound and inactivated by NPR3 and NPR4. When SA emerges in the cell, NPR1 enters the nucleus, and NPR3 and NPR4 bind SA and lose their inhibitory potencies. Thus, the appearance of SA in the cell leads to the activation of PR genes expression, as they are under the control of TGAs [108,109,110,111]. Compared to *A. thaliana*, poplar is characterized by a higher content of SA and its derivatives in tissues, and its biosynthesis is not fully inducible, as constitutive synthesis at a low level occurs in the absence of pathogens as well [111].

JA and its derivatives are collectively known as jasmonates and are known as stress response hormones. Chemically, they are oxylipins as they are obtained through the oxidation of fatty acids and contain a cyclopentanone group. Jasmonates are associated with resistance to both abiotic unfavorable factors such as drought, temperature differences, salinity, heavy-metal toxicity and biotic stresses—infections, especially those caused by necrotrophic pathogens. Volatility allows jasmonate to be transmitted both between distant parts of the same plant and between different plants, increasing the effectiveness of protection and playing the role of a “communicator”. The biosynthesis of jasmonates includes the following steps: first, lipase acts on lipids of the inner membrane of chloroplasts, releasing α-linolenic acid; then, the sequential activity of enzymes 13-LOX, AOS and AOC yields 12-OPDA—the precursor of jasmonates, which is further transported to the peroxisome, where it is converted into JA. Natural JA derivatives include methyl jasmonate (MeJA), cis-jasmone (CJ), conjugates with amino acids, etc. In brief, jasmonate signaling is organized as follows: MYC, which is the main mediator of the JA response, is positioned on G-box elements on the promoters of the corresponding genes but is repressed by a complex consisting of JAZ, NINJA and TPL proteins. When JA or its derivative enters the cell and passes into nucleus, it binds simultaneously to the COI1 component of SCF E3 ubiquitin ligase complex and to the JAZ protein, which leads to the polyubiquitinylation and proteasomal degradation of JAZ, and consequently to the activation of MYC-controlled gene expression. JA signaling activates the expression of several genes such as PDF1.2 [112,113].

ET is a gaseous plant hormone that predominantly plays a role in plant development and growth [114], but this molecule is an important element in signaling in plant tissues. It is synthesized in two steps from S-adenosylmethionine (SAM): first, 1-aminocyclopropane-1-carboxylic acid (ACC) synthase converts SAM into ACC, which is then oxidized by ACC oxidase with the production of ET. ET receptors are transmembrane proteins located on the ER membrane: ETR1, ETR2, ETS1, ETS2 and EIN4. In the absence of ET, they activate the kinase CTR1, which phosphorylates EIN2, leading to its proteasomal degradation. ET serves as an inhibitor for its receptors, so in its presence, EIN2 is not degraded but promotes the transit of EIN3, EIL1 and EIL2 to the nucleus where they act as TFs and activate the expression of ERFs, which are the main mediators of the response to ET [115,116]. ET-induced plant defense may or may not depend on SA and its signaling. The ET-induced defense response against the heterotrophic fungus *D. gregaria* is independent of the SA pathway but requires the involvement of AFC signaling. Administration of ACC (aminocyclopropane-1-carboxylic acid, a precursor of ET) or overexpression of *PtoACO7* (ET biosynthesis gene) led to the increased expression of *PtoRbohD*/*RbohF*, encoding NADPH oxidases and enhanced H_2_O_2_ levels in poplar. The inhibition of NADPH-oxidase, thus, impaired ET-induced disease resistance and *PR* genes expression, whereas H_2_O_2_ application could completely cure disease hypersensitivity [117].

As discussed above, the SA pathway is beneficial against biotrophic pathogens, while the JA/ET response confers resistance to necrotrophs. Consequently, these pathways perform a multilevel crosstalk, and generally accepted at present is the model in which SA and JA are antagonistic to each other. SA negatively affects the expression of genes controlled by JA, such as PDF1.2, whereas JA suppresses SA biosynthesis. As a result, pre-incubation with biotrophs can make the plant more susceptible to necrotrophs [118]. However, this canonical view of SA, JA and ET interactions may be outdated or applicable only to *A. thaliana* and not to poplar, which is a woody plant. For example, in poplar, unlike in *A. thaliana*, SA biosynthesis is induced in response to infection by hemibiotrophs and necrotrophs, not just biotrophs, and JA responds to infection by rust, which is a biotrophic pathogen [119]. In another study, the exogenous administration of either SA or JA resulted in higher concentrations of the second hormone, and the engineered transgenic mutant with the hyperaccumulation of SA also had higher levels of JA than the wild type [120]. Thus, the current data support the presence of a positive SA-JA interplay rather than antagonism in poplar, at least in the case of some pathosystems. 

Systemic acquired resistance (SAR) is a plant organism condition characterized by increased resistance to wide-spectrum pathogens that emerges after primary infection. Even a localized infection or an avirulent pathogen can induce the development of SAR, which can last for several weeks or months. Resistance in SAR occurs due to the accumulation of high amounts of SA, constitutive SA signaling, and synthesis of PRs in great amounts [121,122].

But the most powerful and effective response against invading biotrophs is the hypersensitive response (HR). An HR launch is biologically costly, so it occurs only when there is a serious threat—for example, when NLRs detect an effector of an adopted pathogen. Simplified, HR involves the concentrations of intracellular Ca^2+^, ROS and NO increasing, leading to massive cell death in the invasion zone. HR often precedes the development of SAR [123,124]. In the case of mass lesions of plant tissues, HR can at least reduce the rate of disease spread by sacrificing some biomass and blocking obvious pathogen growth pathways.

The mechanisms described in this section, such as PTI, ETI and SA, JA, and ET signaling, are fundamental to plants, and most of the mechanisms described for *A. thaliana* and other herbaceous plants are relevant to poplar as well. However, we again note the differences between poplar and herbaceous plants such as *Arabidopsis*. Due to multiple duplications, poplar has a total of more TNLs and CNLs than *A. thaliana*, and BNLs are, to a first approximation, unique receptors for poplar, altogether allowing it to generate a more flexible immune response [105]. A second important difference is the likely lack of antagonism between SA and JA in poplar [120], although there are some conflicting studies. A third distinction is the higher concentration of SA in poplar tissues than in *Arabidopsis* tissues and the presence of its constitutive biosynthesis at a low level [111]. Finally, PRs, which are discussed below in Section 3.3 and are widely believed to be regulated by SA in plants, are likely to be SA-independent in poplar and regulated directly by the pathogen [120].

### 3.2. Role of Primary and Secondary Metabolisms in Poplar Disease Resistance

Pathogenic invasion causes, following the activation of signaling pathways, a massive metabolic reaction in plants, including poplars: as in any other stress, the concentration of a large number of chemicals and the intensity of many metabolic pathways change dramatically, which has certain negative consequences for the pathogen. At the same time, synthesized compounds often have nonspecific biological activity against a wide range of pathogens, which is due to evolutionary conservatism. For example, it was discovered that in response to *D. gregaria* infection in *P. beijingensis*, the concentration of fatty acids, some amino acids (Thr), most sugars and their derivatives, including those related to the pentose phosphate and galactose pathways, increases. Sugars and alcohols are utilized for energy production, with glucose being expended so intensely that its concentration is lower than in uninfected controls. Metabolites of the pentose phosphate pathway are used for the synthesis of lignins and phenols, fatty acids serve for the synthesis of jasmonates, and p-hydroxybenzoic acid, the concentration of which also increases under stress, is a precursor of flavonoids. At the same time, the concentration of TCA cycle intermediates is decreased because they are consumed for the synthesis of secondary metabolites. The concentration of most metabolites, among which are substances with fungicidal properties, is expectedly increased. Thus, during infection, the metabolic profile of plant cells is completely altered, with energy production and synthesis of secondary metabolites and signaling substances becoming its priorities [125]. At the same time, the distribution of energy resources may vary, for example, depending on the sex of the plant, as detailed below.

A particular and not fully understood nuance is the regulation of sugar metabolism during infections. Plants possess cell wall invertases (CWIs), vacuolar invertases (VIs) and cytoplasmic invertases (CIs). Invertases, or β-fructosidases, are capable of cleaving sucrose into glucose and fructose, thereby increasing the availability of hexoses in the appropriate compartment. Meanwhile, there are cell wall/vacuolar inhibitors of β-fructosidase (C/VIFs) that can bind to invertases and suppress their activity. *P. trichocarpa* has 39 genes encoding C/VIFs in its genome, and most of them have ABA, SA and MeJA-responsive elements in their promoters; thus, these proteins are involved in the response to both biotic and abiotic stresses. For example, C/VIF1 and 2 have apoplast localization and a higher affinity for CWI than for VI, and the expression of the genes encoding them is up-regulated in response to many abiotic stresses but decreased upon infection with *F. solani* [126]. The other research, which was also conducted on the P. trichocarpa—*F. solani* system—helped to unravel similar trends. Using transcriptome sequencing, it was demonstrated that the expression of 3 *CWI*, 2 *VI* and 7 *CI* genes was altered in the *F. solani* response to infection, with some genes being upregulated and others downregulated. More importantly, the expression of all examined C/VIFs decreased upon infection, which was probably the reason for the increase in CWI activity, while the activity of VIs and CIs remained virtually unchanged. Thus, in this study, the outcome of infection of P. trichocarpa by *F. solani* was a decrease in the expression of C/VIFs and an increase in the activity of CWIs. Additionally, the expression of all pectin methylesterase inhibitor genes (*PMEI*s), which are structurally related to *C/VIF*s, was reduced [56]. Some previous studies on other plants support that this is necessary for CWIs to be able to increase hexose availability in the apoplast, thus supplying the sick organs and activating plant defense responses [127,128,129,130], while others argue that enhancing pathogen invertase may provide pathogen with nutrients, thereby promoting the infection process and, thus, play into the hands of the pathogen [131]. Thus, despite the fact that the increase in the activity of apoplast invertases in the process of pathogenesis is obvious (whereas for the other invertases, this is less clear), the question of whether this contributes to more effective plant defense or, on the contrary, aggravates the course of infection, remains highly debatable.

Another indirect evidence of carbohydrate metabolism activation for defense is the induction of expression of two hexokinase genes (*PtHXK2* and *PtHXK6*) in the *P. trichocarpa* roots in response to invasion [132]. This correlates with our ideas that hexose uptake in adjacent tissues increases in response to infection.

The cancer-causing pathogens *Botryosphaeria* and *Valsa* promote carbon starvation during infection by both inhibiting photosynthesis and reducing the expression of metabolic enzymes and the concentration of soluble carbohydrates in conducting tissues [133,134]. Therefore, the increase in the activity of enzymes of sugar catabolism can also be considered as a plant response.

In addition to carbohydrates, the content of amino acids also changes in poplar. The content of glutamate, proline, glycine and a number of other amino acids increases, while asparagine levels decrease, as was shown in the same *P. trichocarpa*—*F. solani* system. These metabolomic changes occur in accordance with the induction of many genes related to the metabolism of aspartate, probably one of the key players in the amino acid response to infection [135]. 

As for secondary metabolites, plants use them almost universally, including extensively for self-defense against pathogens and phytophages. It is important to note that distinct compounds are active against them: terpenoids, cyanogenic glycosides, benzoxazinoides, alkaloids, phenolic compounds, benzoic acid and phytoecdysteroids such as phytoecdysone are effective against plant-consuming insects and mites, while SA, benzaldehyde, flavones, vanillic acid, alpha-saponin, etc. are toxic for fungi, and a few less compounds from this group are known to be active against bacteria—e.g., (+/−)-catechin. There are also relatively universal compounds, such as glucosinolates, active against both pathogens and phytophages [136]. When speaking about the mechanisms of this toxicity, phenolic compounds are known to alter the permeability of fungal cells by deforming membrane proteins, disrupting the physical gradient and ATP production. Other secondary metabolites, such as alpha-saponin, affect G-proteins and substances active against viruses often affect DNA, thus inhibiting replication. Poplars have a rich set of secondary metabolites, among which have been described flavonoids, proanthocyanidins, anthocyanidins and anthocyanins, phenolic glycosides such as salicortin, hydroxycinnamates mostly involved in lignin biosynthesis, fatty acids and their derivatives (some of which are volatile), terpenoid pathway members, etc. [137]. Next, we focus on those that give plants the ability to resist pathogens.

Figure 4 illustrates the main classes of secondary metabolites present in poplar and the interconversions between them.

One of the most important groups of secondary metabolites are lignins, which are a part of the secondary cell wall. A simplified scheme of lignin biosynthesis begins as follows: phenylalanine, derived from the shikimate pathway, as a result of the sequential action of enzymes PAL, C4H and 4CL, is converted into p-coumaroyl-CoA. One of its further transformation pathways is the sequential reduction to p-coumaryl aldehyde and then to p-coumaroyl alcohol by the action of the enzymes cinnamoyl-CoA reductase (CCR) and cinnamyl alcohol dehydrogenase (CAD), respectively. Alternatively, the conversion of p-coumaroyl-CoA to caffeoyl-CoA and caffeic acid occurs with the involvement of HCT, HQT and C3’H enzymes, and one of the by-products of this pathway is chlorogenic acid, discussed below. Caffeic acid serves as a precursor for ferulic acid and sinapic acid, which are converted to coniferyl alcohol and sinapyl alcohol, respectively, upon sequential catalysis by 4CL, CCR and CAD. P-coumaroyl alcohol, coniferyl alcohol and sinapyl alcohol are also called H-, G- and S-subunits, respectively, and are involved in the complex radical heteropolymerization of lignin under the action of peroxidases (PRX) and laccases (LAC) outside the cell [138,139,140,141,142].

*P. trichocarpa* possesses 16 genes encoding CADs (*PoptrCAD1-16*). Moreover, the expression of *PoptrCAD5*, *PoptrCAD11* and *PoptrCAD15* is upregulated in response to infection with *R. solani*, *F. oxysporum* and *Cytospora* sp., while *PoptrCAD11* and *PoptrCAD15* also respond to herbivore stress and, thus, are universal congeners of poplar defense against biotic stresses [143].

We have already mentioned 3-O-caffeoylquinic acid, also known as chlorogenic acid (CGA), in the biological production of which the enzyme *PtHCT2* is involved. It is the only one of the nine homologs of the *PtHCT* family in the *P. trichocarpa* genome that is induced by *S. musiva* infection. *PtHCT2* is differentially expressed in susceptible and resistant poplars, hinting at a possible role of CGA in resilience to *S. musiva*. In addition, *PtHCT2* is upregulated by several WRKYs—TFs, discussed below [144].

Lignin and related compounds contribute to plant defence against pathogens not only as part of the cell wall. For example, in the case of bacteria, lignin nanoparticles can (1) mechanically damage the cell wall, leading to cell lysis. They also (2) transfer to their surface large amounts of ROS, the uptake of which explains both the antioxidant properties of lignin in plants as well as its antimicrobial activity. Finally, (3) the entry of lignin nanoparticles into the cell disrupts the barrier function of the membrane, causing the intracellular pH to drop and ATP to be depleted (the latter is also explained by the inhibition of ATPase by lignin) [145,146]. The antifungal activity of lignin has also been investigated, as well as its correlation with chemical composition, but less is known about the mechanisms of toxicity in this case than for bacteria [146]. 

Lignans are biochemically related compounds to lignin. One of them is (±)-pinoresinol, which is a product of the oxidative dimerization of coniferyl alcohol under the action of laccase (LAC), a nonspecific copper-containing oxidase, while the stereoselectivity of this reaction is regulated by dirigent (DIR) proteins. *P. trichocarpa* overexpressing *PtDIR11* had increased resistance to *S. populiperda*, and this fungus was suppressed by both the methanolic extract of transgenic poplar and the recombinant PtDIR11 protein itself. The content of lignans and flavonoids increased in the plant, while the content of lignin remained nearly unaltered. At the gene expression level, the up-regulation of JA- and ET-pathway genes, down-regulation of the SA-pathway and up-regulation of most phenylpropanoids, lignans, lignans and flavonoid biosynthesis genes were observed [147]. The *PeDIR19* gene from the *P. deltoides × P. euramericana* hybrid is also capable of responding to SA, MeJA and *M. brunnea* infection and is, thus, probably also involved in plant defense [148].

Less is known about the antimicrobial and antifungal properties of lignans than about lignin itself. In bacteria, lignans have been shown to disrupt cell morphology by affecting membranes and impairing the uptake of valuable carbohydrates, and in fungi, they damage mitochondria, thus causing ROS accumulation; however, a holistic concept of lignans toxicity is still lacking [149]. 

Among the most important secondary metabolites used for defense against infections, there are members of the hormonal pathways discussed above, including SA and its derivatives. Their role has been studied in detail on *P. tomentosa*. *P. tomentosa* accumulates SA in high concentrations and is highly resistant to the biotroph *B. dothidea* [50]. Upon the invasion of this fungus, the expression of SA carboxyl methyltransferase (SAMT), which converts SA into MeSA, is increased in infected parts of the plant, and SA-binding protein 2 (SABP2), which performs the reverse reaction, is elevated in uninfected parts. Thus, MeSA serves as a transport form of SA in poplars, and the presence of a pool of SA allows for the rapid triggering of defense reactions [150]. Moreover, *P. tomentosa* accumulates tremuloidin, salicin, poplin and several other phenylglycosides in its tissues, in contrast to *P. beijingensis*, which is more susceptible to *B. dothidea*. Although these glycosides themselves have not shown antifungal activity, it is likely that they can serve as a reservoir, and SA can be quickly derived from them [50]. An even more striking example is *P. euphraica*, which is proposed to be super-resistant to biotrophic pathogens, which is largely due to its desert ecology and the need to constantly withstand abiotic stresses. Among the biochemical features of *P. euphratica*, it is noted that its bark is enriched with SA, including in the glucose-conjugated form, which allows for continuous and dominant SA signaling [151]. Thus, the presence of the SA pool in both free and conjugated forms determines the increased resistance of *P. tomentosa* and *P. euphratica* to biotrophic pathogens. 

SA is not only involved in plant immune response pathways, but also has a distinct biological activity itself. For example, it is used against various human skin diseases of bacterial nature such as acne [152]. The mechanisms of CA’s effect on bacteria are not completely clear, but the treatment of bacteria with the acid leads to changes in the proteome of *Pseudomonas* strains of bacteria, particularly a decrease in the accumulation of pathogenicity proteins, and provokes a decrease in membrane permeability [153]. SA has antifungal activity, inhibiting the germination of pathogenic fungi [154], and in *F. oxysporum* SA, it inhibits FoTOR complex 1 (FoTORC1), activating FoSNF1 in vivo [155]. Given that the rapamycin signaling pathway is present in a significant fraction of eukaryotic organisms, this growth inhibition mechanism may be a universal mechanism of action of SA against fungal pathogens. Contrary to this view, fungi can adapt to the presence of SA: *F. graminearum* is able to utilize the acid as its sole carbon source [156]. This makes a potentially versatile mechanism highly vulnerable to the ability of pathogens to adapt in short periods of time. 

It is important to take into account that increased synthesis and accumulation of SA lead to phenotypic changes that are almost impossible to predict, as, for example, happened when the *UGT71L1* gene was inactivated using CRISPR/Cas9, which led to the increased accumulation of SA and deterioration of growth and morphological changes in poplars [157]. This fact requires a careful selection of candidate genes and plant transformation methods in order to avoid undesirable effects associated with the consequences of imbalance in the content of secondary metabolites in tissues.

Also, increased SA content leads to increased biosynthesis of catechins and proanthocyanidins, which belong to flavan-3-ols and have negative effects on biotrophic pathogens [51,119,158]. 

Flavonoids are a large group of plant secondary metabolites with more than 9000 known substances, divided into 12 subgroups: chalcones, stilbenes, aurones, flavanones, flavones, isoflavones, phlobaphenes, dihydroflavonols, flavonols, leucoanthocyanidins, proanthocyanidins and anthocyanins. The common structural motif of the flavonoids, a C6-C3-C6 benzene ring, with the C3 ring, or C ring, being an oxygen-containing heterocycle condensed with the aromatic A ring, and the aromatic B ring being linked to, but not condensed with, the C ring. Flavonoids protect the plant against ROS, UV, cold stress, phytophages and infections [51,159,160]. Flavonoid biosynthesis begins, similar to lignin biosynthesis, with the formation of p-coumaroyl-CoA, further condensation of which, with malonyl-CoA under the influence of CHS, leads to the formation of naringenin chalcone—the common precursor of most flavonoids. The CHI enzyme then converts naringenin chalcone to naringenin, a member of the flavanones subgroup. Then, either flavones or isoflavones can be obtained from flavanones, or by sequential reactions catalyzed by flavanone 3-hydroxylase (F3H), dihydroflavonol 4-reductase (DFR), anthocyanin synthase (ANS) and anthocyanin reductase (ANR) flavanones can be converted into dihydroflavonols, then to leucoanthocyanidins, anthocyanins and ultimately into proanthocyanidins. Alternatively, dihydroflavonols can be oxidized into flavonols by the FLS enzyme. Leucoanthocyanidins may be converted into proanthocyanidins by the LAR enzyme. In this case, the B ring, which is not condensed with the A and C ring system, can be hydroxylated at the 3′-position by the F3′H enzyme or at the 5-position by the F5′H enzyme, which are active against flavanones, dihydroflavonols and flavonols, leading to an even greater diversity of all these compounds [161]. Flavonoids are a well-studied class of chemical compounds, so their direct role in plant responses to colonization by other organisms is better understood than for the vast majority of other groups of molecules. For different groups of plants, flavonoids have been shown to regulate the rhizosphere and endophytic community [162], for example, through phytoalexins that are synthesized in response to pathogen infection [163], which, along with lignin, make it difficult for pathogenic bacteria to colonize plant tissues [164].

Phytoalexins are a group of unrelated organic compounds that are united by biosynthesis by plants and the suppression of potential plant pathogens. In poplar, phytoalexins, which are derived from the phenylpropanoid pathway and belong to flavonoids, such as catechin and procyanidin B1, predominate, whereas in *A. thaliana*, for example, this function is fulfilled by camelexin, which has a completely different chemical structure [120].

Thus, poplar phytoalexins are among the end products of the phenylpropanoid pathway [163], which, along with lignin, make it difficult for pathogenic bacteria to colonize plant tissues [164]. In general, the total extract of quaking aspen (*P. tremuloides*) containing flavonoids, terpenoids and a number of other compounds has an inhibitory effect on different types of microorganisms [165].

Flavonoids and related compounds are themselves toxic to fungi and bacteria. Antifungal properties include (1) inhibition of protein synthesis, (2) mitochondrial dysfunction, (3) disruption of efflux pumps, (4) cell membrane damage, (5) impairment of cell wall formation and (6) disturbance of division. Mechanisms of antimicrobial action of flavonoids include (1) stopping bacterial replication by inhibiting dihydrofolate reductase, helicase, topoisomerase and gyrase, (2) blocking NADH cytochrome c reductase, (3) membrane penetration and increased membrane permeability, (4) ROS generation, (5) blockage of efflux pumps, (6) halting of cell wall synthesis, disruption of (7) conjugation and (8) quorum sensing, and disturbance of (9) cell adhesion and (10) biofilm formation [166,167,168,169].

Secondary metabolites and primarily flavonoids respond to pathogen infection in other plants and poplar. We briefly consider only some of the few examples of work on this topic. For example, flavan-3-ols, which include both monomers such as catechin and oligo- or polymeric proanthocyanidins, have been described to accumulate in *P. nigra* stems in response to infection with the canker-causing pathogen *P. populi*. The expression of structural genes for flavonoid biosynthesis, such as *ANS* and *LAR*, was up-regulated [51]. Similar changes occurred in *P. nigra* leaves infected with the rust-causing biotroph *M. larici-populina*. Moreover, the key regulators of this process were SA, but not JA, as well as the MYB-bHLH-WD40 complex, discussed below. And those poplar genotypes with higher content of catechins and proanthocyanidins had higher resistance to rusts [170]. 

In addition to flavonoids, phenylpropanoids and lignin composition are altered in response to infection. As has been shown in wild-type *P. trichocarpa* infected by *S. musiva*, the S-unit/G-unit ratio and Klason lignin content increase, while p-hydroxybenzoic acid contents reduces in the cell walls of the infected zone compared to healthy tissues [171].

Terpenes constitute another class of secondary metabolites responsive to infections. For example, when *P. trichocarpa* roots are incubated with *P. cactorum*, a broad-spectrum rot-causing pathogen, the expression of the *PtTPS5* gene, which is a sesquiterpene synthase producing (1S,5S,7R,10R)-guaia-4(15)-en-11-ol and (1S,7R,10R)-guaia-4-en-11-ol, is dramatically increased in plant cells. A number of terpenes have been previously described as growth-inhibiting substances for oomycetes [76].

Since increased flavonoid content may be associated with a more resistant phenotype, the authors of some relatively early papers attempted to address this through the direct overexpression of biosynthesis genes. For example, the overexpression of *PtrLAR3* in *P. tomentosa* resulted in increased proanthocyanidin content and elevated resistance to *M. brunnea* [172].

However, in a much larger number of articles in which genetic engineering methods were used to increase the content of secondary metabolites, primarily flavonoids, in poplar, the authors overexpressed not structural biosynthesis genes but TFs regulating them, primarily from the MYB and bHLH families. We discuss these works in detail further in the corresponding chapter.

In general, chemical compounds synthesized by different types of poplars are the most thoroughly studied and, at the same time, reliably shield against pathogens. Undoubtedly, it has drawbacks, but this ancient defense mechanism is multifaceted and sufficiently diverse so that the plant can successfully grow and develop even if attacked by a pathogen. Even in situations where the pathogen becomes resistant to chemical compounds, their presence can slow the progression of the disease.

### 3.3. PRs and Related Defensive Peptides and Proteins in Poplars

In addition to ROS and low-molecular-weight compounds, enzymes and peptides not related to these processes may contribute to defense against pathogens. We consider them in this section.

Plant pathogenesis-related (PR) proteins are known to be abundant in plants, and what they have in common is that they are involved in providing plant defense against infection. PRs are divided into different families: PR-1s have antifungal activity; PR-2s are β-1,3-glucanases; PR-3s, PR-4s, PR-8s and PR-11s are chitinases; PR-5s are also known as thaumatin-like proteins; PR-6s are proteinase inhibitors; PR-7s are endoproteinases; PR-9s are peroxidases; PR-10s are “ribonuclease-like proteins”; PR-12s are defensins; PR-13s are thionins; PR-14s are lipid transfer proteins; PR-15s are oxalate oxidases; etc. [173]. Many of the proteins and peptides described in this section are poplar PRs or have a similar function in other organisms.

Antimicrobial peptides (AMPs) found in many organisms constitute an important group: cecropins have been isolated from silkworm hemolymph, histatins have been isolated from primate saliva, cathelicidins are present in mammals, and defensins are widespread in numerous living species, including plants. 

A defensin isolated from *P. trichocarpa* (PtDef) has in vitro antibacterial and antifungal activity. When added to the plant, it affected bacterial growth similarly to the antibiotic cefotaxime while being less toxic, so one possible practical application of PtDef is to add it to plants instead of antibiotics, which is a more environmentally safe strategy. An alternative approach is to create transgenic poplars overexpressing *PtDef* gene. Such a line in the experiment had increased resistance to *S. populiperda*, while the plants had an increased content of GA, ABA, SA and JA and activity of the corresponding pathways but decreased concentration of IAA and expression of genes under its control. It is probable that the increased content of most PRs and non-specific lipid transfer proteins (nsLTPs) is also associated with the super resistance of *PtDef*-overexpressing poplars, in addition to defensin itself [42]. Further studies revealed that transgenic *P. × euramericana* overexpressing *PtDef* carries out a more “aggressive” HR in response to *S. populiperda* infection, as the H_2_O_2_ concentration is higher in the transgenic lines than in WT [174].

The poplar genome contains a number of nsLTPs, the number of which has been increased by tandem and segmental duplications, and in vitro inhibitory activity against pathogens such as *S. populiperda* has been shown for some of them [175]. However, it is probably more promising to utilize AMPs from medicinal plants to enhance plant resistance, such as nsLTP LJAMP2 from motherwort *Leonorus japonicus*. Transgenic *P. tomentosa* with constitutive expression of the *LJAMP2* gene had increased resistance to *A. alternata* and *C. gloeosporioides*. It is likely that a part of the properties of LJAMP2 is related to its ability to transport nonpolar molecules, including styrenes, that elicit an immune response [176].

Another recently explored small cationic protein with antifungal properties is rust-inducible secreted protein (RISP). Its gene was discovered when analyzing the *P. trichocarpa* genome as one of 71 small protein genes of unknown function (*SPUF*s) lying in the LRR-RLP gene clusters [101]. *RISP* expression is induced upon wounding, and in earlier work, it was described in the transcriptome as a marker of response to *Melampsora* infection with unknown function [177]. Further studies revealed that recombinant RISP indeed possesses anti-*Melampsora* activities; it suppresses fungal growth and prevents urediniospore germination by directly binding to them. In plants, RISP undergoes limited proteolysis at maturation, is secreted into the apoplast and leads to an increase in pH in poplar cell culture [178]. However, studies on *RISP* overexpression in poplar have not yet been conducted, so this could be one of the promising avenues for improving its disease resistance.

Transgenic expression of *MsrA2*, which is an N-modified host defense peptide (HDP) dermaseptin β1 from the Amazonian poisonous aboreal frog *Phyllomedusa bicolor*, in *P. nigra × P. maximowiczii* increased plant resistance to *S. musiva* infection. MsrA2 has marked toxicity in vitro against fungi, and its putative mechanism of action and that of other dermaseptins is to increase microbial plasmalemma permeability. Although MsrA2 has some phytotoxicity, its first symptoms begin to appear at a peptide concentration three times the minimum inhibitory concentration required to suppress the growth of *S. musiva* by 95% [179].

Some of the most important enzymes responsible for defense against fungal pathogens are chitinases. In terms of PRs, the chitinases are PR-3, PR-4, PR-8 and PR-11 [173]. The genome of *P. trichocarpa* contains 48 genes of different chitinases belonging to 7 different groups and located on 13 different chromosomes. Their number was largely increased by tandem duplications; so, at the same time, poplar has many more chitinase genes than *A. thaliana*. Some genes are expressed constitutively, while most can increase expression levels in response to exposure to chitin, chitosan, SA and MeJA. Poplar chitinases can cleave both chitin and chitosan, although the activity toward them is different [180].

Although poplars possess their own chitinases, it seems more prospective to create a transgenic plant expressing a chitinase from an organism, which is specialized in chitin degradation to create plants with indeed greatly enhanced resistance to fungal infections. Apparently, it was this logic that led to the emergence of transgenic *P. tomentosa* overexpressing the *Beauveria bassiana chitinase 1* (*BbChit1*) gene which demonstrated increased resistance to *C. chrysosperma*. *B. bassiana* is an entomopathogenic fungus exploited for the biological control of insect pests, so its enzyme is more efficient than homologs from plants, and a strong promoter allows for a lot of protein to be accumulated. The use of transgenic chitinases is a valid and environmentally friendly strategy for preventing fungal diseases [181]. Some generalization of the preceding two works was the creation of *P. tomentosa* overexpressing both *BbChit1* and *LJAMP2*. This double transformant was more resistant to *A. alternata* than plants overexpressing only *BbChit1* or only *LJAMP2*, while amongst the latter ones, plants expressing *BbChit1* were more resistant than plants expressing *LJAMP2* [182].

The PR-5 protein group includes zeamatin, osmotin and thaumatin-like proteins (TLPs). Many TLPs act as β-1,3-glucanases through a structural motif known as the acidic cleft. *P. trichocarpa* has 55 TLP genes, which is significantly more than A. thaliana and O. sativa, which have 10 TLP genes each. At least 20 *TLP*s were under positive selection pressure, and because TLPs possess antifungal activity, the authors of the paper suggest that it is fungal pathogens that have driven the evolution of poplar defense systems more than other biotic stresses [97]. Overexpression of *PeTLP* from *P. deltoides × P. euramericana* in the same poplar resulted in a significantly more resistant phenotype to *M. brunnea*. Although PeTLP, unlike many other TLPs, did not possess antifungal activity in vitro, its role in enhancing resistance probably lies in the activation of plant immunity and expression of other defensive proteins [183].

Universal stress proteins (USPs) are important proteins associated with the response to various types of stresses. The *P. trichocarpa* genome contains 46 *USP*s, which is significantly more than in *A. thaliana*, although the genes themselves are highly conserved. In *P. davidiana × P. alba* var. *pyramidalis*, the expression of many *USP*s is significantly induced in response to *F. oxyporum* infection [184]. Despite the importance of USPs, their mechanism of action in the plant is not fully understood—perhaps they are involved in redox signaling [185].

The heterologous expression of *Hsp24* from the fungus *Trichoderma asperellum* in the hybrid poplar *P. davidiana × P. alba* var. *pyramidalis* yielded an interesting result. In *T. asperellum*, *Hsp24* expression is elevated in response to various stresses. Transgenic plants are characterized by lower membrane permeability, higher SOD and POD activity, increased expression of *PR*s and genes related to the JA (*JAR1*, *MYC1*) and SA (*NPR*, *PR1*) pathways, and, as a result of all this, greater resistance to *C. chrysosperma* and *A. alternata*, compared to the control ones [186].

Proteins associated with the regulation of oxidative stress are worth mentioning briefly, as it is an important part of the response to pathogen invasion. For example, in *P. deltoides × P. euroamericana*, the expression of the *PdePrx12* gene encoding peroxidase is reduced in response to infection by *B. dothidea* and *A. alternata*. The *PdePrx* overexpressing poplar line had decreased H_2_O_2_ and was more susceptible to these pathogens, while plants with reduced *PdePrx12* expression, on the contrary, were more resistant to them due to excess H_2_O_2_ [187]. On the other hand, glutathione, which is a ROS scavenger, is associated with the response to the pathogen. In response to *B. dothidea* infection in *P. tomentosa*, the expression of glutathione transferase L3 (*GSTL3*) and glutathione S-transferase tau7 (*GSTU7*) is increased, which may be either the plant response to excess H_2_O_2_ or the result of the molecular control of the processes by the pathogenic fungus [188].

Poplar’s own defence proteins and proteins from other organisms that can be heterologously expressed in poplar are shown in Figure 5.

### 3.4. Transcription Factors Regulating the Immune Response

The regulation of immune response, synthesis of secondary metabolites and other defense-related processes, is carried out in plants by a limited set of families of transcription factors (TFs). In the following, we will consider the representation and role of these families in poplars and how they can be leveraged to enhance resiliency.

One of the most important ones is the WRKY family, named thus for the presence of the WRKYGQK domain within the protein. WRKYGQK, together with the zinc finger domain, binds to the W-box element in the promoter of its target genes, among which many genes for the synthesis of secondary metabolites, including alkaloids and flavonoids, have been found in many plants. WRKY family genes are pivotal in defense against biotic stresses [189]. Representatives of this family are found in algae and land plants and are divided into four groups: I, II, III and IV based on their domain structure [189]. Like many other poplar genes, most of the WRKY family genes were amplified as a result of Salicoid WGD [190].

When analyzing the genome of *P. trichocarpa*, more than 100 representatives of the WRKY family were detected, and the cis-elements W-box, MBS, CGTCA-motif and TCA-element were found in the vast majority of promoters, capable of responding to signals from WRKY, MYB, MeJA and SA, respectively. Thus, most of them can be activated by abiotic and biotic stresses, biotrophic and necrotrophic pathogens, and TFs of the same and other families. This is in agreement with the experimental data. However, WRKY genes are responsible for different aspects of resistance and have differential expression in plant tissues [189]. Poplar is unlikely to possess more WRKY genes than most other plants, unlike, for example, NBS-LRR genes. When WRKY III family genes were compared in four model plants, 10, 13, 6 and 28 members of this group were found in poplar, arabidopsis, grape and rice, respectively. At the same time, WRKY III genes in poplar were shown to be under a strong purifying selection, indicating their importance [191].

For example, in the same research, it was shown that group III member *PtrWRKY89* is an SA-inducible gene, so its overexpression in *P. tomentosa* led to increased expression of several PR genes and resistance to *D. gregaria*, but at the same time, it had no effect on genes related to JA signaling [189]. The overexpression of this gene in *P. trichocarpa* resulted in increased tolerance to *Melampsora* sp., which is also a biotrophic pathogen. Moreover, it led to increased expression of *PtrWRKY18*, *PtrWRKY35* and *PtrWRKY77* because PtrWRKY89 activated them through W-box elements in the promoters [192]. Thus, PtrWRKY89 can be considered as one of the central genes associated with resistance to biotrophic pathogens.

Analogous to *PtrWRKY89* in *P. trichocarpa*, the group III representative *PsnWRKY70* has broad potencies in *P. simonii × P. nigra* hybrid. *PsnWRKY70*-overexpressing poplars were more resistant to *A. alternata* infection, and *PsnWRKY70*-repressed plants were better able to withstand salt stress, so this gene contributes to defense against biotic stresses but reduces fitness to abiotic stresses. PsnWRKY70 is a subject to both self-regulation and regulation by TFs PsnNAM, PsnMYB, PsnGT1 and some MAPK pathway proteins. In the *PsnWRKY70*-overexpressing line, genes of the MAP kinase cascade, calcium channels and calcium-dependent protein kinases, as well as homologous genes (*WRKY6*, *WRKY18*, *WRKY22* and *WRKY22-1*), LRR domain protein genes, and SA pathway genes were upregulated. Thus, PsnWRKY70 activates crucial defense cascades related to both PTI and ETI [193].

On the other hand, the overexpression of *PtrWRKY40* in *P. tomentosa* conferred it increased susceptibility to the biotroph *D. gregaria*, which correlates with a lower concentration of endogenous SA and the decreased expression of SA-dependent genes such as *PR*s. However, the overexpression of *PtrWRKY40* in *A. thaliana* results in its increased resistance to the necrotrophic pathogen *B. cinerea*, with a constitutively higher expression of JA-dependent genes. *PtrWRKY40* belongs to group IIa. Thus, PtrWRKY40 increases resilience to necrotrophic pathogens but decreases adaptation to biotrophic pathogens. Moreover, PtrWRKY40 localizes in the nucleus but does not act as a transcription activator [194]. In contrast, another group IIa gene, *PtoWRKY60* from P. tomentosa, responds most strongly to SA signaling and weakly to JA, and its overexpression leads to increased resistance to *D. gregaria* by activating the SA-dependent, namely, *PR5* genes, but does not affect the expression of JA pathway genes such as *JAZ8* and JAZ10 [195].

Contrary to *PtrWRKY40*, the overexpression of the SA-inducible gene *PtrWRKY73*, which is a group I member, in *A. thaliana* increases its resistance to biotrophic pathogens as it was shown on *P. syringae* but decreases fitness to necrotrophic pathogens such as *B. cinerea*, and the SA-related defense genes *PR1*, *PR2*, *PAD4* and *CPR5* were upregulated; PtrWRKY73 can be phosphorylated by MAPKs. In contrast to PtrWRKY40, PtrWRKY73 is a transcriptional activator [196].

One of the earliest and most extensively studied genes of the family was the group IIc member *PtWRKY23* from *P. tremula × P. alba*. *PtWRKY23* expression is induced by *Melampsora* sp. infection, chitosan, SA signaling and wounding and is a typical early-response gene. More importantly, both poplars overexpressing and underexpressing *PtWRKY23* were found to be more susceptible to *Melampsora* infection than the wild-type ones. A transcriptome assay of *PtWRKY23* overexpressing plants showed that changes in gene expression were largely consistent with the response to *Melampsora* infection but also provided a clue as to which changes were associated with the susceptibility of this line. Firstly, in *PtWRKY23* overexpressors, genes related to the biosynthesis of lignin and cell-wall-related carbohydrates were down-regulated. This reduced cell wall synthesis, hence negatively affecting the plant’s ability to build a mechanical barrier. Secondly, the expression of several genes related to flavonoid biosynthesis (*PAL*, *F5H*, *C4H* and *CCR*) was decreased. Thirdly, the chitinase gene was downregulated. Finally, PtWRKY23 inhibits oxidative burst also by increasing peroxidase activity, and, therefore, H_2_O_2_ concentration was lower in *PtWRKY23* overexpressing poplars [197].

Another gene studied in this regard is *PsWRKY25* from *P. simonii*. It was found that sorbitol exposure increased poplar resistance to *A. alternata*, and PsWRKY25 was one of the most important effectors: it activated the expression of *PsCERK1* (*ceramide kinase 1*) gene, MAPK cascade genes and secondary metabolites, including phenylpropanoids, biosynthetic genes, etc. Poplar overexpressing *PsWRKY25* also had increased resistance to this pathogen [198].

It should also be kept in mind that defense against pathogens is not the only function of poplar WRKYs—rather, they may be associated with the display of a wide range of resistance to biotic and abiotic stresses. For example, *PeWRKY41* from *P.* × *euramericana* is an SA-, MeJA- and ET-dependent gene, the overexpression of which increases the level of peroxidases, chitinases, etc., and the resulting transgenic plants become more resistant to phytophages and salt stress [199]. Similarly, WRKY1 from *P. euphratica* is induced by HSF1 and contributes to salt tolerance [200].

To summarize the discussion regarding the TFs of the WRKY family, we must note that this is a large family of very important genes both for plants in general and for poplars in particular, which are extremely important for resistance not only to pathogens but also to abiotic stresses. The exact function of most of them has not yet been studied, which future researchers will need to address. At the same time, when creating transgenic poplars, the overexpression of one WRKY gene may increase resistance to some pathogens and decrease resistance to others, such as in the case of biotrophs and necrotrophs. Moreover, many WRKY genes are expressed in wild-type plants at exactly the level that gives them maximum protection against infections, as we have seen in the example of *PtWRKY23*, and attempting to alter their expression will only weaken poplar. Thus, when creating transgenic plants with increased disease resistance, it will be necessary to screen using different pathogens to understand exactly which infections the line has increased resistance to, by how much, and why. It is also necessary to study in more detail the role of other genes of this family and to consciously choose a target for genetic manipulation.

Among the groups of TFs which are important for resistance in plants, there are *bHLH* and *MYB* families. The name bHLH stands for basic helix-loop-helix, and this corresponds to the protein domain structure. This family includes the effector of JA signaling MYC2 and TT8, which is discussed below. Members of the MYB family have one, two, three or four repeats of DNA-binding domains and often function as homomultimeric complexes. bHLH and MYB frequently form heterodimeric and heteromultimeric complexes that participate in the regulation of the gene expression of biosynthesis of secondary metabolites: flavonoids, anthocyanins and proanthocyanins—for example, this role was shown for the MYB-bHLH-WDR (MBW) complex. The latter protein is WD40 repeat (thus, WD40 or WDR) containing TF, for example, TTG1 [201].

It should first be noted that although most MYB TFs positively regulate flavonoid biosynthesis, there are also negative regulators among the representatives of this family, for example, *MYB182*. Its overexpression in *P. tremula × P. alba* led to a decrease in the transcription of both structural flavonoid biosynthesis genes and their positive regulators from *MYB* and *bHLH* families, resulting in a lower content of anthocyanins and PAs in transgenic plants than in WT plants [202]. Two other repressors, MYB165 and MYB194, have a similar function. They both interact with bHLH131 and repress the flavonoid and phenylpropanoid biosynthesis pathways as well as the *MYB* activators, as was shown in experiments on their overexpression [203]. PtrMYB57 exerts its repressor functions through the C-terminal LxLxL domain; it interacts with bHLH131 and PtrTTG1; and after *PtrMYB57* knockout by the CRISPR/Cas system, transgenic plants have higher levels of anthocyanins and PAs [204].

Next, we consider the activators of flavonoid biosynthesis. Among them, MYB134, which responds with increased expression to UV-B irradiation, mechanical damage and *Melampsora* infection, was the first to be studied in detail; hybrid poplars with *MYB134* overexpression have increased PA content and decreased content of small phenylpropanoid metabolites [205]. In a more recent study, the overexpression of *MYB134* in *P. tremula × P. alba* increased plant resistance to *Melampsora* rust [170].

*MYB115* is another important MYB activator, and its overexpression in *P. tomentosa* led to the increased expression of several flavonoid biosynthesis genes, enhanced proanthocyanidin content and made the plant more resistant to *D. gregaria* infection. Since TT8 and TTG1 interact with MYB115 and form a WDR complex with it, simultaneous coexpression of *PtoMYB115*, *PtoTT8* and *PtoTTG1* in *Nicotiana benthamiana* can lead to the increased expression of *ANR1* and *LAR3,* and if they are overexpressed in poplar, this may be a more effective strategy to increase secondary metabolite levels and, thus, pathogen resistance compared to sole overexpression of *MYB115* [55]. Although this was not tested in the article, it is likely that such transgenic poplar has even higher proanthocyanidin contents and is even more resistant to pathogens than poplar overexpressing *MYB115* alone.

But what was tested for resistance is a line of transgenic *P. alba* var. *pyramidalis* overexpressing both *PalbHLH1* and *PalMYB90*. Upon infection, these plants more efficiently up-regulated the expression of genes related to flavonoid biosynthesis: *F3H*, *DFR*, *ANS* and *ANR*, and, therefore, showed increased resistance to *B. cinerea* and *D. gregaria*. At the same time, basically, transgenic plants had increased levels of total phenols, proanthocyanidins, anthocyanins, quercetin and kaempferol compared to the control ones. This correlates with the observation that *P. euphratica*, which is superior to *P. alba* in resistance to *D. gregaria*, has both higher contents of anthocyanins and expression levels of *bHLH*, *DFR* and *F3H*. Overexpressor plants also showed higher ROS content and higher transcription of *WRKY70*, *NAC12* and *ERF1* genes compared with WT [206]. 

The overexpression of *PtrMYB119* or *PtrMYB120* genes from *P. trichocarpa* in the hybrid poplar *P. alba × P. tremula* resulted in the upregulation of a large number of flavonoid biosynthesis genes, especially *PtrCHS1* and *PtrANS2*, and enhanced production of anthocyanins such as cyanidin-3-O-glucoside. *PtrMYB182*, a repressor of anthocyanin and proanthocyanidin biosynthesis, was downregulated in the transgenic plants, and no negative effect on growth was detected despite the increased anthocyanin content [207]. Interestingly, *PtrMYB120* is a positive regulator of both flavonoid and lignin biosynthesis, as shown in experiments on its overexpression and suppression in hybrid poplar [208].

In addition to the ones discussed above, *MYB6*, *MYB118* and *MYB117* were overexpressed in various studies, but the resulting transgenic poplars were not studied for disease resistance. *MYB6* overexpression in *P. tomentosa* increased the biosynthesis rate and flavonoid content in the tree, while through the interaction of MYB6 with KNAT7, a competing pathway of lignin biosynthesis was inhibited by the downregulation of *CCR2*, *CCoAOMT1*, *F5H*, *COMT2* and *CAD* genes, and here lies the difference in regulation by MYB6 and MYB120 [209]. PdMYB118 promotes the accumulation of anthocyanins after wounding, and their levels were also increased when *PdMYB118* was overexpressed in *P. deltoides*. Interestingly, PtrJAZ1, which is a repressor of the JA pathway, also suppresses the function of PdMYB118 because it binds PdTT8 and thereby disrupts the MDR complex [210]. *MYB117* overexpression in hybrid poplar resulted in increased B-ring hydroxylation of flavonoids but did not alter their total content, but this can still alter their biological activity [211]. In conclusion, none of the *MYB119*-, *MYB120*-, *MYB6*-, *MYB118*- or *MYB117*-overexpressors have been tested for pathogen resistance, and this is an area for future research.

MYB TFs also regulate the biosynthesis of other secondary metabolite classes. For example, *PtoMYB142* responds to drought, and its overexpression in *P. tomentosa* leads to the upregulation of the wax biosynthesis genes *CER4* and *KCS* and, thus, increased wax accumulation in leaves, which helps them to survive water deficit conditions [212]. In the context of pathogen defense, it would be very interesting to overexpress *MYB142* and examine whether increasing the amount of wax contributes to improved pathogen defense. It is known that the cuticle can inhibit pathogen penetration both mechanically (in which case the amount of wax may play a crucial role) and biochemically, the latter being relevant for leaf pathogens such as rust fungi penetrating stomata [213].

Thus, MYBs and the transcription factors bHLH and WDR interacting with them are among the most important regulators of the biosynthesis of secondary metabolites, especially flavonoids. By overexpressing activating *MYB*s, the authors of the studies were able to achieve an increase in flavonoid content in poplars, and in some cases, it was directly shown that such transgenic plants are more resistant to diseases. The increase in flavonoid biosynthesis in double and triple overexpressions is more effective when other components of activatory MBW complexes are overexpressed in addition to *MYB*s. Thus, obtaining and introducing such overexpressing poplars may be one line of work. Another line of work could be to create knockouts for those *MYB* genes that reduce flavonoid biosynthesis, such as *MYB182* or *MYB165*.

The regulation of poplar defence responses by TFs is shown in Figure 6.

*CAMTA*s (calmodulin-binding transcription activators) are a family of TFs which contain Ca^2+^-binding and two DNA-binding domains. They are involved in normal growth and development as well as in reaction to both abiotic and biotic stress and are responsive to ABA, SA and JA. Seven members of this family have been found in the genomes of *P. trichocarpa* and *P. ussuriensis*: *CAMTA1-7*. It is currently known that in response to *A. alternata* infection, *PtCAMTA1* and *PtCAMTA2* are induced, the expression of *PtCAMTA5* is unchanged and the others are decreased; after some time, these kinetics change [214]. More research is needed to understand how CAMTAs are related to pathogen resistance and whether they can be used to produce improved transgenic plants.

Another family studied in poplar is *NAC*. NAC proteins contain A, B, C, D and E domains, and their DNA-binding domain is in the N-terminus. It is known that in some plants, these TFs can downregulate the expression of flavonoid and anthocyanin biosynthesis genes [215]. NAC proteins are discussed in more detail in the next section due to being a target of miRNA164 [216]. One of the NAC TFs is *PtATAF1-1* from *P. trichocarpa*. It is probably involved in the rapid launching of the immune response, since its expression is activated as early as one hour after treatment of poplar cells with Flg22 [217]. 

Representatives of the AR2/ERF (ET response factor) family have a 60 amino acid DNA-binding domains, which consist of three beta-sheets and one alpha-helix. These TFs affect the biosynthesis of lignin, nicotine, saponins, resveratrol, etc. [215]. Whereas 170 *ERF* genes were initially detected in the *P. trichocarpa* genome, an exploration of the transcriptome of rust-susceptible *P. nigra* × *P. deltoides* revealed 143 members of this group, of which 21 were upregulated and 72 were downregulated 4 days after inoculation with *M. larici-populina*. Moreover, most defense-related ERF targets were downregulated in infected leaves and, in contrast, *histone deacetylase 1*, which is an inhibitor of poplar innate immunity and a target of the ERF40 transcriptional repressor, was upregulated. Thus, the failure of *P. nigra* × *P. deltoides* to upregulate *ERF* genes expression in response to infection by the rust fungus and the resulting reduced expression of defense proteins and unblocked immune inhibitor signaling explain the susceptibility of this genotype, emphasizing the importance of ERFs in defense [218]. In another paper, *PdPapERF109* was cloned from and overexpressed in the hybrid poplar *P. davidiana × P. alba* var. *pyramidalis* (Pdpap). This increased resistance to the fungus *F. oxysporum*. At the same time, the activity of POD and CAT in transgenic plants was higher than in WT ones, and the content of H_2_O_2_ and MDA was lower. Thus, the overexpression of *PdPapERF109* increased ROS scavenging capacity and likely provided a more adequate immune response [219].

The bZIPs are named for their basic and leucine zipper domains, and they regulate the production of diterpenoid phytoalexins, anthocyanins, etc. [215]. One well-studied TF from the bZIP family is the VirE2-interaction protein 1 (VIP1), which regulates mitogen-activated protein kinase 3 (MPK3). VIP1 was named as such because it acts as an adaptor for the VirE2 effector from *Agrobacterium tumefaciens*, conditioning the interaction of VirE2 with importins and its transport into the nucleus [220]. And, although many molecular details of the VIP1 response to pathogenic stress are still unknown, transgenic *P. trichocarpa* overexpressing either *AtVIP1* from *A. thaliana* or its own *PtVIP1* has been shown to have increased resistance to *B. salicis*, and increased expression of genes from the *PR1* group was observed in them [85]. Probably the main reason for the increased resistance of *VIP1*-overexpressing plants is the VIP1 downstream target MPK3, which, like MPK6, activates *NBS-LRR* genes expression in plants and allows for SA signaling to be triggered [221]. Intriguingly, the MAPK pathway is not only important for plant defense, but also for fungal infection: for example, in *C. gloeosporioides*, the genes *CgSte50*, *CgSte11*, *CgSte7* and *CgMk1* encode an adaptor protein, MAPKKK, MAPKK and MAPK, respectively, and this whole cascade controls processes such as appressorium formation, invasive growth, melanin biosynthesis, hydrophobicity, conidia germination, response to ROS and overall pathogenicity [222].

Another characterized bZIP protein is PeTGA1, a TGACG-binding (TGA) TF isolated from *P. euphratica*. PeTGA1 activates the expression of *PeSARD1*, which is an important regulator of SA biosynthesis. Therefore, transgenic white poplar overexpressing *PeTGA1* has an increased SA content and responds more efficiently to *C. gloeosporioides* infection compared to WT [223]. 

*PdPapHB12* gene from the hybrid poplar *P. davidiana × P. alba* var. *pyramidalis* encodes a protein that is a TF from the HD-Zip I subfamily. *PdPapHB12* expression is induced by abiotic stress and is dependent on biotic stress (it is increased in response to *F. oxysporum* infection and decreased upon *C. chrysosperma* invasion), as well as increased in response to JA, SA and ABA hormone signaling [224]. However, a more detailed study of the role of PdPapHB12 in the regulation of the poplar immune response is required.

Trihelix genes are transcription factors that recognize the GT element. There are a total of 56 trihelix genes (*PtrGT1-56*) in the *P. trichocarpa* genome. The expression of most of them increases in response to abiotic stress, pathogen invasion and signaling by phytohormones (ABA, MeJA, SA) [225]. 

Transcription factors are closely linked to chromatin function. The alteration of epigenetic markers in response to infection is an important but poorly studied process, as sequencing studies usually focus more on changes in the transcriptome. The infection of poplar *Lonsdalea populi* was found to decrease DNA methylation levels, especially in CHH islands. The repeats and promoters of protein-coding genes are hypomethylated, leading to increased expression of the latter, especially *R*-, *PR*- and hormone signaling genes; genes encoding microRNAs are also demethylated [226].

### 3.5. Contribution of microRNAs to the Regulation of Defense Responses

Small non-coding RNAs (both siRNAs and miRNAs) are involved in plant defense responses, with miRNAs being more conservative and usually having a single target, while siRNAs are more variable and may have multiple targets [227].

Some miRNAs are essential for the proper plant response to pathogen invasion. For instance, the expression of miR172b in *A. thaliana* is increased upon infection, and it suppresses *TOE*, which is a repressor of *FLS2*—a component of the PTI system which senses bacterial flagellin [228]—while miR393 promotes PTI by repressing auxin signaling which can antagonize SA [229]. In defense, plants can engage the host-induced gene silencing (HIGS) mechanism, which consists of the suppression of virulence genes in the pathogen by host miRNAs. For example, in cotton, infection with the fungus *Verticillium dahliae* induces the expression of miR159 and miR166, which are then exported to hyphae [230]. HIGS is also used against parasitic plants, sometimes resulting in horizontal gene transfer [231].

However, several miRNAs differentially expressed at rest and during infection are not required for the activation of the immune response but rather to repress defense responses in the absence of the pathogen. For example, miR472, miR482 and miR2118 suppress *NB-LRR*s in plants [232]. Upon invasion, SNC1 accumulates in the nucleus and promotes the destruction of these “peacetime” miRNAs [233]. Some pathogens leverage these miRNAs in order to weaken host immune reactions. For example, miR319 is upregulated by a virus because it suppresses JA pathway effectors, thereby suppressing it [234], while miR528 and miR168 repress *AO* (*ascorbate oxidase*) and *AGO1*, respectively, and are inhibited by *AGO18* in plants because they may interfere with antiviral defense [235].

Thus, during infection, miRNAs are utilized by both the plant and the pathogen, with both sides using miRNAs to affect themselves as well as each other. Within a single plant, miRNAs are transmitted through the plasmodesmata system and the phloem, and transit between the plant and the parasitic fungus is often via haustoria. MiRNAs can be transmitted in both a “naked” form and as part of extracellular vesicles [227].

One of the first studies was conducted in 2012, where the transcriptome of *P. trichocarpa* infected with *B. dothidea* was analyzed. The infection led to elevated levels of 41 miRNAs from 12 families: miR156, miR319, miR166, miR164, miR408, miR160, miR168, miR159, miR172, miR398, miR1448 and miR1450. Most of them are also known to be involved in responses to different types of abiotic stresses, and miR156 is associated with all types; hence, miR156 may be related to the nonselective stress tolerance of poplar trees [236]. In this case, the miR156—*SPL* regulatory axis is one of the most important because SPL TFs destabilize the MYB-bHLH-WD40 complex, thereby preventing flavonoid accumulation [237]. A larger study conducted in the same year on *P. beijingensis* infected with *D. gregaria* allowed for the studying of not only 37 conserved miRNA families, but also 27 *Populus*-specific ones. Moreover, the expression of most of the miRNAs studied in both studies changed in the same direction in response to infection [238]. These two works were excellently reviewed in 2014 [239], so we move on to more recent studies.

We now note that the first studies already found that many of the miRNAs were differentially expressed in poplar in response to infection target *NBS-LRR* genes [236]. Subsequent work has shown that a disproportionate number of poplar-specific miRNAs, compared to those miRNAs that are conserved among plants, target specifically *R*-genes [240]. This is essential for a more precise and distinct regulation of the immune response [216].

It may seem counterintuitive that a plant would upregulate microRNAs targeting *NBS-LRR* genes and, thereby, block its own defense signaling. However, many microRNAs targeting *R* genes realize their regulatory functions not by upregulation during infection, but rather by expression at relatively high levels under normal growth conditions, repression of their targets in the absence of disease and downregulation during infection. One such microRNA is ptc-miR472, which can be downregulated in *P. trichocarpa* leaves after treatment with SA, JA or flagellin. The first experiments on the genetic engineering of microRNAs in poplar were also conducted with it: both an overexpression line and a reduced expression line were obtained, the latter being created using the short tandem target mimic (STTM) technology originally proposed for *A. thaliana* in 2012. In the STMM method, an artificial nucleotide sequence containing two binding sites mimicking the target gene and a linker between them is created; such a transgenic construct is transcribed and then binds to the target small RNA (microRNA or any other type), leading to a degradation of the latter by small RNA degrading nucleases and, as a consequence, to its silencing. As a result, the miR472 overexpressor line was the most susceptible to the hemibiotrophic pathogen *C. gloeosporioides*, as it failed to adequately upregulate *NBS-LRR*s, trigger ETI and activate ROS production, but, at the same time, it was the most resistant to the necrotroph *C. chrysosperma*, as it had the most active JA/ET signaling and increased expression of its marker gene *ERF1*. On the other hand, poplar line STTM472a with reduced levels of ptc-miR472 had maximum resistance to *C. gloeosporioides*, while its susceptibility to *C. chrysosperma* was comparable to WT plants [241]. Thus, editing microRNAs that somehow target *NBS-LRR* genes can increase resistance to either biotrophs or necrotrophs, depending on current needs. Creating a system in which the overexpression of such microRNA or STTM, or even both, would be conditional rather than constitutive will allow us more flexibility to artificially modify plant resistance depending on the current needs and type of pathogen.

Several papers in this area have focused on the interaction of poplar with *M. larici-populina*, a biotrophic pathogen that causes rust. In a relatively recent work, two novel miRNAs were isolated from *P. szechuanica*. They were named novel_mir_11 and novel_mir_357, and they target and negatively regulate the expression of *RPM1* and *RPS2/5* genes, respectively, while both targets contain NBS and LRR domains, thus being resistance genes. What was most significant in this work was that when the plant was infected with the avirulent *M. larici-populina* strain Sb052, the expression of both novel_mir_11 and novel_mir_357 declined, and the expression of their targets changed only slightly, which appears to be the normal immune response of the plant. If, however, poplar was inoculated with the virulent strain Th053 instead of Sb052, the expression of novel_mir_11 and novel_mir_357 increased, while the expression of their targets decreased steadily during the first 48 h after infection, which corresponds to the *M. larici-populina* growth phase. This probably indicates the ability of this pathogen to utilize the plant’s regulatory mechanisms to at least temporarily shut down its immune response [242]. Previously, the same group of researchers had shown that the infection of *P. szechuanica* with strain Sb052 resulted in the upregulation of most microRNAs, while infection with Th053 led to the downregulation of most microRNAs, indicating that there are some general differences in microRNA changes in response to avirulent and virulent pathogens [243]. 

Another study attempted to elucidate the cause of susceptibility of *P. nigra × P. deltoides* “Robusta” to rust fungus. In this poplar, PTI was activated after infection, but further *R*-genes did not respond properly: of 64 *TIR-NBS-LRR*s and 119 *CC-NBS-LRR*s, only 1 *TIR-NBS-LRR* family, which is regulated by miR472b, was upregulated, whereas members of the miR482 superfamily and their targets from both groups of *R*-genes hardly responded to rust infection. There was also no proper activation of SA-, JA- and ET-signaling, which was also possibly due to the dysregulation of microRNAs. Thus, the misregulation of microRNAs may be the cause of poplar rust susceptibility [244]. 

One important regulatory module in the plant response to fungal infection is an axis composed of miRNA164a, *NAC*s and *R* genes. NACs are extremely important transcription factors that have been shown in other plants to be upregulated in response to both biotrophs and necrotrophs, and their overexpression can enhance plant disease resistance. miRNA164a, which regulates *NAC*s, is highly conserved among flowering plants. In poplar, the targets of miRNA164a are two *NAC1* genes and one *NAC100* gene, and their expression is negatively correlated with miRNA164a expression [216]. 

Two studies were conducted with a focus on the response of *P. tomentosa* to infection by the bacterial pathogen, *Lonsdalea quercina*. One showed that the expression of miR1447 and miR171c was upregulated [245]. In the second one, the authors studied how the expression of microRNA targets, instead of microRNAs themselves, is altered during the infection. Among the differentially expressed genes, there were many participants of such processes as hormonal signaling pathways, plant–pathogen interactions, and phenylpropanoid biosynthesis. Genes related to auxin, cytokinin and brassinosteroid signaling were downregulated, and although SA-related *TGA* and *NPR1* genes were also downregulated, *PR-1* mRNA levels were increased. Among pathogen interaction genes, the expression of *FLS2*, calcium-dependent protein kinases and MYB TFs was upregulated, whereas the expression of various *NBS-LRR* genes changed differently. As for phenylpropanoid biosynthesis genes, the expression of most of them, especially *cinnamoyl-CoA reductase*, was upregulated. Although researchers did not measure the expression of all microRNA genes, a negative correlation between the expression of microRNAs and that of their targets was demonstrated [246]. 

A total of 78 microRNAs from 21 families, the largest of which are miR156, miR395 and miR167, are also differentially expressed in *P. alba* var. *pyramidalis* leaves in response to infection with a novel bean mosaic virus (BCMV) from the genus *Potyvirus*. Representatives of the families miR1444, miR390, miR397, miR398, miR399, miR408 and miR478 were upregulated, while miR1446, miR167, miR169, miR394, miR396 and miR350 were downregulated throughout the duration of infection. Among the miR395 and miR482 families, differences in expression were observed between different members and phases of infection. Since most miR395 representatives were upregulated at the beginning and downregulated at the end of infection, and miR395 is a target of genes associated with protein ubiquitination, it is possible that the manifestation of mosaic symptoms on leaves is also related to miR395, which requires further experimental confirmation [92]. 

Changes in miRNA profile occur not only during pathogenesis, but also in the course of the establishment of contacts with endophytes. For example, the endophytic strain *Streptomyces* sp. SSD49 upregulates 25 host miRNAs, including miR156 and miR160, and downregulates 13 miRNAs, including miR168, miR319, miR398 and miR408, when incubated with *P. tomentosa* seedlings [247]. MiR156a and miR168a may be important for mycorrhizal formation, for example, with the ascomycete fungus *Cenococcum geophilum* [248]. It is likely that endophytes leverage miRNAs expression regulation the same way that pathogens do, so studying interactions in the poplar–symbiont system can be accurately applied to poplar–pathogen relationships.

Not only miRNAs but also siRNAs are able to respond to infection and provide plant defense. For example, when *P. tomentosa* was infected with the fungus *M. larici-populina*, more than 95% of the fungal genes became targets of poplar sRNAs: siRNAs and miRNAs, with their numbers being particularly high at the stages of biotrophic growth and urediniospore formation and release, reaching 15954 and 15472, respectively (and the total number of genes targeted in all stages was 16372). This includes both housekeeping genes encoding ribosomal proteins, etc., and pathogen effector genes [240].

Moreover, many microRNAs in plants, which are about 30% in poplar, owe their evolutionary origin to transposons and pseudogenes, which are, thus, evolutionary catalysts for microRNA formation. For example, miR478e and miR6427 originate from transposons, while miR393c and miR6438b are derived from pseudogenes [216]. The same situation exists for with siRNAs: a great proportion of them is located in transposons and pseudogenes. The main differences in expression in response to different stages of rust fungus infection are attributed to siRNAs derived from the transposon regions [240].

In addition to small RNAs such as miRNAs and siRNAs, lnRNAs are involved in the regulation of defense responses in plants, including poplar. When studying the response of *P. × euramericana* to *M. larici-populina* infection, of the 3994 lncRNAs detected, 53 were differentially expressed between healthy and infected tissues: 18 were detected only in control samples, another 18 were downregulated and 7 were upregulated during infection, and 10 were present only in infected leaves, with most of the differentially expressed lncRNAs being in close proximity to and likely coexpressed with protein-coding genes. Interestingly, two lncRNAs were predicted to be capable of complementarily binding to microRNAs and, thus, competing with their protein-coding targets, namely, TCONS_00086550 binds to Ptc-miR396a and was downregulated during infection, while TCONS_00025165 binds to Ptc-miR530a and was upregulated in infected leaves [249]. In more recent work, the responses of *P. × euramericana* to the application of exogenous SA were studied. The expression of 606 mRNAs and 49 lncRNAs was altered in SA-treated samples compared with H_2_O-treated controls, with DEGs being associated with general responses to stress and light, disease resistance, growth and developmental processes, while mRNA-lncRNA interactions appeared to play an important role in the orchestration of all these responses [250]. Thus, lncRNAs, although still poorly understood in the regulation of plant immune reactions, are extremely important, as they interact with both mRNAs and miRNAs. Further research in this area is required to better understand their role in plant defense.

Some of the microRNAs involved in poplar defence are shown in Figure 7.

### 3.6. Intra-Population Differences in Resilience and Related Molecular Mechanisms

Like any other population, plant populations are heterogeneous in their susceptibility/resistance to infection. A number of genes are known in poplars, changes in which increase their resistance to disease or, conversely, weaken their immunity.

One good example is *MLO* (*mildew resistance locus O*) genes, which are known as plant vulnerability genes to powdery mildew because they are negative regulators of the immune response. *P. trichocarpa* has 26 *MLO* genes in its genome: *PtMLO1-26*, and the breakage of *PtMLO 17*, *18*, *19* and *24* may be associated with resistance to powdery mildew, since the knockout of homologous genes in *A. thaliana* confers resistance to a similar disease [251].

*Resistance* (*R*) genes are all those plant genes that have functions which are related to disease resistance: they include extracellular receptors, including those associated with kinases, intracellular receptors and kinases, etc. The largest group of *R*-genes are the *NBS-LRR* genes, of which poplar has more than 400: 64 *TNL*s, 119 *CNL*s, 34 *BNL*s and a large number of genes of other domain composition, such as *TNLT*s or *CN*s [252].

One way to classify plant resistance is to divide it into qualitative and quantitative resistance. Qualitative resistance provides complete resistance to a pathogen (i.e., makes the plant a non-host) by inducing a rapid and strong immune response against it, it is due to one or more genes (often *R* genes), but it is race-specific and unstable to evolutionary change. In contrast, quantitative resistance is only partial but, due to polygenicity (it is usually controlled by quantitative trait loci (QTLs)), it allows the plant to fight a wide range of pathogens and is, thus, more robust than quantitative resistance. Examples of alleles conferring qualitative and quantitative resistance to *M. larici-populina* are *R_US_* from *P. trichocarpa* and *R_1_* from *P. deltoides*; a cross between these poplar trees produces offspring with the genotypes *R_US_R_1_*, *R_US_r_1_*, *r_US_R_1_* and *r_US_r_1_* with different resistance to the pathogen; both alleles map to the NBS-LRR-rich region on chromosome 19 [253].

Poplar QTLs are described in the paper [254]. In *P. trichocarpa*, 4 QTL intervals containing a total of 38 markers are associated with a quantitative control of fungal pathogens, and another 3 QTL intervals with 40 markers are required for defense against phytophagous insects. The QTL intervals contain *TNL*, *lipoxygenase*, *oxidoreductase* and *cytochrome P450* genes, which have undergone numerous tandem duplications in the recent evolutionary past, most of which have led to increased plant resistance.

Disease resistance depends on the status of not only *R* genes or genes located within the QTL, but also any other genes. In the paper [255], a total of 40 SNPs were found to have a significant effect on the rust susceptibility and severity of this disease. Many of them were located, in addition to exons, in introns, intergenic regions, 3′ and 5′-UTRs, and they were associated with a wide range of genes and not just the R genes. On the other hand, in another study, one constitutively expressed *TNL*, one constitutively expressed *CNL* and one *G-type lectin receptor-like kinase* (*G-type lecRLK*), the expression of which is induced by pathogens, were described as *R*-genes, certain allelic variations of which were associated with resistance to *Melampsora* [256].

The whole-genome sequencing of 1000 *P. trichocarpa* individuals and testing them for resistance to *S. musiva* infection in another study [257] found 3 resistance-associated loci: 1 *L-type lecRLK* and 2 *receptor-like proteins* (*RLP*s), *RLP1* and *RLP2*, and one vulnerability locus, *G-type lecRLK*. Accordingly, poplars with a resistant phenotype have a higher expression of *L-type lecRLK*, *RLP1* and *RLP2*, and a lower expression of *G-type lecRLK*, than more susceptible individuals. All these proteins are associated with fungal cell wall receptivity, and RLPs interact with RLKs, but whereas some of them activate defense responses (e.g., L-type lecRLK), others are needed to establish symbiotic relationships and then reduce the immune response (G-type lecRLK) [257].

On the other hand, the interaction between plants and pathogens is a long-term arms race, with the latter also possessing a certain genotypic variability that causes their different phenotypes. For example, in 1994, the poplar qualitative resistance gene *RMlp7* ceased to confer resistance to some genotypes of rust fungi because a major deletion occurred in the genome of *M. larici-populina* in the region of the *AvrMlp7* effector to which RMlp7 binds [258]. Similarly, *M. brunnea* has genotypes that differ in host specificity due to the transcription of different genes and, in particular, the use of different effectors [259].

Resistant and susceptible individuals of *P. trichocarpa* differ not only genotypically but also in the dynamics and nature of the response to *S. musiva* infection, as shown by comparative proteomics analysis. For example, susceptible poplar (BESC-801) had an increased amount of *S* gene products such as *G-type lecRLK*, which was not observed in the resistant one (BESC-22). Moreover, the levels of proteins associated with SA-dependent responses, including PRs (such as chitinase PR-4), MAPK-dependent and Ca^2+^-dependent signaling compounds, SAR and HR proteins, increased faster and to a greater extent in resistant poplar than in susceptible poplar. It has higher levels of PTI members, such as PRRs (especially important ones such as PtCERK1), PRR coreceptors (such as BAK1 and SOBIR) and HR trigger-related receptors HRLI4 and HIRP, as well as ABC transporters that release phytoalexins. In contrast, ET and JA signals are more active in susceptible poplar [260]. Similar observations were obtained when comparing the response of susceptible *P. deltoides* and resistant *P. tomentosa* to *Lonsdalea quercina* subsp. *populi* bacteriosis. The expression of genes related to SA signaling (such as *PR1*s, *NPR1*s, *TGA1,2*) was higher in the resistant phenotype, while JA signaling genes showed a more diverse pattern: *MYC2*s were more actively expressed in the resistant plant, while *JAZ1* and *COI1*s in the susceptible plant [260]. Finally, in a system where the intolerant genotype was *P. nigra × P. deltoides* and the tolerant ones were *P. trichocarpa × P. deltoides* hybrids, while the infection agent was *M. larici-populina*, a rust-causing biotroph, in intolerant plants, the JA pathway was activated at the early stages and SA was inhibited; in contrast, in tolerant plants, JA was inhibited, and the expression of genes associated with the SA response (among which *NBS-LRR*s, *EDS1*, *NDR1*, *WRKY*s and *PR*s) was upregulated. In addition, *Kunitz-type trypsin inhibitor* (*KTI*), a gene associated with the development of apoptosis, which is important for the containment of biotrophs, was activated [261]. Thus, a resistant phenotype in different poplar species is often associated with a higher ability to rapidly and strongly activate SA signaling and related processes. 

Such a susceptible phenotype with decreased SA and increased JA activity can be reproduced artificially. For example, myo-inositol is an antagonist of SA signaling, and galactinol is an agonist of JA signaling. In the normal response of *P. alba × P. grandidentata* to infection by the biotroph *M. aecidiodes*, there is a decrease in galactinol levels; activation of SA, calcium and phosphatide signaling; increased expression of *NPR1* and *PR1*; and increased ROS levels, which do not occur in the mutant line overexpressing *Arabidopsis galactinol synthase 3* (*AtGolS*) and *Cucumber sativa raffinose synthase* (*CsRFS*) and, therefore, making it more susceptible to rust diseases [262]. On the other hand, the overexpression of *PdbLOX2*, which is the initial enzyme in the JA biosynthesis pathway in the *P. davidiana × P. bollena* hybrid, enhances its resistance to such a dangerous necrotroph as *A. alternata* [263]. Thus, both natural and artificially created genotypes with enhanced SA signaling are more resistant to biotrophic pathogens, while those with enhanced JA signaling are more resistant to necrotrophic ones. This consideration can be used both in breeding and in genetic engineering when it is necessary to obtain trees resistant to a certain type of pathogens.

Metabolomic methods provide another insight into the causes of susceptibility. The genotype of *P. trichocarpa* resistant to *S. musiva* accumulated SAR-related metabolites such as trehalose, sucrose SA and gentisic acid in greater amounts than the susceptible one in response to infection by this fungus, whereas the susceptible poplar did not, and many of its metabolites were catabolized by the fungus [264]. This, once again, emphasizes the role of SA in defense against biotrophs, but from the metabolomic point of view. The limit of such resistance is *P. euphratica*, which has almost absolute resistance to biotrophs but is susceptible to a number of necrotrophic pathogens [151].

Interestingly, the results of all these papers are consistent with old ideas about the existence of SA-JA antagonism and contradict the articles we discuss earlier at the end of Section 3.1, whose authors concluded that in poplar, SA- and JA-signaling pathways positively affect each other’s activity. This is partly because in the articles supporting antagonism, the evidence was indirect, based on genotype correlation: plants with increased SA signaling activity usually had relatively reduced JA activity, and vice versa [260,261,262,263]. However, these observations may not have a direct causal relationship, whereas in the paper [120], the authors experimentally proved a positive interaction by applying exogenous SA or JA to poplar, which led to an increase in the concentration of JA or SA, respectively. However, more research is needed to definitively clarify this contradiction.

Surprisingly, when infected with *Lonsdalea populi*, the resistant poplar has a more active auxin signaling and expression of photosynthesis-related genes, allowing it to maintain its growth, whereas the susceptible one is unable to quickly block the infection and activates more *NBS-LRR* and *PR* genes that inhibit growth processes [265]. Thus, paradoxically, in the growth-defense trade-off, growth processes are more important in resistant poplar, while susceptible poplar displays stronger defense reactions. In the same study, it was hypothesized that a likely cause of poplar susceptibility is the delayed accumulation of antimicrobial compounds such as catechin [263]. 

Some interesting data have recently been obtained to study epigenetic differences—the different pattern of DNA methylation in susceptible (*P.* × *euroamericana*) and resistant (*P. tomentosa*) poplars during infection with the canker-causing bacterium *Lonsdalea populi*. It was found that during infection, there is a decrease in DNA methylation, apparently necessary for the immune response, with the resistant poplar already having a lower basal level of DNA methylation than the susceptible one. DNA hypomethylation in response to infection occurs in both poplar genotypes, but certain patterns of this process differ between them [226].

Another previously discussed reason for susceptibility of poplar to *M. larici-populina* is impaired regulation by microRNAs: whereas PTI is adequately regulated, microRNA-mediated regulation of ETI, SAR and HR appears to be impaired by the rust strain in susceptible poplar [244].

Finally, within the same species, poplars can differ in the copy number of a particular gene, a phenomenon called gene copy number variations (CNVs). For example, the *P. balsamifera* from Northern and Southern Canada studied as reported in paper [266] differed in the number of ~1700 of the nearly 20,000 genes studied: southern populations had more copies of defense genes (*chitinases*, *leucine-rich transmembrane protein kinase*, *TNLs*, etc.) because they were under greater pathogenic pressure, while northern populations were more adapted to withstand abiotic stresses associated with light and water deficiency.

Information on the resistance of certain phenotypes and genotypes to diseases is of great importance, as it allows for the production of resistant plants by simple crossing. Often (as in the case of CNVs), a large number of genes are involved in this resistance, and selection is more relevant than the use of genetic engineering.

Figure 8 shows the differences between “susceptible” and “resistant” poplar at different levels.

## 4. Environmental Factors Protecting Poplar from Infections: Endophytes, Phytophages and Chemical Elements

### 4.1. Endophytes

In nature, poplar interacts with a huge number of organisms, not all of which can even be classified as potential pathogens. Organisms from the external environment—usually fungi and bacteria—that interact with the plant on the surface are called the rhizosphere in the case of the root and the phyllosphere in the case of the leaves. Those that establish a closer symbiotic relationship with the plant by penetrating its tissues are called root and leaf endophytes, respectively. In addition to these, a number of poplar endophytes can live inside the wood of the stem and, taking advantage of the anaerobic conditions created there, fix atmospheric nitrogen [267].

To date, >5000 endophytic strains, including >3000 bacteria and >2000 fungi, have been isolated from members of the genus *Populus*, with >550 bacterial and ~65 fungal genomes sequenced [268]. One of the best known is the fungus *Laccaria bicolor*, which forms ectomycorrhiza. Others include the arbuscular mycorrhiza-forming fungus *Rhizophagus irregularis*; the fungi *Atractiella*, *Phialophora*, *Illyonectria* and *Mortierella*; and the bacterium *Pseudomonas fluorescens* [267]. In our recent study, we described bacteria from the genera *Bacillus*, *Peribacillus*, *Staphylococcus*, *Kocuria*, *Micrococcus* and *Corynebacterium* as endophytes of *P. alba* roots [269]. The poplar microbiome is formed at early stages of ontogenesis under the influence of both selection and stochastic factors. Its composition depends on the genotype, habitat and developmental stage of the plant. While bacteria are influenced by a strong homogenizing selection during this process, the fungal part of the community develops under the influence of a mixture of weak selective and stochastic processes [270].

Among others, a very important function of the normal plant microbiome, i.e., the totality of all microorganisms interacting with it, is to protect the plant against pathogenic invasion. In this way, they contribute to the stability of the ecosystem to which they belong. Their combined name is DefenseBiome—a community of bacteria associated with defense against infections [271].

Plants during infection can target changes in their microbiome to provide protection—for example, the “cry for help” strategy involves releasing compounds (benzoxazinoids, aminocyclopropane-1-carboxylic acid (ACC), etc.) into the soil that will attract beneficial microorganisms. Therefore, often, the relative abundance of bacteria associated with plant defense is elevated during a pathogen attack [272].

Bacteria and fungi utilize several mechanisms to combat infectious agents: (1) direct microbial antagonism, which includes the secretion of antibiotics and antimicrobial toxins, volatiles, cell-wall degrading enzymes, siderophores, etc.; (2) positive effects on plant health, thus indirectly allowing the plant to fight infections more effectively; and (3) interactions with the host immune system: priming and induced systemic resistance (ISR). The terms priming and ISR refer to a state of heightened alertness of the whole plant which allows it to respond more quickly and strongly to pathogenic invasion; this is called priming in the early stages, and in the later stages it is known as ISR [272].

A good example of direct microbial antagonism can be provided by the recently isolated wild poplar strains of *Burkholderia vietnamiensis* WPB, *Bacillus velezensis* AFE 4A and AFE 21B, *Pseudomonas* sp. AFE 5 and AFE 8. *B. vietnamiensis* WPB has strong in vitro antifungal activity due to its ability to produce occidiofungin, and clusters of ornibactin, cepacin A, difficidin, macrolactin, bacillaene, fengycin, bacilysin, bacillibactin, sessilin, viscosin and tolaasin, compounds toxic to fungi, bacteria and oomycetes, were detected in the genomes of these strains [69]. *B. velezensis* and the closely related species *B. amyloliquefaciens* are often endophytes of plants and can protect them from infection. For example, *B. velezensis* strain EB14, according to mass spectrometric analysis data, produces five cyclic lipopeptides: iturins A1, A2 and A9, as well as subtulene A and fengycin, thereby inhibiting the growth of *S. musiva*. Clusters of antibiotic biosynthesis of surfactin, rhizoactin, difficidin, macrolactin, and bacilysin were detected in its genome, and the possibility of the biosynthesis of subtulene and fengycin was also confirmed [273,274]. Also, *Streptomyces* sp. strain SSD49, originally isolated from *Stropanthus divaricatus* and which is a growth-promoting endophyte for *P. tomentosa*, is able to inhibit in vitro the growth of phytopathogenic strains of fungi (*Cryphonectri parasitica*, *S. sclerotiorum*, *D. gregaria*, and *B. dothidea*) and bacteria (*Lonsdalea quercina* subsp. *populi*) [247]. A number of other compounds, including primary metabolites, may also participate in such antagonism. For example, endophytic strains of *B. velezensis* 33RB and *Aspergillus niger* 46SF suppress the pathogens *C. gloeosporioides* BJ02 and *F. oxysporum* 20RF, and the possible bioactive compounds identified by LC-MS are the fumaric, DL-malic, citric, isobutyric, glutamic, lauric, linoleic, oleic, stearic and myristic acids as well as a number of others, totaling about 30 metabolites with proven antimicrobial activity [54].

Symbiotic fungi isolated from *P. alba* demonstrate mechanisms of antagonism different from those known in bacteria. Belonging to the orders Pleosporales, Eurotiales, Dothideales and Hypocreales, these leaf endophytes in vitro inhibit the growth of *Venturia tremulae* due to (1) parasitism on pathogen mycelia and (2) substrate competition, which suppresses pathogen development, whereas (3) antibiosis was rarely observed, and it has a relatively weak effect. In any case, *P. alba* when treated with a suspension of these strains, becomes more tolerant to shoot blight caused by *V. tremulae* [275]. Thus, spraying on leaves is one of the practical possibilities of utilizing these strains.

A very interesting example of the first strategy is *Cladosporium oxysporum*, which is a hyperparasitic fungus towards *M. medusae* f. sp. *deltoidae*. This parasitism consists of three distinct mechanisms involving enzymatic, direct and contact parasitisms. Thus, *C. oxysporum* protects poplar from rust but has no negative effect on the leaves of the plant itself [276].

*B. amyloliquefaciens* AW3 is an example of a microorganism that prevents the rot caused by *F. oxysporum* Fox68 by priming *P. davidiana × P. bolleana* (*PdPap*) seedlings. Indeed, *B. amyloliquefaciens* AW3 stimulated the expression of *PAL*, *PPO*, *SOD*, *CAT*, SA- (*PR1* and *NPR1*), JA- (*MYC2* and *JAR1*) and auxin-related (*MP*, *AUX1*, and *LAX2*) genes in plants, thereby enhancing the natural resistance of poplars and being an eco-friendly biocontrol agent for this fungus [277]. *Laccaria bicolor*, which is a mycorrhizal fungus, affects the expression of a huge number of genes in the plant and protects it from infection by *B. dothidea*. Exposure to the fungus alters the activity of pathways related to hormone signaling, plant immune responses, cell wall metabolism and ROS scavenging. For example, when exposed to *L. bicolor*, ROS clearance is a more efficient in infected tissues, which reduces ROS-induced stress in the plant and, thus, allows it to combat infection more effectively and increases the synthesis of cell wall proteins such as expansins, which physically prevent the fungus from entering; the expression of defensive genes increases; etc. [278]. Fungal foliar endophytes isolated from *P. trichocarpa* have a more localized and limited effect. *Stachybtrys* sp., *T. atroviride* and, to a lesser extent, *Ulocladium atrum* and *Truncatella angustata* reduce the degree of leaf damage by the rust fungus *M.* × *columbiana*. At the same time, only those leaves, which were previously inoculated with the endophyte, became resistant, which may indicate a local stimulation of immune response as well as direct antagonism. In any case, these strains do not cause a systemic reaction of the plant [279].

Disease development is often associated not with the interaction of a single endophyte and a single pathogen, but with a critical change in the entire microbial community. For example, in healthy heartwood of *P.* × *euramericana*, the dominant bacterial genera are *Proteiniphilum*, *Dysgonomonas* and *Bacteroides*, while during the development of a poorly studied bacterial wet heartwood disease, *Proteiniphilum*, *Actinotalea* and *Methanobacterium* increase in abundance and become dominant. At the same time, both pathogen and disease-associated bacteria may be among these three genera [280].

The microbiome of poplars, and, hence, their disease resistance, changes with age. When studying the dynamics of the root community in *P. tomentosa* from 1 year to 35 years of age, it was found that the representation of Actinobacteria increased with age. This process is closely related to the biosynthesis of flavonoids and their content in the soil surrounding the root, and their biosynthesis, in turn, depends on the activity of the *Transparent Testa 8* (*TT8*) gene, which is a *bHLH* TF and a component of the WDR complex, as discussed above. Actinobacteria are beneficial in suppressing plant diseases by synthesizing antibiotics—for example, the *Streptomyces andamanensis* bj1 strain studied in this work is able to biosynthesize hygromycin B or related substances due to the presence of the *hyBl* cluster in its genome. It is likely that this age-related accumulation of Actinobacteria may be related to poplar longevity, which should be verified in further studies [281].

Endophyte-pathogen interactions are not purely unidirectional. The anthracnose-causing fungus *C. gloeosporioides* has been shown to reduce the diversity of both bacterial and fungal phyllosphere communities, with the fungal community changing more dramatically. While in healthy leaves the most abundant genera were *Mortierella*, *Fusarium*, *Colletotrichum*, *Aspergillus* and *Cladosporium*, after infection, the representation of *Colletotrichium* increased from about a few-20% to 85–100% as the pathogen itself displaced all other fungi. This trend was replicated on different poplar genotypes: *P.* × *canadensis*, *P.* × *beijingensis* and *P. tomentosa*, which have different resistances, respectively. At the same time, this effect may be not only a consequence of direct antagonism by *C. gloeosporioides*, but also mediated by secondary metabolites, many of which were altered during infection [282].

For a more comprehensive overview, it should be said that, naturally, the microorganisms associated with poplar also help it survive abiotic stress. For example, a review of ectomycorrhizal fungi indicates that ectomycorrhiza provides resistance to stresses such as (1) osmotic stress (mycorrhizal roots are characterized by a higher capacity for water transport and hydraulic conductivity of roots, which ultimately allows plants to better withstand conditions such as drought and salinity; mechanisms for increased tolerance include the fact that the mycelium of the fungus can grow and absorb moisture under more extreme conditions than the plant, as well as the effect of the fungus on aquaporin expression in the root, phytohormone balance, etc.; for example, the fungus *Paxillus involutus* helps *P. euphratica* to survive in its arid native habitat), (2) heavy-metal pollution (since many of the metal ions are captured in the exudate secreted by the fungus, on its cell wall, and in its cytosol or vacuoles), and (3) the presence of organic pollutants (ectomycorrhizal fungi can increase the biomass of poplars growing on contaminated soil). Thus, inoculation with mycorrhizal fungi can increase poplar resistance to abiotic stresses, too [283]. Similarly, bacterial endophytes of poplar roots help it resist stress associated with water deficit, shading and soil contamination with copper ions [284]. In our paper on the root bacterial endophytes of *P. alba*, we found genes in many of them, the presence of which may be useful for the plant to withstand heavy-metal and organic-compound pollution [269].

Figure 9 summarizes the antagonism of poplar endophytes and pathogens.

Not all endophytes are beneficial against infections. Among epiphytes and endophytes isolated from wild *P. trichocarpa*, antagonists of the rust-causing fungus *M.* × *columbiana* were found in the genera *Cladosporia*, *Trichoderma*, *Chaetomium* and *Penicillium*, whereas pathogen facilitators, which exacerbated the disease pattern, although they were not pathogenic themselves, were observed in the genera *Epicoccum*, *Alternaria* and *Cladosporia* [285]. Interestingly, this can be leveraged as another tool for plant competition: for example, *Penicillium raistrickii* was a pathogen antagonist for *P. trichocarpa* in this study, but it also was earlier described as a pathogen facilitator for *Pinus ponderosa* [286].

How this level of competition works is shown in Figure 10.

The boundary between the normal microbiome and pathogens is an intriguing one. Indeed, since commensals and symbionts carry the same PAMPs as pathogens, they could be detected by PRRs and trigger PTI, but, in reality, this does not happen because microbes (1) mitigate the immune response by producing compounds that capture both ROS and organic acids produced by the plant, (2) modify their PAMPs to reduce their recognition efficiency, and (3) contain fewer trigger molecules on the membrane than their free-living relatives. This indicates a significant co-evolution of plants and their microbial symbionts [271]. It should be realized that the endophytes of some plants may be pathogens for others. The A *F. culmorum* strain isolated from *P. trichocarpa* leaves was identical to that of a wheat pathogen causing blight and rot. In the same research, in the leaf microbiome of *P. trichocarpa*, sequencing revealed representatives of 56 taxa, including *Cryptodiaporthe pulchella*, *Knufia cryptophialidica*, *Pyrenophora tritici-repentis*, *Phaeosphaeria pontiformis*, *Phaeosphaeria nodorum*, *Phoma macrostoma*, *B. cinerea*, *Elytroderma deformans*, *Taphrina* spp., *Ciborinia camelliae*, *Ramularia pratensis*, etc. They are pathogens to other plants in the same region, Pacific Northwest USA, including those in the genera *Populus* and *Salix*. It is likely that carrying such strains may be another way that poplars use to compete with other plants [57]. A great number of potential pathogens and saprotrophs were found among fungal genera when analyzing early poplar microbiome assembly [270]. And, as discussed above, under certain conditions, such as climate change, endophytes can change their life strategy and become pathogens [287]. And it appears that a similar ecological role may be played by the transfer of some viruses, such as earlier discussed begomoviruses and their satellites, which are asymptomatic to poplar and potentially pathogenic to other plants [93]. Thus, the boundary between poplar endophytes that protect the tree from pathogens and the pathogens themselves is quite blurred.

### 4.2. Phytophages

Despite the traditional view of the damage phytophages cause to plants, there are examples where plant-eating organisms are beneficial. For poplars, there are several such examples.

In addition to endophytes, phytophagous animals can also perform a protective function. For example, it was shown that caterpillars of *Lymantria dispar*, which are pests of poplars, are more willing to eat leaves of *P. nigra* infected with the rust fungus *M. larici-populina*, and, on such leaves, they first of all eat the sporangia of the fungus. The attractant for caterpillars is mannitol, which is more abundant in infected leaves, and the reason for such behavior is that the spores of the fungus have a higher nitrogen content, a more suitable ratio of amino acids for the insect body and a high concentration of B vitamins: B3, B5, B7 and B10 [288].

Pathogens also affect the arthropod community associated with a plant: in a study on *P. fremontii*, *P. angustifolia* and their F1 hybrid on the one hand, and the fungus *Drepanopeziza populi* on the other, it was found that both species diversity and the number of arthropod individuals were reduced in those plants exposed to the pathogen, and not only first-order consumers but also predators were affected. The main reason for this decline in diversity was infection-induced leaf fall. Thus, there is an “equalization” of poplars in terms of adaptability: more pathogen-resistant plants lose their advantage over less resistant ones under herbivory conditions, as infection significantly reduces phytophagy levels [289]. This may also explain why plants do not develop “absolute” resistance to pathogens. Finally, as we discussed earlier, carrying certain viruses (LdNPVs) may be directly dangerous to the phytophage itself (*Lymantria dispar*) and, thus, provide an advantage to the plant [94]. Other microorganisms, including those pathogenic to poplar, may be involved in similar defense mechanisms against phytophages.

Finally, poplars can, with the help of their secondary metabolites, protect associated insects from their pathogens. For example, in a research on the honeybee *Apis mellifera*, six different 3-acyl-dihydroflavonols, which originate from *P. fremontii* and several other poplar trees in North America and are incorporated into bee propolis, were found to be active against the bacterium *Paenibacillus larvae* that causes American foulbrood and the fungus *Ascosphaera apis* that causes chalkbrood [290].

Figure 11 shows the complex network of interactions between poplar, associated microorganisms and arthropods.

Thus, on the one hand, phytophagous insects may reduce pathogenic pressure on the plant, and infections may lower phytophagy. The plant can both contribute to the infestation of insects with dangerous diseases and give increased resistance to them. All of this demonstrates the immense complexity of the ecosystem consisting of plants, phytopathogens, arthropods and arthropod pathogens and emphasizes the need for its further study.

### 4.3. Elemental Defense Hypothesis

Not only living organisms but also abiotic factors may contribute to the resistance of plants, including poplars, to infections. Elemental defense hypothesis claims that plants can use chemical elements to protect themselves against pathogens. 

In support of this hypothesis, leaves of *P. yunnanensis* grown under conditions of excess Cd^2+^ had a higher resistance to both phytophages (*Spodoptera exigua* and *Botyodes diniasalis*) and the pathogenic fungus *Pestalotiopsis microspora*, while they did not differ from the control ones in the concentration of secondary metabolites [291,292]. It is also interesting that the degree of cadmium protection of poplar from both phytophages and leaf pathogens (*Pestalotiopsis microspora*) depends on other elements of mineral nutrition: for example, when exogenous nitrogen was applied, the plant accumulated more Cd in leaves, which were, thus, less eaten by insect larvae or affected by fungus [293]. 

Very intriguingly, Cd-mediated defense can depend on the gender of the plant. In the experiment, male and female individuals of *P. deltoides* were grown on Cd-contaminated soil. Moreover, male individuals not only accumulated more Cd in leaves but were also more resistant to the pathogen *Pestalotiopsis microspora*. This may be due not only to the direct toxicity of Cd, but also to indirect effects through the microbiome, which appeared healthier in male plants than female plants when exposed to Cd [294]. In addition, the species *P. deltoides* shows that males are more resistant to leaf herbivores and more susceptible to root herbivores compared to their female relatives under soil stress in the presence of this heavy metal. Such sex differences in resistance to herbivores are most likely due to differences in the patterns of Cd accumulation and distribution in plants of different sexes [295].

The opposite situation was reported for the other metal. The authors grew *P. simonii* seedlings on a medium with different zinc concentrations and examined the effect on how vulnerable they would be to rust disease. It turned out that Zn-stress not only reduced seedling biomass, but also increased their susceptibility to the disease, which is probably associated with a decrease in the natural chemical defense of the plant organism [296]. 

Figure 12 shows how the elemental defence hypothesis works in poplars.

Whether an element protects against pathogens probably depends on the chemical and biological properties of the element itself—for example, Cd^2+^ may be more toxic to phytopathogens than Zn^2+^, but exposure of the plant itself to Zn^2+^ may be sufficient to weaken its own chemical defense mechanisms. It is also possible that different pathogen groups and different poplar genotypes vary in their resistance to different heavy metals. Thus, the elementary defense hypothesis may be appropriate in some cases and completely unacceptable in others, and it all depends on the specifics of a particular system.

## 5. The Sex of the Tree as a Factor in Determining the Effectiveness of Plant Protection

Representatives of the genus *Populus* belong to dicotyledonous plants, which implies that the plants have two heterogamous sexes—male and female. The *ARR17* gene [297], located in the sex determination region (SDR), is known to be a master regulator of sex in poplar, and this was proven by the knockout of *ARR17* using the CRISPR-Cas system. In most poplars, the feminizing factor *popARR17* is not located in the SDR but as its negative regulator [298]. Poplars, depending on sex, may respond differently to abiotic factors such as drought, soil salinity, etc. [299,300]. Such factors include mineral acquisition and distribution in the plant. Nitrogen is an essential inorganic element present in almost all vital organic compounds, and its uptake and translocation in plant tissues appear to be dependent on the sex of the tree [301]; nitrogen is generally more actively accumulated by females [302]. Male individuals respond faster to salt stress in the presence of nitrogen, as suggested by the results of the differential expression assessment of multiple genes [303]. But under salt stress, for example, the expression of genes related to isoprenoid biosynthesis was elevated in female individuals [304]. Looking at the effects on poplar defense mechanisms, the representation of pipecolic and N-hydroxypipecolic acids increased, and SA and jasmonoyl isoleucine decreased in xylem sap when nitrogen availability increased; a number of secondary metabolites (salicinoids, phenylpropanoids, phenolic compounds, flavonoids and benzoates), with the exception of coumarins, were more actively synthesized under nitrogen deficiency [305]. However, in another study, different results were obtained: nitrogen limitation increased the synthesis of several secondary metabolites, which include phenolic compounds, in roots, except salicylates in leaves, with nitrogen concentration in leaves being negatively correlated with flavonoid and tannin concentrations [302]. Rather, the physiological response to nitrogen deficiency depends on the plant organ, and it is heterogeneous throughout the plant, suggesting a strategy by which the plant can survive under unfavorable conditions. In addition to nitrogen, the biosynthesis of phenolic compounds is also influenced by phosphorus concentration. Phosphorus has a more pronounced effect than nitrogen on the accumulation of flavonoids and tannins in poplar leaves, and this effect is stronger in female than in male individuals; however, it has only a minor effect on the synthesis of phenolic compounds [302]. 

Considering that females grow worse than males in cases of abiotic stress, while accumulating nitrogen more actively, it can be assumed that they spend mineral resources on the synthesis of secondary compounds for protection, which is indirectly confirmed by experimental data.

Despite the lack of convincing evidence that the efficiency of realization of poplar defense responses to pathogenic infection is sex-dependent, the above data suggest that sex-dependent response to abiotic stress and, consequently, changes in compound biosynthesis depending on nutrient availability may lead to differentially expressed defense responses. Secondary metabolites, as discussed above, are an important factor in the realization of plant defense mechanisms, and in a situation where stress or the concentration of available nutrients affects their biosynthesis, the plant becomes more vulnerable to the pathogen, and sexual physiology mediates this effect. This is also logical from the point of view that under unfavorable conditions, the quality of life of any organism is generally reduced, and defense systems are no exception. 

Female plants are more sensitive to drought, which is expressed in a pronounced restriction in their growth compared to the opposite sex. As in the case of soil salinization, trees of different sexes react in different ways to the lack of moisture at the transcriptome and metabolome levels, which is expressed, among other things, in the selective preference of insects as food: universal biotrophs in case of drought preferred to eat the leaves of a female plant, and specialized biotrophs—male leaves; in the absence of drought, gender preferences were diametrically opposed [306]. These data indicate that in the case of drought, female poplar individuals become unpleasant for specialized insects, but, at the same time, the individual does not become repulsive for universal biotrophs. This is consistent with the data in the work, according to which, under conditions of drought, the biosynthesis of flavonoids and alkaloids in females decreases more markedly than in males, and as a result, resistance to universal biotrophs sharply decreases, since the concentration of compounds in the leaves of the plant that could restrict nutrition is lower than in the absence of stress. Specialized pests are probably guided by the appearance of the plant, in addition to the presence of toxic compounds in the tissues, in choosing a food source, and since the male plant grows better, it, therefore, looks more attractive to the insect. Specialized biotrophs react, as a rule, to a certain group of compounds synthesized by the plant, and normally, apparently, their concentration is high enough in male plants for insects to prefer trees of this sex to a lesser extent, while with the onset of drought, the concentration drops to such a level that insects can feed on these plants more actively. From this position, undoubtedly, male poplars are more resistant to specialized types of pests.

Provided that plants obviously utilize nutrient resources and follow different developmental strategies (males grow more actively and females preferentially synthesize secondary metabolites and grow more poorly), microorganisms that normally reside on the plant surface and perform, among other things, tissue defense against invading pathogens, are sensitive to these differences and infect plants of different sexes differently. The phyllosphere community is more changeable than the rhizosphere one, and it reacts more strongly to any changes around [307]. Both females and males of *P. cathayana* may have unique phyllosphere bacterial and fungal microbiota, e.g., the bacteria Gemmata spp. and fungi *Pringsheimia* spp. are found in females, while the bacteria *Chitinophaga* spp. and fungi *Phaeococcomyces* spp. are found in males; differences in the relative abundance of bacteria of the phylums Proteobacteria and Planctomycetota and fungi of the phyla Ascomycota and Basidiomycota were also found: some species of bacteria of the genera *Spirosoma* and *Amnibacterium* and fungi of the genera *Venturia*, *Suillus* and *Elmerina* were reliably more represented in male poplar samples. And the number of fungi of genera *Phoma* and *Aureobasidium* spp. was significantly higher in female plants [308]. A lack of moisture, as described earlier, significantly affects the allocation of resources, particularly the synthesis of bioactive compounds involved in plant defense mechanisms against pathogens. However, the assumption that it is chemical compounds that play a fundamental role in defense against pathogens in the presence of drought as an abiotic stressor has proven incorrect: in a study conducted on *P. deltoides*, females under drought conditions showed greater resistance to the pathogen mainly due to a shift in the composition of the phyllosphere microbiome, with females being more susceptible to pathogen infection than males in the absence of drought [309]. Under drought conditions, the activity of metabolite synthesis is markedly reduced, so the efficiency of the chemical response of the plant is reduced, while the phyllosphere microbiome is also altered, taking on the role of a “border control” that limits the ability of the pathogen to develop on the leaf under conditions of pronounced competition. 

Root growth is the main way to enhance the production of nutrients from the soil for woody plants. And with an increase in the volume of root tissue, therefore, the concentration of synthesized substances in the surrounding soil increases, which, undoubtedly, should have a certain effect on the organisms inhabiting the roots or rhizosphere. The soil is a medium that closely links both the organisms directly inhabiting it and the plants occupying this niche and found in each other’s neighborhoods, and these interactions can be both positive and oppressive. Using *P. cathayana* as an example, it was shown that due to the synthesis and release of various phenolic compounds into the soil, females suppress the growth of poplar’s same-sex neighbors and negatively affect the bacterial community of roots, while the presence of males reduces the content of phenolic compounds in the soil, promoting the survival of organisms and stimulating the growth of the neighboring female, an effect that is also supported by the diverse composition of the neighboring male’s microbiome [310]. The demands on nutritional resources of poplars of different sexes also play a role, affecting the composition of the rhizosphere, but access to nitrogen plays a lesser role than the plant itself and its protective mechanisms [307]. 

Thus, mixed poplar plantations enhance the “population immunity” provided by the diversity of the rhizosphere community, whereas populations consisting of only females should be much more sensitive to certain groups of pathogens due to the suppression of this community. A rather important observation is that it is the active synthesis of secondary metabolites that is a negative factor for female individuals. From the plant’s point of view, the synthesis of secondary metabolites is a coin with two sides: on the one hand, chemical compounds suppress the growth of pathogens, while on the other hand, the growth of beneficial bacteria is impaired. 

Summarizing the above data regarding differences in the natural resistance of poplars depending on gender, if the poplar is planted in order to obtain an economically valuable resource or as a tree for landscaping, it makes sense to plant either mixed-sex populations or male populations, since female plants may be less resistant to pathogens and more sensitive to resource scarcity; this will lead to slower growth, which, as a result, will lead to the loss of valuable biomass. In the context of choosing poplar lines with valuable traits of interest for further mass planting, it is possible to change the sex of the plant by knocking out the ARR17 gene in vitro using the CRISPR/Cas9 genomic editing system, as shown earlier [297], and vegetatively propagate plants of the preferred sex to obtain a large number of genetically identical seedlings.

## 6. Development of Sustainable Genetically Modified Poplars—Prospects and Challenges

### 6.1. Methods for Creating Genetically Modified Poplars and the Potential of CRISPR/Cas-Based Genomic Editing

Knowledge of the basic mechanisms of poplar disease resistance can and should be used to obtain more resistant plants by faster and more efficient methods than traditional breeding methods. As is well known, breeding involves obtaining resistant genotypes that maintain trait stability over a number of generations, which is a labor-intensive and time-consuming process. For example, instead of increasing the stability of poplars already used in forestry, the adaptation of supposedly more resistant and wild species, such as P. euphratica, discussed above, could be accelerated, just as it is proposed to accelerate the domestication of wild plant forms instead of increasing the stability of domestic varieties [135]. But there is genetic engineering, an effective way to more quickly obtain a variety of source material for further breeding, which undoubtedly reduces the time to obtain a stable plant line. Engineering methods can be roughly divided into two groups: genetic engineering of poplars themselves, which allows for strengthening their immune response, and microbiome engineering, which can be considered as a more environmentally friendly substitute for fungicides, and the tree can be supplied with the necessary bacteria several times during its life. The adaptation of any technology should be based on experience in developing other plants with improved disease resistance.

In general, engineering based on changing the expression of targeted genes includes several fundamentally different strategies: introducing new sequences into the plant genome or editing existing ones (Table 1). To increase the resistance of poplars, regardless of the final implemented protection mechanism, whether it is an increase in the synthesis of secondary metabolites, a change in signaling pathways or the introduction of proteins mediating plant immunity by pathogen, various genetic engineering methods are used, which have their advantages and disadvantages.

The simplest, most efficient and widespread way to obtain transgenic plants is their transformation using agrobacteria or agrotransformation. However, this approach has a number of disadvantages: in particular, the risk of obtaining chimeric tissues, and the risk of a lack of transgene transfer to offspring in cases of the incomplete transformation of generative tissues. The risk of chimeras in poplar can be reduced via isolation and transformation of individual protoplasts [313]. Protocols for regeneration of whole plants from poplar protoplasts have been described previously [314]. Potential limitations of agrotransformation also include the random integration of T-DNA into the plant genome, which can lead to either overexpression of the transgene or its silencing depending on the functional context of the chromatin; the sensitivity of plant lines to *Agrobacterium* strains can vary dramatically (some lines can be completely resistant, so their agrotransformation is impossible) and cannot be used to alter several genomic targets at once [315].

CRISPR/Cas-based genome editing technologies are being developed for plants, and poplar in particular, that can overcome some of the limitations of agrotransformation. The CRISPR/Cas9 type IIA system from *Streptococcus pyogenes* is currently the most widely used, both in basic research and in various applications in biotechnology and medicine. Genome editing approaches utilize the system as two components: a multi-domain Cas9 of 1368 amino acids and sgRNA, which is a hybrid of natural cRNA and trasRNA. The cRNA includes a spacer that directs Cas9 to a genomic target. The Cas9-mediated recognition of the genomic target requires a short protospacer motif (PAM) located near the genomic target. The general mechanism is as follows: the Cas9 complex with sgRNA moves either by three-dimensional diffusion in space or by one-dimensional diffusion along the DNA strand; then, it finds the PAM, after which the sgRNA spacer binds due to complementarity with the target DNA strand. And if they are completely or nearly complementary, SpCas9 makes a double-strand break and dissociates from the DNA strand, which usually results in gene knockout [316]. In addition to Cas9 endonuclease, its modifications—base editors and primer editors—are used to make point changes in the genome [317]. The next level is epigenetic editors that do not change the genome sequence and affect the activity of genes at the levels of their transcription, splicing and translation [318]. SpCas9 orthologues from other bacteria are also used, as well as CRISPR/Cas systems of other types [319]. A second frequently used edit is Cas12a [320], a smaller protein that has only one nuclease domain and, therefore, leaves sticky rather than blunt ends.

The advantages of using CRISPR/Cas systems over agrobacterial transgenesis include the following: precise editing of genome regions; the ability to make a variety of genomic changes (single base changes, small indels, gene knockouts, deletions of large portions of the genome); use as mRNA or RNP particles or non-integrating vectors based on the yellow dwarf virus (BeYDV); and the ability to work simultaneously with multiple targets, making it possible to engineer signaling or metabolic pathways or multiple traits.

Currently, CRISPR/Cas is widely used in the genetic engineering of herbaceous plants to increase productivity, improve resistance to various stress factors, improve aesthetic appeal, etc. However, the technical difficulties of using CRISPR-Cas systems in both herbaceous and woody plants are related to delivery. To date, the following CRISPR/Cas9 system delivery approaches have been used in plant cells: (1) agrotransformation, (2) biolytic bombardment, (3) electroporation, (4) nanoparticle application, (5) polyethylene glycol (PEG), (6) viral delivery, and (7) meristem transplantation for editing into a Cas9 overexpressing plant [318]. Each of these methods has disadvantages in obtaining genetically homogeneous edited poplar lines, which can be overcome by obtaining protoplasts, and CRISPR/Cas-editing followed by whole-plant regeneration [312]. The development of CRISPR/Cas9-based genome editing technologies using plant cell cultures may completely change the concept of poplar breeding.

Next, we discuss several examples of using CRISPR/Cas systems to obtain more pathogen-resistant poplar lines.

One of the CRISPR/Cas-engineered poplar genotypes with increased resistance to pathogens is MYB57, discussed above. This mutant contains an increased amount of flavonoids because MYB57 is an inhibitor of their biosynthesis [204].

CRISPR/Cas9 was used to simultaneously modify six genes of the poplar lignin biosynthetic pathway [321]. This work demonstrated the principal possibility of targeting multiple genes and, thus, engineering biochemical pathways in poplar. In addition, during the work, key enzymes of lignin biosynthesis were identified, which can be used to increase lignin production and, thus, enhance the mechanisms of the passive protection of poplar from pathogens by increasing the mechanical resistance of poplar to pathogens and activating the synthesis of secondary metabolites dependent on this pathway. An increase in the accumulation of lignin will definitely lead to an increase in the economically valuable biomass of poplar, which is extremely important for the production of biofuels and the possibility of using wood for other industrial purposes.

Because Cas9 is known for high levels of off-target activity [322], researchers are exploring other more precise Cas editors. Recently, efficient (up to 70%) multi-targeted editing in poplar has been demonstrated using different Cas12a editors [323]. Although in these studies, the authors targeted genes unrelated to poplar immunity, they are important because they showed the principal possibility of using more precise Cas12a editors that can be applied to edit very similar in-sequence paralogous genes in order to tune the activity of only those associated with pathogenic processes.

Another way to avoid harmful Cas9 off-targets is through the use of base editors. In recent work, a Cas9-based cytosine base editor was used to knockout PLATZ (plant AT-rich protein and zinc-binding protein), encoding a transcription factor involved in plant response to fungal pathogens, in the poplar hybrid *P. tremula* × *P. alba* clone INRA 717-1B4 [312]. The knockout was accomplished by introducing premature stop codons into the PLATZ coding region via C-to-T transformations with an efficiency of 13–14%. PLATZ was previously shown to be associated with numerous disease-related genes based on an analysis of the BESC-22 resistance genotype and the BESC-801 susceptibility genotype to *Sphaerulina musiva* in *P. trichocarpa* [257]. Thus, this work provides a model plant for further elucidation of the molecular mechanisms associated with PLATZ-mediated resistance of poplar to pathogens.

CRISPR-based artificial transcription factors provide another way to effectively and reversibly influence gene activity without changing their sequence. For example, CRISPR activator was applied to increase the expression of two target endogenous genes, TPX2 and LecRLK-G, which play an important role in plant growth and the defense response of poplar trees [312]. The application of CRISPRa resulted in a 1.2- to 7-fold increase in expression of the target genes through temporal expression in poplar protoplasts and stable *Agrobacterium*-mediated transformation.

To date, only a few studies have been conducted on the CRISPR/Cas-mediated editing of poplar genes related to pathogen defense. Significantly more studies have been conducted on herbaceous plants (see excellent reviews [324,325,326]). However, herbaceous plants can be used as a model to study and improve pathogen defense mechanisms in trees such as poplar. Therefore, we next discuss several examples of candidate target genes that can be used as targets of CRISPR/Cas systems to improve poplar immunity.

There are genes that confer pathogen resistance to plants (including poplars) when overexpressed or added to the plant genome as transgenes. However, the addition of pathogen-protective genes, such as BbChit1 and LJAMP2, may lead to coevolution of pathogens and, thus, be ineffective in the long term [324,325]. In addition, the use of plants with foreign DNA is difficult from a legal point of view in several countries [327]. Therefore, gene knockout strategies are more preferable. In plants, CRISPR/Cas systems are better suited for gene knockout and are commonly used to knock out S genes, the presence of which is associated with pathogen susceptibility [318].

One of the most studied targets for CRISPR/Cas-mediated plant resistance are genes that increase plant susceptibility to fungi, as fungal pathogens are perhaps the most common and severe biotic stress to plants. A well-known example is the MLO genes already briefly mentioned above, the knockout of which in wheat [328], tomato [329] and grape [330] increases the resistance of the respective plants to powdery mildew. As mentioned above, *P. trichocarpa* has 26 MLOs. It has been suggested that PtMLOs 17, 18, 19 and 24 may be associated with resistance to powdery mildew [251]. Given the ability of the CRISRP/Cas systems to operate simultaneously at multiple loci, the knockout of several MLO genes in poplar appears to be a feasible task.

The *NFXL1* gene in wheat is a TF inducible by the mycotoxin deoxynivalenol, so its knockout in wheat increases its resistance to *F. graminearum* [331]. The *NFXL1* gene, encoding a putative suppressor of elicitor-induced defense responses involved in SA and ABA signaling, has been previously described in *P. tremula* [332] and may, therefore, be a potential target for CRISPR/Cas editing in poplar.

Finally, LOX3-mutant maize was more resistant to the biotrophic pathogen *Ustilago maydis* because mutant plants have reduced activity of the JA pathway, which is competitive with SA [333]. It can be hypothesized that the knockout of LOX genes would increase poplar resistance to biotrophic pathogens such as rust fungi. However, SA-JA antagonism is less pronounced in poplars, and, in addition, LOX knockouts may increase poplar susceptibility to necrotrophic pathogens, so we do not consider these targets promising.

There are fewer studies that have examined increased plant resistance in bacteria than in fungi.

*SWEET* genes encode sucrose transporters into the apoplast, e.g., from sieve elements [334]. These genes are S-genes in rice during *Xanthomonas oryzae* invasion and are induced by the bacterial effector PthXo2. The knockout of the *OsSWEET13* gene increases rice resistance to bacterial infection [335]. Similar results were obtained when the promoters of OsSWEET14 and OsSWEET11 genes were altered so that PthXo2 could not bind to them [336]. Since *Xanthomonas* causes poplar diseases and 27 SWEET genes have been found in the genome of *P. trichocarpa* [337], they can be considered as potential targets for CRISPR/Cas modification, although their role in poplar susceptibility to bacterial diseases is not currently understood.

The LATERAL ORGAN BOUNDARIES 1 (LOB1) gene, which is also an S gene in *Citrus paradisi* and *C. sinensis*, makes them susceptible to *Xanthomonas citri* infection, which causes citrus canker. The mutation of the LOB1 promoter so that the bacterial effector PthA4 cannot interact with it protects plants from this pathogen [338,339,340]. Genes encoding LOB-domain-containing TFs are present in the poplar genome and regulate its secondary growth [341]. It is likely that the CRISPR/Cas-based knockout of LOB-related genes or their promoter modification could protect poplar from at least some types of cankers.

Thus, the strategy of modifying S-gene promoters so that pathogenic effectors cannot bind to them is effective and is currently prevailing among CRISPR/Cas-based approaches for increasing plant resistance to bacterial pathogens.

Finally, studies with viruses, most of which have been conducted on *Nicotiana benthamiana*, have shown that genes that enable replication and assembly of the virus capsid [342,343] and host plant genes required for the translation of viral proteins, such as elF4E [344], elf(iso)4E [345] and elF4G [346], are promising targets. Elf4E, elf(iso)4E, elf4G, etc. are translation initiation factors, so, probably all plants, including *Populus*, possess them [347], and the CRISPR/Cas-mediated modulation of their activity can be used to prevent viral mosaic disease in poplar trees.

What conclusions can be drawn about future directions for the genetic engineering of poplars? Based on all the above articles, the strategies can be as follows:

To introduce a fundamentally new receptor (e.g., AtRLP23) or defence protein (BbChit1, LJAMP2, MsrA2, etc.) into the genome; the same can be achieved by the overexpression of the native defensin gene (PtDef). This approach is almost guaranteed to increase plant protection with the minimal tradeoffs, so we consider it to be the most simple and reliable. The result can be achieved using agrobacterial editing, but it is better to use CRISPR/Cas for a more precise introduction of transgenes.

To influence the activity of transcription factors, mainly from the MYB and WRKY families, the overexpression of MYB factors that activate the biosynthesis of secondary metabolites can be used to increase the concentration of flavonoids in the plant. This can be achieved by *Agrobacterium*-mediated transformation. The knockout of MYBs that inhibit the biosynthesis of secondary metabolites can be accomplished using CRISPR/Cas. The overexpression of WRKYs from different families may increase disease resistance in poplar, and, although there is evidence that such overexpression in *Arabidopsis* may increase susceptibility to some pathogens due to SA-JA antagonism, this may not be an issue for poplar because SA and JA probably have a positive feedback in poplar.MicroRNA editing. It can be carried out through overexpression or STTM-mediated knockdown. Transgenic constructs can be introduced using agrobacteria. This field is just developing now, and there are only a few related works.MLO gene knockout. There are four candidate genes whose knockout can provide poplar resistance to powdery mildew: *PtMLO17*, *18*, *19* and *24*. Agrobacterium must be used for delivery, and the editing itself is performed using the CRISPR/Cas system.The knockout of genes from the *SWEET*, *LOB*, etc. families discussed in this chapter or already carried out on other plants. But to do this, it is necessary to find exactly those homologues in the poplar genome whose knockout will provide it with resistance to certain diseases. The protocol will be similar to that used for *MLO*s: transformation using agrobacteria and CRISPR/Cas-mediated knockout.

### 6.2. Possible Approaches to Engineering the White Poplar Microbiome

Obtaining genome-edited poplar trees is a very long process, so it is important to consider more rapid approaches to improve immunity and other characteristics of poplars by altering the composition of their microbiome. Microbiome engineering involves the directed modification of the symbiotic microbial community associated with a plant. In a general sense, this approach dates back to ancient times when people invented various fertilizers: organic and inorganic, acting as prebiotics (i.e., promoting the growth of pre-existing beneficial organisms) and probiotics (i.e., directly introducing beneficial organisms—e.g., manure and compost). The role of endophytic bacteria in the context of pathogen resistance has been discussed above, and current approaches to microbiome engineering include (1) the use of root exudates and organic substances (coumarins, benzoxazinoids, 1-aminocyclopropane-1-carboxylic acid) that can attract desirable bacteria from the soil; (2) artificial microbial consortia (AMCs)—creating artificial microbiomes as combinations of essential bacteria; (3) transplantation of microbiomes from donor plants to recipients; (4) microbe-optimized plants—creating genetically engineered plants that secrete substances that attract beneficial microbes; (5) plant-optimized microbes and microbiomes—creating such bacteria (including genetically engineered ones) and their communities that are ideally suited to a given plant. It is highly desirable to combine strategies for matching plants, microbiome and soil to each other, starting from the seed stage rather than the adult plant. There are interesting technical solutions for certain situations, such as phyllosphere biocontrol—spraying plant leaves with beneficial bacteria—which helps to maintain a normal leaf microbiome and protect against leaf pathogens [272,348].

There are few examples of microbiome engineering performed on poplars. At present, most works are related to the study of poplar microbiome composition and the assessment of the influence of individual “useful” components on the plant. For example, the presence of diazotrophic bacteria has been demonstrated for *P. trichocarpa* [349], the use of which, in the long term, can certainly allow for increasing the productive biomass of the plant due to a greater availability of nitrogen. On the other hand, there are experimental works that have studied the effect of either a single bacterial culture or several on poplar and its bacterial environment. Seedlings of *Populus* × *euramericana* cultivar “Neva” were inoculated with a bacterium derived from the rhizosphere of poplar, *Bacillus subtilis* T6-1, which has a strong antagonistic effect against poplar rot pathogens *F. oxysporum, F. sol. oxysporum, F. sol. oxysporum, F. solani, R. solani, A. alternata* and *Phytophthora capsici*; this bacterium stimulated poplar growth and also had antagonistic effects against members of the genus *Rhizoctonia* [350]. Another member of the genus, *Bacillus cereus* strain BJS-1-3, which can act as a biocontrol element against poplar pathogens, also promoted tree growth after inoculation and suppressed the growth of pathogenic species in the rhizosphere [351]. In another case, a simplified root microbiome (consisting of *Pseudomonas* and *Burkholderia* bacterial strains) derived from *P. deltoides* was inoculated on aseptic cultures of *A. taliana* and *P. deltoides*, resulting in increased root density, enhanced photosynthetic activity and a number of metabolic pathways [352]. Importantly, for the non-natural host, *Arabidopsis*, the result was similar, allowing us to consider valuable strains derived from other plants as well. Mycorrhizal inoculation has also shown positive effects on poplar growth, inducing tree growth and biomass accumulation under growing conditions on heavy-metal-contaminated soils [353], and mycorrhizal helper bacteria, such as some strains of the genus *Pseudomonas*, can be used to improve mycorrhiza formation [354]. 

It is important to realize that microbiome engineering will not be an available method to improve poplar in all cases. Some poplar endophytes, such as *F. culmorum* [57], may be pathogens of other plants, which is an important limitation for their application. Regarding this pathogen, the *B. vietnamiensis* WPB strain, also derived from poplar, is active against it and some other pathogens (*R. solani*, *Gaeumannomyces graminis var. tritici* and *Pythium ultimum*) [69,355]. 

In general, the plants themselves serve as a source of valuable bacteria with antagonistic activity against pathogens. With a sufficient studies on individual strains, it is possible to develop biopreparations for increasing the productivity of poplars.

### 6.3. Problems and Perspectives Related to the Application of Genome Editing Techniques to Increase Poplar Resistance to Pathogens

A combination of different strategies to enhance pathogen resistance appears promising. For example, a combined increase in SA and JA synthesis, despite their different roles in pathogen defense, does not antagonize and has a positive effect on poplar resistance to rust caused by *M. larici-populina* [120]. The formation of a symbiotic relationship of poplar resistant to the similar rust *M. larici-populina* with the ectomycorrhizal fungus *Hebeloma mesophaeum* resulted in the compensation of flavonoid synthesis compared to an infected plant without ectomycorrhiza [356]. A combination of genetic engineering techniques aimed at enhancing flavone synthesis and the use of symbionts will achieve more pronounced plant resistance to pathogens. However, the presence of SA, JA and ET may prevent the colonization of poplar roots by symbiotic fungi [357,358], which seems to be an adaptive response of the plant and is independent of the host fungus in terms of negative effects on it. But not all ectomycorrhizal symbioses can lead to the increased synthesis of secondary metabolites, as in the case of the ectomycorrhizal fungus *Glomus interradices*, and, in general, the formation of a symbiotic relationship with it may contribute to reduced insect resistance [359]. On the other hand, the modification of the plant genome may result in phenotypic changes that are almost impossible to predict. This fact requires a careful selection of candidate genes and plant transformation techniques to avoid undesirable effects.

In addition to technical and methodological challenges, there are also risks associated with the release of genetically modified organisms into the environment. Many microbial organisms that can be used to improve plant resistance can have harmful effects, including through horizontal gene transfer, which can ultimately lead to environmental disasters if used uncontrolled [360]. Rhizosphere engineering by a transplantation of the wild-type strain *P. fluorescens* 89B-27 and its genetically modified derivative had no significant effect on the bacterial community [361], but it should be borne in mind that this bacterial species is a normal representative of the soil microbiome and has long been known to have positive effects on a number of crops, and in a later experiment, the species and functional composition of the rhizosphere were altered by the introduction of two different strains of this bacterial species [362], but the composition of the basic soil microbiota did not change. The most reasonable way to modify the microbiome is to use those bacterial species and modified strains that are not pathogenic to humans, animals and other plants and generally exist as normal members of the soil community.

Much more intriguing is the question of how transgenic plants affect the environment. Cultivated genetically modified organisms are usually used in agriculture and grown in agrocenoses, i.e., in a human-controlled environment. When transgenic maize with a cloned Cry1Ah gene from *B. thuringiensis* was grown, no significant differences were found in the bacterial composition of the rhizosphere, but the metabolic profile was altered, with most of the altered metabolites being related to the metabolism of the plant itself rather than the pathways of the plant–bacteria relationship [363]. A similar result was obtained when growing the genetically modified potato variety Modena, tubers of which have altered starch content, but in this case, it is likely that the modulation of the microbial community was due to root exudate [364], which was also observed in an experiment on transgenic poplars [365]. Overall, a number of studies confirmed that rhizosphere composition can change only slightly, but the outcome still depends on which genes and metabolic pathways were involved [366,367,368].

The risk associated with changes in the composition of the beneficial microbiome can be mitigated by the judicious use of biologics and biofertilizers in agriculture, but there are also fewer human-controlled situations, such as the vegetative transfer of transgenes, hybridization of transgenic plants with wild species and accumulation of transgenes, which can eventually lead to the emergence of new weeds; the conservation of crop biodiversity is also at risk [369].

With perennial plants, the situation may be quite different, especially with woody forms: they grow for a long time with a lifelong impact on the environment, are less controlled by humans and can be planted outside agrocenoses. The effect of genetically modified *P. radiata* through the biolistic insertion of LEAFY and npt II genes on rhizosphere microbial communities of trees was found to be negligible [370], as in the case of cultivated annual plants. Modifications associated with increased lignin accumulation can affect colonization by ectomycorrhizal fungi, but these changes were also observed using conventional breeding methods rather than genetically engineered technologies [371], with changes in endosphere composition but not rhizosphere composition [372], and other modifications may show no change at all in the long term [373]. And when the level of lignin biosynthesis enzyme was reduced via RNA interference, the composition of fungal and bacterial communities of roots and the soil was changed [374]. In general, genetically modified poplars appear to modulate little or no microbial and fungal community compositions in soil and alter mainly the microbiome in response to plant tissue and environmental conditions [375]. 

Cloning of the *Cry1Ah1* gene to synthesize Bt-toxin, which confers insect resistance in poplars, can increase the final biomass of poplars and maintain their aesthetic appearance in ornamental plantings. Transgenic poplar synthesizing Bt-toxin has no adverse effects on the soil microbiome [376], and the risks associated with the spread of this gene from a plantation of genetically modified poplars are extremely low due to the negligible seed germination and small dispersal area [377], and it does not pose a threat to arthropod biodiversity in long-term experiments [378,379].

Taking into account the above results of studies on the environmental impact of transgenic plants, it is possible to minimize the burden by using those changes in the poplar genome that will minimally affect such a group of organisms as insects, since with the potential creation of plant lines capable of synthesizing toxic substances, these animals will suffer first. At the same time, in order to reduce the risks of spreading resistance genes among wild plants, it is possible to create sterile poplar lines that would not be able to produce offspring. Other approaches can be used for microbiome engineering as another tool to increase the resistance of poplars to pathogens: the rejection of any introduction of antibiotic resistance genes as, for example, selective genes, selection and modification mainly of those strains that may be natural inhabitants of the soil in a given area.

However, it should be noted that, at this stage, it is difficult to assess the possible environmental risks associated with transgenic plants, since the commercialization and use of genetically modified plants outside agricultural activities are legally limited in most countries, which significantly limits the possibility of studying the associated long-term consequences of using modified poplars. At the same time, the evidence to date suggests that it is at least possible to develop genetically modified poplars that, when integrated into the environment, will not lead to disaster. 

It is important to keep in mind that the functions of many sequences in plant genomes have not yet been unraveled, and further research is needed to understand which resilience approaches can be used alone or in combination, and which metabolic pathways they affect, to put environmental risks into perspective by expanding the knowledge of the interactions of specific natural entities.

The cultivation of genetically modified poplars is promising, not only from the point of view of studying the functions of individual parts of the genome, but also from the position of searching for new approaches to creating poplar cultures resistant to certain species or groups of pathogens, which will undoubtedly lead to an increase in plant productivity when grown in plantations for industrial use. If it is possible to minimize the negative impact of genetically engineered poplars on the environment, such trees can be planted for ornamental purposes under certain conditions. 

The results of such studies, especially those related to microbiome engineering, will undoubtedly become a source for the development of biological preparations, the use of which can reduce the use of agrochemicals that inhibit the growth of phytopathogens.

## 7. Conclusions

We sincerely hope that this review will have a real practical outlet and help researchers working with poplars in different parts of the world to establish sustainable genetic lines more efficiently and thoughtfully. This is important because poplar has tremendous application value and we anticipate that this will increase as humanity transitions to green energy. Therefore, below, we propose strategies that, after a detailed literature review, we assess as reasonable.

Accelerate the adaptation to cultivation in other regions of species with naturally high resistance to infection. For this purpose, the pathogenic resilience of poorly studied poplar species should be examined in more detail, and the most suitable ones should be selected. This approach should be implemented with great care and taking into account the specifics of the region, as the appearance of an invasive species can have a highly negative impact on the state of the ecosystem. It is also preferable to plant populations of male poplars or those consisting of both sexes for better plant survival.

Genetic lines growing in more southern regions are preferred. Faced with more biotic stresses, they are likely to be more resistant to pathogens in the north as well. We consider it important to continue fundamental research on the functions of unexplored poplar genome sequences in order to assess the possibility of manipulating them to increase the stability of poplars. The knockout of MLO genes and other susceptibility genes, as similar experiments, have already been successfully conducted on other plants. Also, changing the expression of transcription factors of the WRKY and MYB families through the use of different genetic engineering methods can lead to increased resistance to various pathogens of a fungal nature. When editing sections of the genome involved in biosynthetic pathways, one should be careful, since there is a high probability of undesirable side effects such as developmental disorders and deformities. In addition, if we exclude the possibility of transferring genes to other races, such an approach would be quite environmentally friendly. The transgenic expression of genes encoding individual defence proteins such as *BbChit1*, *LJAMP2* or *MsrA2*, as well as overexpression of poplar’s own defensin gene, results in less systemic changes in the plant but confers good, predictable and reliable protection against pathogens and is probably the most promising genetically engineered approach to improving disease resistance. Another quick and effective approach is the bioengineering of the poplar microbiome in order to introduce modified strains of microorganisms that can be an element of biocontrol. In order to find promising strains that can stimulate poplar protection from pathogen invasion, it is necessary to conduct exploratory studies to detect new microorganisms and evaluate their direct effect on poplar. However, genetically modified trees and their symbionts may carry certain risks that need to be carefully studied before the improved plants and their symbionts can be propagated in nature.

## Figures and Tables

**Figure 2 ijms-25-01308-f002:**
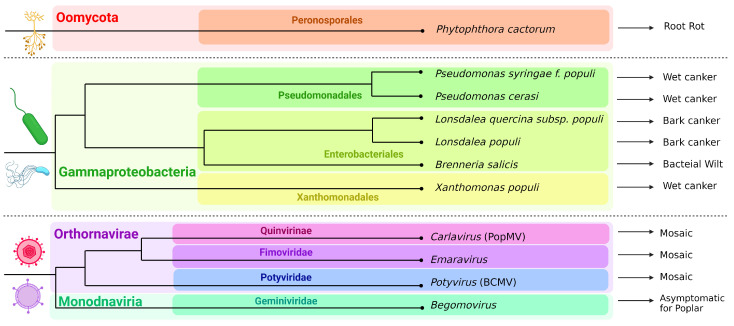
List of poplar non-fungal pathogens and the diseases caused by them. For the reader’s convenience, their evolutionary and systematic relationships are depicted in the form of a manually constructed cladogram using external data from NCBI Taxonomy and articles [95,96]. One rot-causing oomycete generalist, several pathogenic bacteria usually resulting in cankers and viruses, most of them manifest as mosaics, are shown on the phylograms. Among them, *Lonsdalea populi* and mosaic viruses cause the most damage and are most actively studied. Created with BioRender.com (accessed on 18 January 2024).

**Figure 3 ijms-25-01308-f003:**
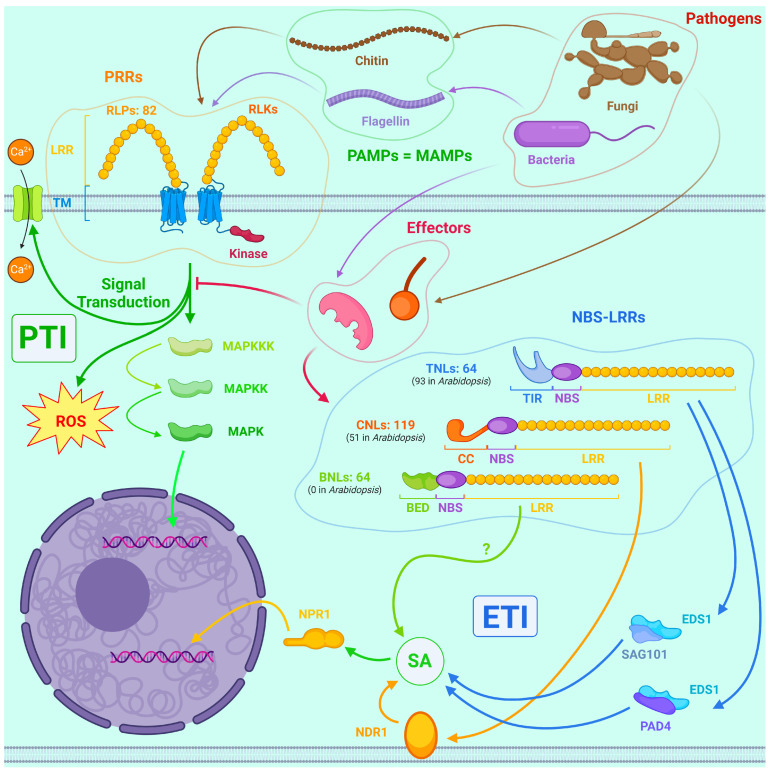
A general scheme of plant immunity with nuances for poplar. Pathogens secrete pathogen-associated molecular patterns (PAMPs), also referred to as microbe-associated molecular patterns (MAMPs) such as chitin (in fungi) or flagellin (in bacteria). The plant can detect the presence of PAMPs through two types of pattern recognition receptors (PRRs): the transmembrane leucine-rich repeat (LRR)-containing receptors, receptor-like proteins (RLPs) and receptor-like kinases (RLKs), followed by signaling involving (1) the mitogen-activated protein kinase (MAPK) cascade, (2) Ca^2+^, (3) reactive oxygen species (ROS), and, finally, PAMP-triggered immunity (PTI) is launched. PTI is a relatively weak and non-selective process, and many pathogens, particularly specialized pathogens, are adept at suppressing it with proteins called effectors. But the plant can perceive effectors by means of cytoplasmic receptors, which belong to the broad group of nucleotide-binding site (NBS)-LRRs and include groups of toll interleukin-like (TIR)-NBS-LRRs (TNLs), colied coil (CC)-NBS-LRRs (CNLs) and BED-NBS-LRRs (BNLs; the BED domain was named after the BEAF and DREF proteins from *Drosophila melanogaster*, in which it was first discovered), representatives of which differ in their N-terminal domain structure. Upon activation, TNLs transmit signals through Enhanced disease susceptibility1 (EDS1)/SENESCENCE-ASSOCIATED GENE101 (SAG101) and EDS1/PHYTOALEXIN DEFICIENT 4 (PAD4) protein complexes, CNLs do this via the membrane-bound NON-RACE-SPECIFIC DISEASE RESISTANCE1 (NDR1) protein, and BNLs do this in a poorly understood manner. This ultimately leads to the activation of salicylic acid (SA) signaling, which induces key defense mechanisms through the Nonexpressor of pathogenesis-related genes 1 (NPR1) protein. Among the physiological processes triggered by SA is effector-triggered immunity (ETI), a more potent response than PTI, during which serious reactions such as systemic acquired resistance (SAR) and hypersensitivity response (HR) are possible. Created with BioRender.com (accessed on 18 January 2024).

**Figure 4 ijms-25-01308-f004:**
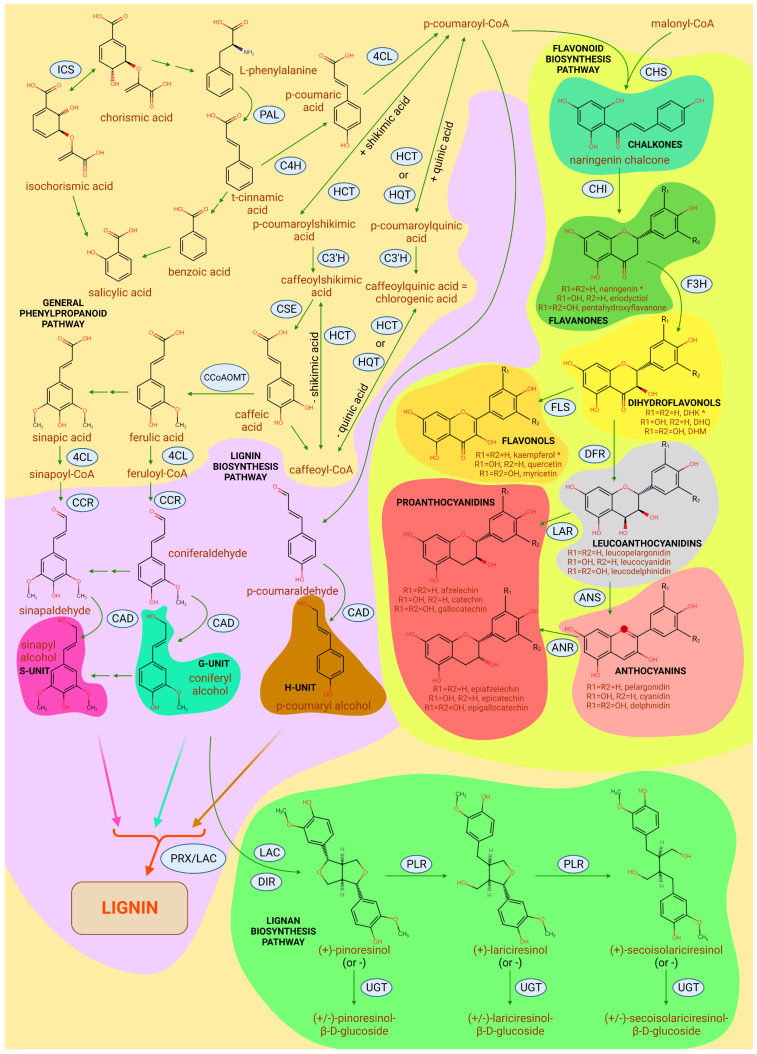
Illustration of the diversity of secondary metabolites providing protection for poplar against pathogens and the pathways of their biosynthesis. General phenylpropanoid pathway, as well as lignin, lignan and flavonoid biosynthesis pathways are shown. Depending on the poplar species, unique compounds can be found for them; classes of synthesized compounds are presented in the figure. The common phenylpropanoid pathway (shown in pale pink) starts with chorismic and isochorismic acids, which can be converted to SA, p-coumaric acid, etc. SA is a key biotic stress defense hormone, and its derivatives represent an important class of secondary metabolites. P-coumaric acid by successive transformations can yield sinapyl alcohol, coniferyl alcohol and p-coumaryl alcohol, which are S-, G- and H-subunits in lignin synthesis (shown in lilac), respectively; they are also precursors in the biosynthesis of lignans (shown in light green). In addition, p-coumaric acid can form chalcones and then, sequentially, other classes of flavonoids (shown in yellow-green): flavanones, dihydroflavanones, flavonols, leucoanthocyanidins, anthocyanins and proanthocyanidins. Created with BioRender.com (accessed on 18 January 2024).

**Figure 5 ijms-25-01308-f005:**
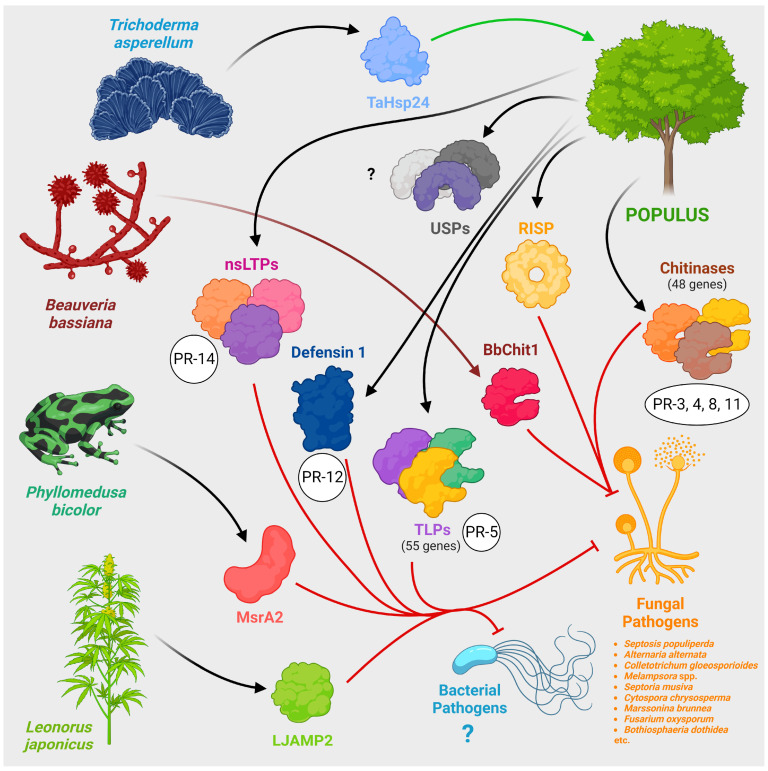
Plant defense proteins, including PRs. Both poplar innate proteins, such as chitinases, non-specific lipid transfer proteins (nsLTPs) and thaumatin-like proteins (TLPs), and proteins from other organisms, the transgenic expression of which, in poplars, can increase disease resistance are shown. Poplar defence against pathogens is provided by a number of proteins active against fungi and bacteria, many of which belong to the group of PR proteins: chitinases, defensins, nsLTPs, TLPs, rust-induced secreted protein (RISP), universal stress proteins (USPs) and others. Overexpression of genes of many of them can increase poplar disease resistance, as can expression of transgenes derived from specially selected organisms: BbChit1 which is a chitinase from the chitinolytic fungus *B. bassiana*, LJAMP2 which is an antimicrobial peptide from the medicinal plant *L. japonicus*, MsrA2 which is an N-modified dermaseptin β1 from the poisonous frog *P. bicolor*, and TaHsp24 which is a heat shock protein from the fungus *T. applanatum*. Examples of fungal pathogens that have been shown to be inhibited by these proteins are given; although these proteins have not been tested on bacterial pathogens of poplar yet, we hypothesize that many of them have antibacterial activity, as indicated by “?” [42]. The images of organisms are schematics and are not accurate biological drawings. Created with BioRender.com (accessed on 18 January 2024).

**Figure 6 ijms-25-01308-f006:**
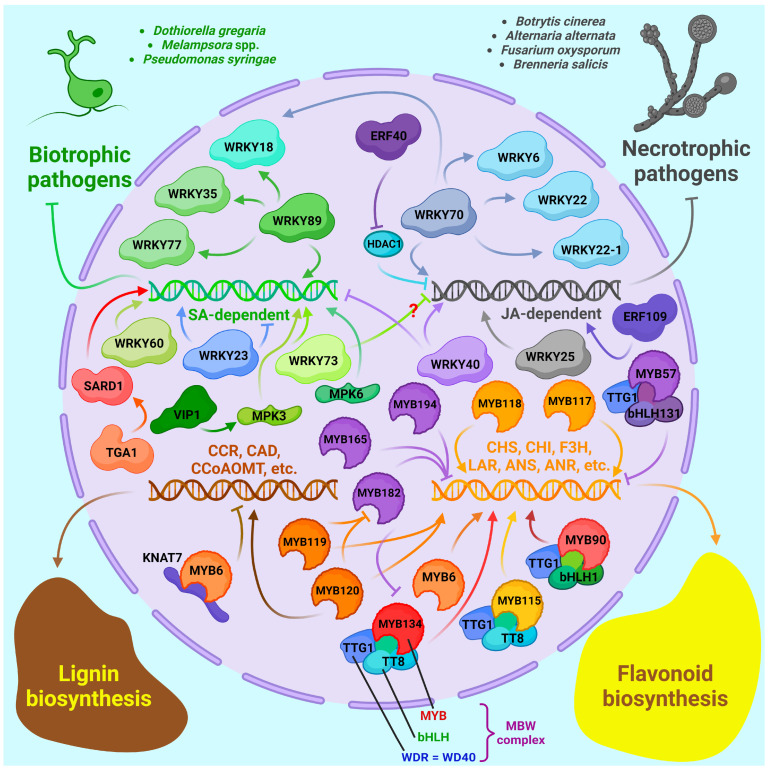
Transcription factors (TFs) involved in plant defense. The results of experiments on the effect of overexpression of TFs from the WRKY family (named so for the conserved WRKYGQK motif) on resistance to biotrophic and necrotrophic pathogens, as well as on changes in the composition of secondary metabolites as a result of overexpression of TFs from the myeloblastosis (MYB), basic Helix-Loop-Helix (bHLH, and this family also includes TRANSPARENT TESTA 8 (TT8) protein) and WDR~WD40 (which is called for WD repeat, a structural motif containing ~40 amino acids and terminating at Trp-Asp, or W-D; TRANSPARENT TESTA GLABRA 1 (TTG1) protein belongs to this group) families are illustrated. The role of TFs from the Ethylene Responsive Factor (ERF) and TGA (because they bind TGACG motif) families and the VIRE2-INTERACTING PROTEIN 1 (VIP1) protein in the regulation of defense responses is also briefly demonstrated. Plants control defence responses through a number of TF families. WRKY act on salicylic acid (SA)-dependent and jasmonic acid (JA)-dependent systems, conferring resistance to biotrophs and necrotrophs, respectively (which, however, may work differently in poplar than in *Arabidopsis* due to a possible lack of SA-JA antagonism). MYBs, bHLHs and WDR together form MYB-bHLH-WDR (MBW) complexes that activate or (less frequently) inhibit the biosynthesis of secondary metabolites, primarily flavonoids (not all MYBs are drawn as part of MBW complexes due to space limitations in the figure). ERFs mainly regulate JA/ET-dependent responses to necrotrophic pathogens, including inhibition of histone deacetylase 1 (HDAC1), which prevents resistance; VIP1 interacts with mitogen-activated protein kinase 3 (MPK3), a component of MAPK signaling; TGA1 activates Systemic Acquired Resistance Deficient 1 (SARD1), which is an important regulator of SA biosynthesis. Examples of some biotrophic and necrotrophic pathogens used in the articles and TF-activated secondary metabolite biosynthesis genes are given. ?—since this effect was obtained by transgenic expression of *PtrWRKY73* in *A. thaliana*, it may not be reproducible in poplar because of the possible lack of SA-JA antagonism [120]. Created with BioRender.com (accessed on 18 January 2024).

**Figure 7 ijms-25-01308-f007:**
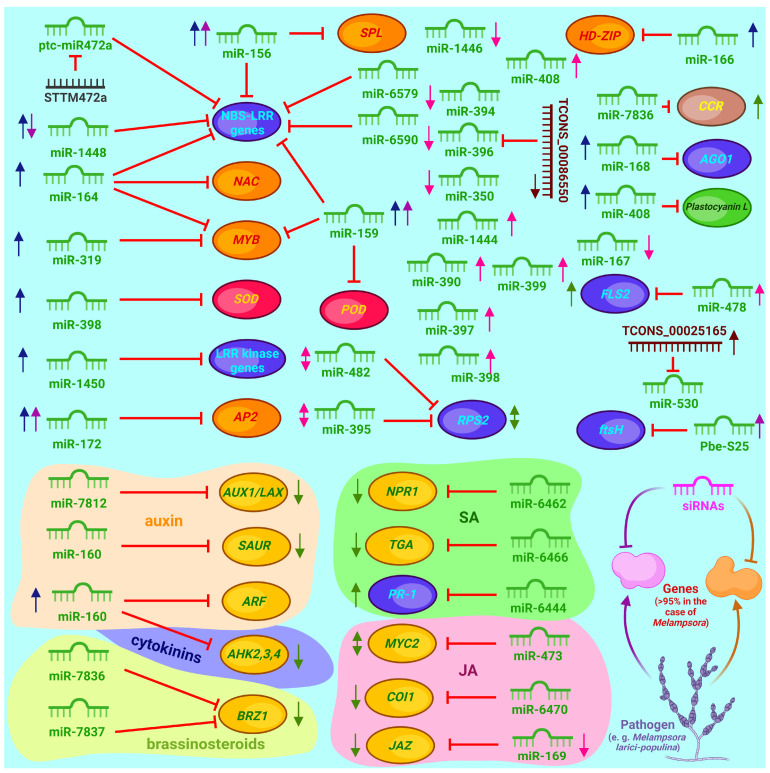
Differential expression of microRNAs and their targets under the influence of infection. Some interactions between microRNAs and their targets are shown. Defense proteins are blue, TFs are orange, hormone-signaling components are yellow and ROS-related proteins are pink. Changes in expression level detected in [236] are indicated by blue arrows, in [238] by purple arrows, in [246] by green arrows, in [92] by pink arrows, and in [249] by burgundy arrows. Upregulation is shown by an up-arrow and downregulation by a down-arrow; a bidirectional arrow indicates that the expression of different family members changes differently. This scheme is far from exhaustive because of the enormous complexity of microRNA regulation, the lack of data, and our low level of understanding of these processes. Non-coding RNAs can regulate defence responses in plants, including poplars. The large proportion of microRNAs target NBS-LRR genes, primarily TNLs, and under normal conditions, these microRNAs repress the expression of their targets, but during infection, this repression is attenuated, allowing for a strong and robust immune response to be triggered. miR-156 is a highly expressed microRNA, and its upregulation during infection allows it to suppress SPL TFs and, through it, auxin signaling, which competes with SA signaling and, thus, interferes with the immune process. miR-164—NAC TFs are an important regulatory axis of the immune response that is conserved in many plants. Many miRNAs control gene expression of hormonal pathways. Some lncRNAs are able to bind to miRNAs and repress their activity, representing another level of regulation. siRNAs are able to target and repress more than 95% of the *Melampsora*-derived genes. Ptc-miR472a is, so far, the only example of microRNA engineering in poplar trees in the context of defense against infection that has been both knocked in and knocked down (the latter by short tandem target mimic (STTM) technology). List of abbreviations used in this figure: NBS-LRR: nucleotide-binding site–leucine-rich repeat; NAC: NAM, ATAF1/2, and CUC2; MYB: myeloblastosis; SOD: superoxide dismutase; AP2: Apetala 2 AP2; SPL: Squamosa promoter-binding protein-like; POD: Peroxidase; RPS2: Resistance to Pseudomonas syringae protein 2; HD-ZIP: Homeodomain–leucine zipper; CCR: cinnamoyl-CoA reductase; AGO1: argonaute 1; FLS2: Flagellin-sensing 2-like protein; AUX1: AUXIN1; LAX: LIKE AUX1; SAUR: small auxin upregulated RNA; ARF: ADP ribosylation factor; AHK: Arabidopsis histidine kinase; BRZ1: BRASSINAZOLE-RESISTANT 1; NPR1: Nonexpressor of pathogenesis-related genes 1; TGA: TGACG motif-interacting; PR-1: Pathogenesis-related protein 1; MYC2: Myelocytomatosis; COI1: CORONATINE INSENSITIVE 1; JAZ: JASMONATE ZIM-DOMAIN. Created with BioRender.com (accessed on 18 January 2024).

**Figure 8 ijms-25-01308-f008:**
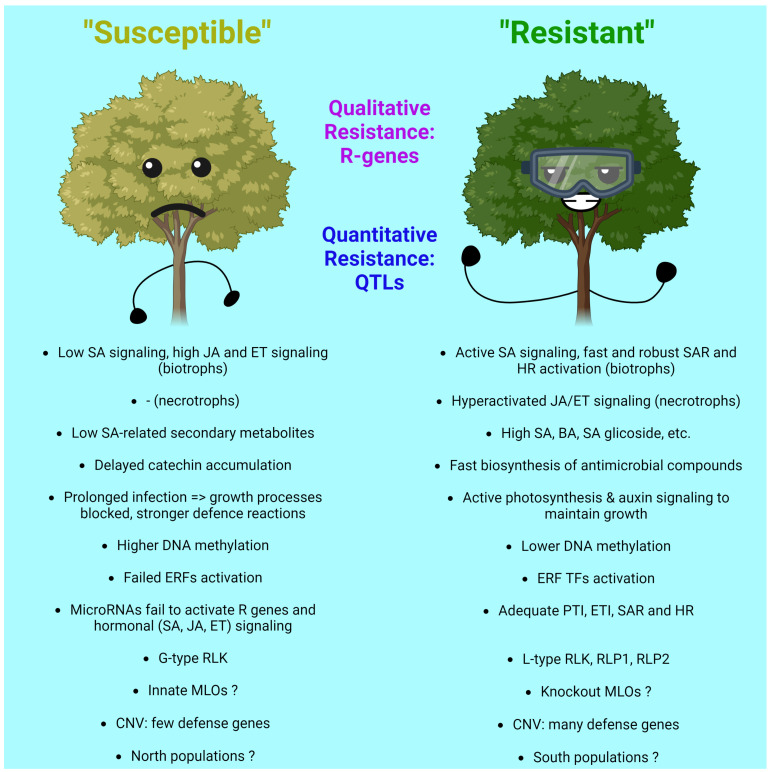
Metabolic, physiological, genetic and populational differences between “susceptible” and “resistant” poplars. “Generally resistant” and “generally susceptible” poplars differ from each other in a number of genes that are still used for breeding purposes. Resistance can be qualitative, which is absolute, highly pathogen-specific and provided by only a single gene, in this case, we speak of R-genes; and it can be quantitative, which is not absolute but has a broader spectrum of action and is provided by a group of genes, named quantitative trait loci (QTL). Poplar plants that are more resistant to biotrophs often have more active SA signaling and acquire SAR more rapidly, while those that are more resistant to necrotrophs have more active JA/ET signaling. More resistant plants may have higher concentrations of secondary metabolites and synthesize them more rapidly during infection. They have lower total DNA methylation levels and more adequate regulation by microRNAs. They continue to photosynthesize actively and maintain growth even during infection, maintaining an optimal trade-off between growth and defence. The genome of more resistant plants may contain more copies of defence genes, which is more characteristic of southern populations; however, these are data from only one study conducted on Canadian *P. balsamifera* populations, and so this trend may not be universal. It should be noted that the concepts of susceptibility and resistance are used here in a general sense, and in relation to different specific pathogens, these patterns may have a number of nuances. Created with BioRender.com (accessed on 18 January 2024).

**Figure 9 ijms-25-01308-f009:**
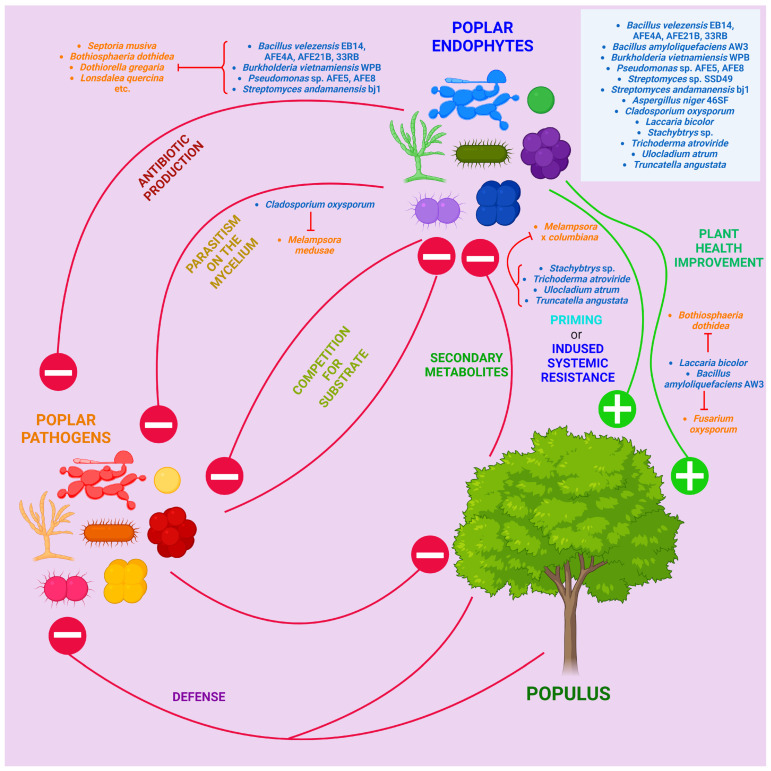
The role of endophytes in the defense of poplar against infection. Endophytes are beneficial for the plant in terms of defence against diseases. They suppress pathogens both directly (by producing antibiotics, parasitizing on pathogen mycelia and competing with them for substrate) and indirectly (by improving plant health and stimulating its immune system, termed priming in the case of more localized responses and induced systemic resistance in the case of more global responses). In turn, pathogens also affect the endophytic community, probably through similar direct mechanisms and by altering the secondary metabolite profile of the host plant. Examples of pathogen inhibition by endophytes which were considered in the text are also indicated in the figure. Created with BioRender.com (accessed on 18 January 2024).

**Figure 10 ijms-25-01308-f010:**
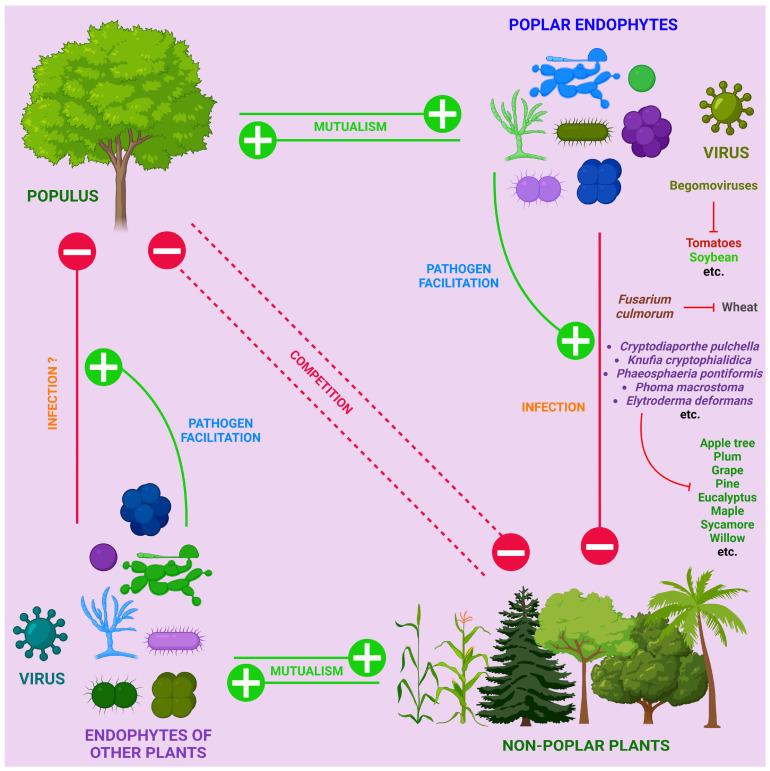
Indirect competition between plants through microorganisms. Endophytes can be used by plants as another mechanism of interspecific competition. The same species and strains of microorganisms may be beneficial to some plants, but, at the same time, may either cause diseases themselves or be pathogen facilitators for other plant species, which can be used by plants to suppress each other. Similarly, viruses that are asymptomatic in some plants can cause disease in others, such as begomoviruses in poplar. Created with BioRender.com (accessed on 18 January 2024).

**Figure 11 ijms-25-01308-f011:**
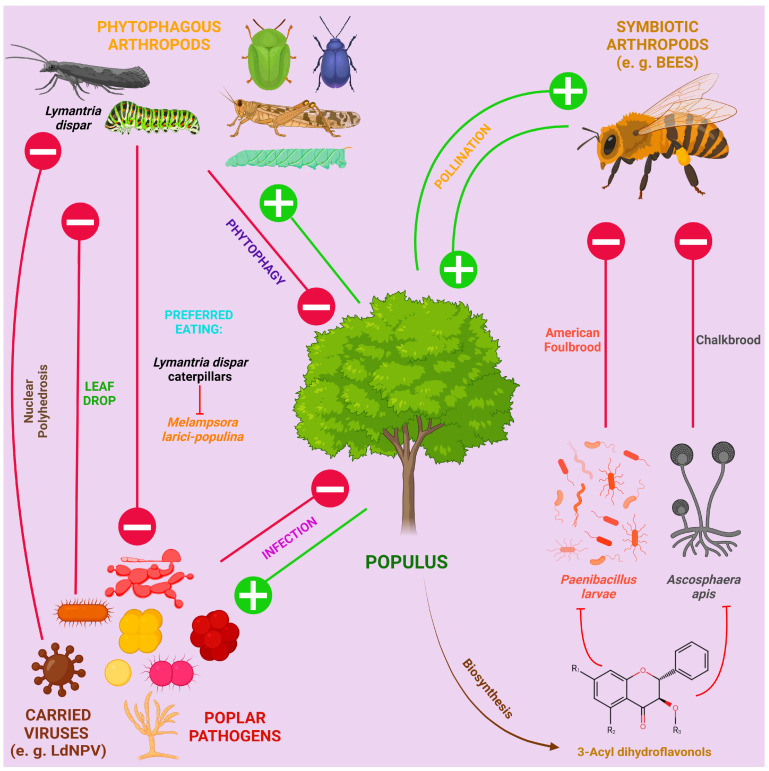
Ecological interactions between poplar, microorganisms and arthropods. Plants, associated microorganisms and arthropods constitute a complex and multifaceted regulatory network. Phytophagous insects can control the spread of phytopathogens by eating preferentially affected leaves because of their more suitable amino acid composition, vitamin content, etc. (*Lymantia dispar* caterpillars prefer *Melampsora larici-populina* infected leaves). In turn, pathogens may limit phytophagy by causing stress-induced leaf loss. At the same time, some viruses in the plant may be pathogenic to phytophages (such as LdNPV infects *Lymantia dispar*). The plant can protect associated arthropods (e.g., bees) from entomopathogenic microorganisms (e.g., *Paenibacillus larvae* and *Ascosphaera apis*) by synthesizing secondary metabolites (e.g., 3-acyl dehydroflavonols) that the insect obtains with food and, thus, becomes more protected. Created with BioRender.com (accessed on 18 January 2024).

**Figure 12 ijms-25-01308-f012:**
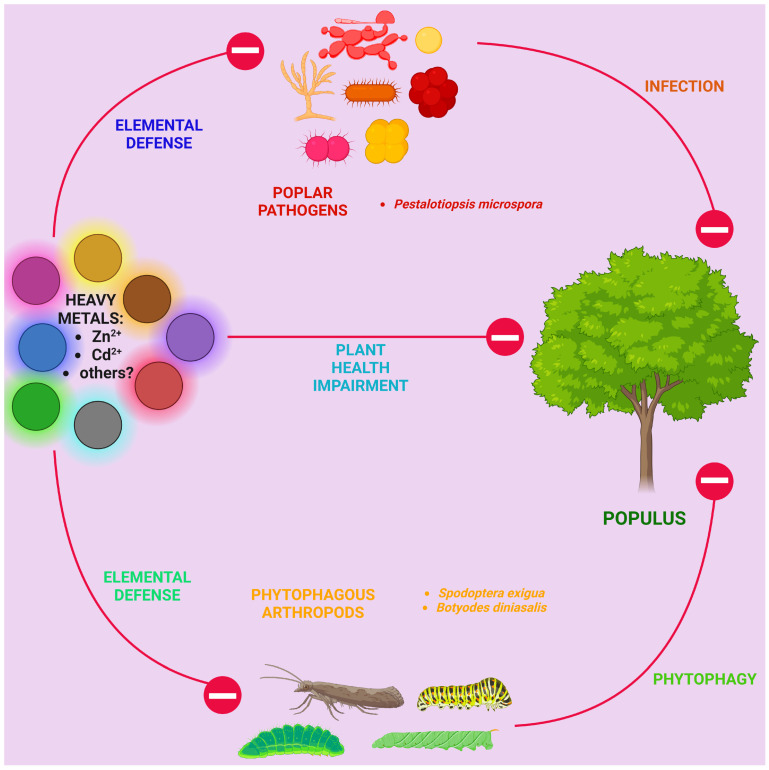
Elemental defence hypothesis. Atoms of heavy metals (zinc, cadmium, and, we suppose, many more), which are frequent soil pollutants and are largely absorbed by poplar because of their non-specific toxicity, protect the leaves of the plant (where metals accumulate in high concentrations) from being eaten by phytophages and infected by pathogens. This is called the elemental defence hypothesis. At the same time, heavy metals themselves sometimes make the plant more susceptible to disease by negatively affecting its own biochemical defenses and health. Created with BioRender.com (accessed on 18 January 2024).

**Table 1 ijms-25-01308-t001:** Examples of studies on genetic engineering of poplar and the effect of these modifications on resistance to certain pathogens. A—*Agrobacterium tumefaciens*—mediated genetic engineering and transformation; *—genes from other poplar species to be non-foreign to poplar; **—more effective than other modifications in the same work; ***—although not all transgenic plants have been tested for disease resistance, we predict its enhancement due to an increase in flavonoid content.

Article №	Poplar Species	Pathogen	Gene	Transgene	Method of Genetic Engineering	Change in Expression	Change in Resistance	Mechanism	Reference
**1**	*P. davidiana* *× P. bolleana*	*M.* *brunnea*	*AtRLP23*	YES	A	↑	↑	Recognition of additional PAMP, which cannot be detected by poplar receptors—NLP24	[53]
		*E*. *australis*					↑		
**2**	*P. tomentosa*	*D*. *gregaria*	*PtoACO7*	NO		↑	↑	Increased ET biosynthesis, levels and signaling	[117]
**3**	*P. trichocarpa*	*S. populiperda*	*PtDIR11*	NO		↑	↑	Increased content of lignans and flavonoids, activation of JA and ET pathways	[147]
**4**	*P. tomentosa*	*M. brunnea*	*PtrLAR3*	NO *	A	↑	↑	Increased proanthocyanidin content	[172]
**5**	*P. trichocarpa*	*S. populiperda*	*PtDef*	NO	A	↑	↑	Defensin toxicity to fungus, increased SA and JA activity, increased PRs and nsLTPs accumulation, increased HR amplitude and H_2_O_2_ accumulation	[42]
**6, 7**	*P. × euramericana*			NO *	A	↑	↑		[174,311]
**8**	*P. tomentosa*	*A. alternata*	*LJAMP2*	YES	A	↑	↑	LJAMP2 antifungal toxicity, maybe plant immunity activation as nsLTP	[176]
		*C. gloeosporioides*					↑		
**9**	*P. nigra × P. maximowiczii*	*S. musiva*	*MsrA2*	YES	A	↑	↑	MsrA2 (modified dermaseptin β1) is toxic to fungi	[179]
**10**	*P. tomentosa*	*C. chrysosperma*	*BbChit1*	YES	A	↑	↑	Chitinase activity => fungal cell wall degradation	[181]
**11**	*P. tomentosa*	*A. alternata*	*LJAMP2* or *BbChit1*	YES	A	↑	↑	Same to articles 10 or 11	[182]
			*LJAMP2 + BbChit1*		A	↑ + ↑	↑ **	Combined action of LJAMP2 and BbChit1	
**12**	*P. deltoides × P. euramericana*	*M. brunnea*	*PeTLP*	NO	A	↑	↑	Activation of other defense proteins	[183]
**13**	*P. davidiana × P. alba*	*C. chrysosperma*	*TaHsp24*	YES	A	↑	↑	Increased activity of SA and JA pathways and PRs content	[186]
		*A. alternata*					↑		
**14**	*P. deltoides × P. euroamericana*	*B. dothidea* or *A. alternata*	*PdePrx12*	NO	A	↑	↓	H2O2 scavenging => H2O2 content reduced in overexpressor and increased in underexpressor line	[187]
					A	↓	↑		
**15**	*P. tomentosa*	*D. gregaria*	*PtrWRKY89*	NO *	A	↑	↑	Activation of SA signaling and PR genes expression with no effect on JA signaling	[189]
**16**	*P. trichocarpa*			NO	A	↑	↑		[192]
		*M. brunnea*					↑		
**17, 18**	*P. simonii × P. nigra*	*A. alternata*	*PsnWRKY70*	NO	A	↑	↑	Activation of PTI, ETI and SA	[193]
**19**	*P. tomentosa*	*D. gregaria*	*PtrWRKY40*	NO *	A	↑	↓	Activation of JA, but inactivation of SA signaling => increased resistance to necrotrophs, but decreased to biotrophs	[194]
	*Arabidopsis thaliana*	*B*. *cinerea*		YES	A	↑	↑		
**20**	*P. tomentosa*	*D*. *gregaria*	*PtoWRKY60*	NO	A	↑	↑	Activation of SA signaling, no effect on JA	[195]
**21**	*Arabidopsis thaliana*	*P. syringae*	*PtrWRKY73*	YES	A	↑	↑	Activation of SA => increased resistance to biotrophs, but decreased to necrotrophs	[196]
		*B. cine* *rea*			A	↑	↓		
**22**	*P. tremula × P. alba*	*Melampsora* sp.	*PtWRKY23*	NO	A	↓ (RNAi)	↓	WRKY23 expression is at an optimal level in the WT plant and should be subject to adequate and flexible regulation?	[197]
					A	↑			
**23**	*P. simonii*	*A. alternata*	*PsnWRKY25*	NO	A	↑	↑	Activation of PsCERK1, PR1 and secondary metabolism	[198]
**24**	*P. tremula × P. tremuloides*	-	*MYB182*	NO	A	↑	-	Repression of flavonoid biosynthesis pathway => decreased flavonoid content in overexpressors and increased in knockout plants ***	[202]
**25**	*P. tremula × P. tremuloides*	-	*MYB165*	NO	A	↑	-		[203]
			*MYB194*	NO	A	↑	-		
**26**	*P. tomentosa*	-	*PtrMYB57*	NO *	A	↑	-		[204]
				NO	CRISPR/Cas9	↓ (knockout)			
**27**	*P. tremula × P. alba*	*M. larici-populina, M. aecidiodes*	*MYB134*	NO	A	↑	↑	Upregulation of flavonoid biosynthesis pathway => increased flavonoid content ***	[170]
**28**	*P. tremula × P. alba*	*-*		NO	A	↑	-		[205]
	*P. tremula × P. tremuloides*				A				
**29**	*P. tomentosa*	*D. gregaria*	*PtoMYB115*	NO	A	↑	↑		[55]
	*Nicotiana benthamiana*	*-*	*PtoMYB115 + PtoTT8 + PtoTGA1*	YES	A	↑ + ↑ + ↑	-		
**30**	*P. alba*	*B. cinerea*	*PalMYB90 + PalbHLH1*	NO	A	↑ + ↑	↑		[206]
							↑		
**31**	*P. alba × P. tremula*	-	*PtrMYB119*	NO *	A	↑	-		[207]
			*PtrMYB120*	NO *	A	↑			
**32**	*P. alba × P. glandulosa*	-		NO *	A	↑	-		[208]
					A	↓			
**33**	*P. tomentosa*	*-*	*PtoMYB6*	NO	A	↑	-		[209]
**34**	*P. deltoides.*	-	*PdeMYB118*	NO	A	↑	-		[210]
**35**	*P. tremula × P. tremuloides*	-	*MYB117*	NO	A	↑	-		[211]
**36**	*P. tomentosa*	-	*PtoMYB142*	NO	A	↑	-	Increased wax content ***	[212]
**37**	*P. davidiana × P. alba*	*F. oxysporum*	*PdPapERF109*	NO	A	↑	↑	Increase in ROS-scavenging activity => more adequate immune response	[219]
**38**	*P. trichocarpa*	*B. salicis*	*AtVIP1*	YES	A	↑	↑	PR1 activation; full molecular mechanism is unknown	[85]
			*PtVIP1*	NO	A	↑			
**39**	*P. tomentosa*	*C*. *gloeosporioides*	*PeTGA1*	NO *		↑	↑	PeSARD activation => upregulation of SA biosynthesis	[223]
**40**	*P. trichocarpa*	*C. gloeosporioides*	*ptc-miR472*	NO	A	↑	↓	Fail to activate ETI, NBS-LRRs, ROS => susceptible to biotrophs; active JA/ET signaling => resistant to necrotrophs	[241]
		*C. chrysosperma*					↑		
		*C. gloeosporioides*			A	↓ (via STTM)	↑	Hyperactivated NBS-LRRs => quick and robust ETI and SA response	
		*C. chrysosperma*					~unchanged		
**41**	*P. alba × P. grandidentata*	*M. aecidiodes*	*AtGolS + CsRFS*	YES	A	↑	↓	Suppressed SA, Ca2+, phosphatidic acid, activated JA signaling => failed ROS and PR1 accumulation => increased susceptibility to biotrophs (but resistance to necrotrophs?)	[262]
**42**	*P. davidiana × P. bollena*	*A*. *alternata*	*PdbLOX2*	NO	A	↑	↑	Hyperaccumulation of JA => hyper-resistance to necrotrophs	[263]
**43**	*P. tremula × P. alba, P. deltoides*	*-*	*LecRLK-G*, *TPX2*	NO	CRISPRa (dCas9)	↑	-	More active immune response ***	[312]
		*-*	*PLATZ*	NO	CBE (nCas9)	↓	-		

## Data Availability

The datasets generated during and/or analyzed during the current study are available from the corresponding author on reasonable request.

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
