# Peer review of "Editing Metabolism, Sex, and Microbiome: How Can We Help Poplar Resist Pathogens?"

_ijms, 2024, doi:10.3390/ijms25021308_

Round 1

Reviewer 1 Report

Comments and Suggestions for Authors

Below are my comments for the manuscript: 

- Revise the abstract (Lines 46-56) to distinctly emphasize the novel findings and contributions of this review, avoiding general statements about poplar and its importance. The abstract should explicitly highlight the key findings of this review, rather than providing a general overview of the topic.

- Expand the introduction (Lines 99-115) to include a nuanced historical context and the evolution of challenges in poplar cultivation, especially in relation to pathogen resistance. This would provide a richer context for the review.

- In the section on poplar immunity (Section 3.1, Lines 468-470), a more comprehensive analysis of poplar-specific immune responses and evolutionary adaptations is needed. The section requires a more in-depth exploration, especially focusing on distinctive immune mechanisms and evolutionary adaptations unique to poplars. Comparative analyses with other plants would add value.

- The discussion on metabolic reactions in poplars to pathogenic invasion (Section 3.2, Lines 853-859) should include more concrete examples and case studies to illustrate how these metabolic changes contribute to disease resistance. Integrate more specific and detailed examples to demonstrate how metabolic responses in poplars contribute directly to disease resistance.

- The lignin biosynthesis section (Lines 1032-1038) requires a more detailed exploration of how this process specifically enhances poplar's resistance to pathogens. Discuss the role of individual enzymes and their genetic regulation, and illustrate its specific role in enhancing resistance to pathogens.

- Expand on the role of salicylic acid in poplar immunity (Lines 1143-1148). This section needs to delve deeper into its mechanism in poplars and explore potential genetic engineering applications for improved disease resistance. The role of salicylic acid in plant immunity needs further elaboration, particularly its specific actions in poplar disease resistance.

- Include specific examples or recent research on the application of CRISPR/Cas techniques in poplars. The manuscript should demonstrate how these technologies are being used to enhance disease resistance.

- Add a section on methodological approaches used in cited studies, critically evaluating their designs, limitations, and implications for the conclusions drawn.

- Articulate clear future research directions in the conclusion, identifying gaps in current knowledge and proposing specific investigative areas.

- Update the literature review to incorporate recent, relevant studies that support the findings and discussions in the review.

- Enhance the integration of figures and tables to better illustrate key concepts, pathways, or genetic strategies, thereby aiding comprehension.

- Address any editorial issues such as grammatical errors and inconsistencies in formatting throughout the manuscript.

- Include a more critical analysis of the cited studies, highlighting potential biases or conflicting findings in the field of poplar disease resistance.

- Improve cross-referencing within the manuscript for better navigation and coherence between related sections.

- Elaborate on microbiome strategies in enhancing poplar resistance to pathogens, including recent advancements or experimental findings.

- Explore and integrate the role of sex-related aspects in disease resistance, as suggested by the title, to ensure a comprehensive coverage of the topic.

- Provide an in-depth focus on specific pathogens that pose the greatest threat to poplars, rather than a broad overview of various pathogens.

- Discuss the economic impact of improved pathogen resistance in poplars, particularly in relation to the pulp industry and biofuel production.

- Address the environmental implications of genetic engineering in poplars, including potential risks and mitigation strategies.

- Bridge the gap between theoretical research and practical applications, discussing how the findings can be translated into real-world improvements in poplar cultivation.

Author Response

Dear reviewer!

We are very grateful to you for your extremely detailed review of our manuscript. Indeed, it turned out to be enormous and it was very difficult to organise it competently and present everything clearly, so many points required substantial revision. Below are our responses to each of the points in the review.

  1. Revise the abstract (Lines 46-56) to distinctly emphasize the novel findings and contributions of this review, avoiding general statements about poplar and its importance. The abstract should explicitly highlight the key findings of this review, rather than providing a general overview of the topic.

Response. We thank the reviewer for this valuable comment, we have changed the abstract of the article so that it includes more details related to our review and the conceptual ideas that we propose in conclusion.

  1. Expand the introduction (Lines 99-115) to include a nuanced historical context and the evolution of challenges in poplar cultivation, especially in relation to pathogen resistance. This would provide a richer context for the review.

Response. We thank the reviewer for the wonderful idea that we implemented in the article. We have added a paragraph with brief information about the history of poplar use and their selection.

However, we believe that this topic is quite complex and voluminous, in addition, it requires a separate detailed meta-analysis. We agree that poplar cultivation affects resistance to pathogens, but it is important to take into account that pathogens overcome plant resistance and acquire new virulence factors, so that the process of obtaining resistance and overcoming it is constant throughout the evolutionary history of plants. Therefore, we cannot unequivocally state that this issue requires detailed consideration in the review, since one way or another, the ways in which we can help plants maintain productivity are methodologically similar today, regardless of the historical context of breeding processes with poplar culture.

However, as an important element of the introduction, we consider your comment more than appropriate and have included the relevant information in the manuscript.

  1. In the section on poplar immunity (Section 3.1, Lines 468-470), a more comprehensive analysis of poplar-specific immune responses and evolutionary adaptations is needed. The section requires a more in-depth exploration, especially focusing on distinctive immune mechanisms and evolutionary adaptations unique to poplars. Comparative analyses with other plants would add value.

Response. We thank the reviewer for this comment, we provided additional information about the specific immune responses of poplar to pathogen invasion. We would like to note that in the review we consider sources in which the molecular mechanisms of the reaction of poplars of different species to infection were directly studied, to a lesser extent we provide general known data on all plant groups, which makes our review especially valuable for researchers working with representatives of the genus Populus.

  1. The discussion on metabolic reactions in poplars to pathogenic invasion (Section 3.2, Lines 853-859) should include more concrete examples and case studies to illustrate how these metabolic changes contribute to disease resistance. Integrate more specific and detailed examples to demonstrate how metabolic responses in poplars contribute directly to disease resistance.

Response. We are very grateful to the reviewer for an important addition, which will undoubtedly make the work more detailed. We have introduced new data on the mechanisms of action of different groups of secondary plant metabolites on pathogenic fungi. However, let us clarify that in this section, in addition to considering the groups of metabolites synthesized by poplar themselves, we pay attention to more general issues in order to create a holistic picture of the reader's perception. Since it is impossible to perceive particular issues of metabolism without considering the fundamental accepted concepts, we considered it important to pay enough attention to them.

  1. The lignin biosynthesis section (Lines 1032-1038) requires a more detailed exploration of how this process specifically enhances poplar's resistance to pathogens. Discuss the role of individual enzymes and their genetic regulation, and illustrate its specific role in enhancing resistance to pathogens.

Response. We thank the reviewer for this comments, we have provided additional information about lignin and the metabolic pathways associated with the participation of the compound and its precursors. We also clarified about the immune function of lignin itself in section 6.

  1. Expand on the role of salicylic acid in poplar immunity (Lines 1143-1148). This section needs to delve deeper into its mechanism in poplars and explore potential genetic engineering applications for improved disease resistance. The role of salicylic acid in plant immunity needs further elaboration, particularly its specific actions in poplar disease resistance.

Response. We have added case studies on the role of salicylic acid in poplar immune responses, as well as on its interaction with other biochemical pathways and elements of poplar protection.

  1. Include specific examples or recent research on the application of CRISPR/Cas techniques in poplars. The manuscript should demonstrate how these technologies are being used to enhance disease resistance.

Response. Section 6 of the manuscript is devoted to various methods of genetic engineering of poplars, the use of which can be aimed at increasing the resistance of poplars to pathogens of phytopathologies. In particular, we pay great attention to the studies that were performed using the CRISPR/Cas9 genomic editing system. In addition, other sections also mention the work, the results of which were obtained through the use of this method of genome editing. We pay great attention to its application possibilities, advantages and limitations.

  1. Add a section on methodological approaches used in cited studies, critically evaluating their designs, limitations, and implications for the conclusions drawn.

Response. We thank the reviewer for this comment, we paid more attention to the methodological features of the mentioned manuscripts and critically reviewed their results for a more objective assessment, including analyzing the results of works with contradictory results and making our own assumptions based on the available data.

  1. Articulate clear future research directions in the conclusion, identifying gaps in current knowledge and proposing specific investigative areas.

Response. We have changed the conclusion in accordance with the remark, supplemented with research directions that can be performed to expand knowledge about the immune system of poplar and for the possibility of application in practice.

  1. Update the literature review to incorporate recent, relevant studies that support the findings and discussions in the review.

Response. In the review, we discuss more than 350 articles devoted to case studies of poplar immunity. We try to cover both the fundamental generally accepted aspects of plant protection, as well as the latest data that complement the knowledge about the immunity of poplars. In accordance with all the comments of the reviewer, we have reviewed additional sources to make the review more objective and detailed.

We want to clarify that the field under consideration is quite highly specialized. Studies of the immune system of poplars have been conducted for a long period, and the interest of researchers in this issue does not fade, and therefore we present the results of work from different years, focusing on the latest data obtained over the past five years. We hope that all the data we provide will be valuable to researchers and can stimulate research to improve the stability of poplars and increase their productivity.

  1. Enhance the integration of figures and tables to better illustrate key concepts, pathways, or genetic strategies, thereby aiding comprehension.

Response. We thank the reviewer for this valuable comments on the figures and tables. We have tried to make them more understandable and accessible to the reader. We have improved their integration into the text, supplemented the captions for a better perception.

  1. Address any editorial issues such as grammatical errors and inconsistencies in formatting throughout the manuscript.

Response. Thanks for the comment, we have corrected stylistic, grammatical flaws and improved formatting.

  1. Include a more critical analysis of the cited studies, highlighting potential biases or conflicting findings in the field of poplar disease resistance.

Response. We thank the reviewer for the recommendation, we have reviewed a number of data and made appropriate adjustments. In addition, where necessary, we have provided contradictory results and tried to find the reason for their occurrence.

  1. Improve cross-referencing within the manuscript for better navigation and coherence between related sections.

Response. In accordance with the reviewer's comment, we have cross-referenced the text to ensure a more consistent connection between sections and improve reader convenience.

  1. Elaborate on microbiome strategies in enhancing poplar resistance to pathogens, including recent advancements or experimental findings.

Response. We have supplemented section 6, devoted to the genetic engineering of poplars, with data on microbiome engineering performed on poplars. In addition, the paragraphs on environmental risks also cover some studies on this issue.

  1. Explore and integrate the role of sex-related aspects in disease resistance, as suggested by the title, to ensure a comprehensive coverage of the topic.

Response. Section 5 of our manuscript is devoted to the issue of differences in the physiological reactions of poplars to invasion by pathogens depending on the sex of the tree, including those related to the peculiarities of the formation of consortia of microorganisms and intrapopulation relationships of trees. We have supplemented this section with our conclusions based on the reviewed thematic papers.

  1. Provide an in-depth focus on specific pathogens that pose the greatest threat to poplars, rather than a broad overview of various pathogens.

Response. We thank the reviewer for this remark, we have separately listed the most dangerous pathogens of poplar diseases, which cause the most pronounced economic damage when cultivating poplars.

  1. Discuss the economic impact of improved pathogen resistance in poplars, particularly in relation to the pulp industry and biofuel production.

Response. We have made appropriate adjustments to the manuscripts, but at the moment it is difficult to assess the economic consequences of growing such poplars, since the use of genetically modified plants in most countries is currently legally limited.

  1. Address the environmental implications of genetic engineering in poplars, including potential risks and mitigation strategies.

Response. We thank the reviewer for this comments. In section 6 of our review, we consider the potential environmental risks associated with the use of genetically modified poplars and present thematic articles in which we checked the danger of such plants to the environment. In accordance with the reviewer's comment, we have presented strategies that seem rational to us to reduce the risks discussed.

  1. Bridge the gap between theoretical research and practical applications, discussing how the findings can be translated into real-world improvements in poplar cultivation.

Response.We took into account this comment of the reviewer and supplemented section 6 of the article, which is entirely devoted to the genetic engineering of poplars, and added, in accordance with an earlier comment, specific potential research directions that may lead to a more pronounced resistance of poplars to a number of pathogens.

However, it should be noted that in most countries the cultivation of genetically modified poplars is legally limited, and therefore it is difficult to bridge the gap between theoretical research and practical applications, since it is difficult to assess the real consequences of planting such poplars and it is impossible to predict economic prospects. In the manuscript, in conclusion, we separately note that the functions of a number of elements involved in the processes of the immune response of poplars have not been sufficiently studied, and it remains unclear how the physiology of the plant will be affected by a change in the expression of a number of genes. A striking example illustrating this from our work is the change in the accumulation of salicylic acid, the increased accumulation of which led to morphological changes.

Reviewer 2 Report

Comments and Suggestions for Authors

The manuscript reviewed the findings of poplar genes and metabolites that were related to resistance to the pathogens. The relationships between the sex effects of the poplar and the resistant were also discussed. Genetical engineering, CRISPR/Cas genome editing, and microbiome engineering were proposed to create disease-resistant poplars with economic values. The revised manuscript massively modified the text and the figures but may not fully describe the mechanisms in concert with the comprehensive title. In addition, some of the figures were too simplified to benefit minorly the researchers in this field. It is suggested that the manuscript be modified accordingly, as shown below, before the recommendation.

1.     The method for the construction of the phylogenetic tree for fungal pathogens in Figure 1 should be described. The relative ages should be indicated.

2.     The method for the construction of the phylogenetic tree for non-fungal pathogens in Figure 2 should be described. The relative ages should be indicated. In addition, specific species were discussed and should be used for the construction of the phylogenetic tree, and the species names were shown on it. An additional outgroup should be used to show the relationship with oomycete Phytophthora cactorum.

3.     The full names of the abbreviations used in Figure 3 should be described in the legend, whether shown in the text or not. It would be beneficial for the understanding of the gene/protein functions. In addition, the gene/protein function could be further discussed. For example, the EDS1/PAD4, EDS1/SAG101, CNLs, NDR1, NPR1, and others.

4.     The full names of the abbreviations used in Figure 5 should be described in the legend, regardless of whether they were shown in the text or not. It would be beneficial for the understanding of the gene/protein functions. For example, the nsLTPs, TLPs, RISPs, USPs, and others. In addition, the bacterial and fungal pathogens could be specified in accordance with the discussion in the text.

5.     The full names of the abbreviations used in Figure 6 should be described in the legend, regardless of whether they were shown in the text or not. It would be beneficial for the understanding of the gene/protein functions. For example, the ERF40, HDAC1, and others. In addition, the necrotrophic and biotrophic pathogens could be specified in accordance with the discussion in the text.

6.     The full names of the abbreviations used in Figure 7 should be described in the legend, regardless of whether they were shown in the text or not. It would be beneficial for the understanding of the gene/protein functions. For example, the SAUR, ARF, BRZ1, and others. In addition, the siRNAs-mediated inhibition of virulence proteins could be specified.

7.     The genomes of southern populations showed more resistant genes than those of northern populations, which were interesting findings. The definition should be carefully explained according to the results of different research articles. The indications of north and south populations in Figure 8 should also be described in the legend.

8.     Figure 9 was too simplified. The poplar endophytes and pathogens could be specified in the figure or the figure legend in accordance with the discussion in the text.

9.     As described above, the Figure 10 was too simplified. The poplar endophytes, the endophytes from other plants, and the non-poplar plants could be specified in the figure or the figure legend in accordance with the discussion in the text.

10.  Figure 11 was also too simplified. The poplar pathogens and carried viruses, the insect pathogens, phytophagous arthropods, and symbiotic arthropods could be specified in the figure or the figure legend in accordance with the discussion in the text.

11.  As described above, the Figure 12 was also too simplified. The poplar pathogens and the phytophagous arthropods could be specified in the figure or the figure legend in accordance with the discussion in the text. In addition, Cu2+, Co2+, and Ni2+ in the figure were not discussed in the text. They should be further described.

12.  Some figures were not indicated in the text. They should be mentioned before the presentation of the figures. For example, Figures 4, 5, 6, 7, 8, 9, 10, 11, and 12.

13.  The position of Section 6. Development of sustainable genetically modified poplars - prospects and challenges should be corrected.

14.  Table 1. A - Agrobacterium tumefaciens should be in italics. * - we consider genes from other poplar species to be non-foreign to poplar and may not be suitable. It could be modified as * - genes from other poplar species to be non-foreign to poplar.

15.  It is suggested that an additional section for microbiome engineering be added to reveal its importance in front of Section 7 Conclusions.

Author Response

Dear reviewer,

We thank you for your detailed analyses of our work. This helped us improve our manuscript a lot, especially in terms of the figures and their interaction with the text.

  1. The method for the construction of the phylogenetic tree for fungal pathogens in Figure 1 should be described. The relative ages should be indicated.

Response. We apologise for the error. The image in Figure 1 is not strictly a phylogenetic tree. We did not construct it using bioinformatic methods such as UPGMA, ML, etc., but simply drew it manually in BioRender software using information from NCBI Taxonomy and the articles listed in the figure captions. Thus, it is a cladogram, and the divergence times of the groups, although we can find them in the articles, are hardly important in the context of our paper. We decided that because of the large number of pathogens under consideration, presenting them in the form of a cladogram would make the material discussed more understandable to the reader. We have corrected inaccuracies in the figure caption.

  1. The method for the construction of the phylogenetic tree for non-fungal pathogens in Figure 2 should be described. The relative ages should be indicated. In addition, specific species were discussed and should be used for the construction of the phylogenetic tree, and the species names were shown on it. An additional outgroup should be used to show the relationship with oomycete Phytophthora cactorum.

Response. The method of creating Figure 2 is similar to that of Figure 1, so there are no relative ages and we have corrected inaccuracies in the captions. For the same reason, we think it is possible not to add an outgroup to Phytophthora castorum, besides it may confuse the reader, because all organisms mentioned in Figures 1 and 2 are pathogens of poplar. Species names have been added to Figure 2.  

  1. The full names of the abbreviations used in Figure 3 should be described in the legend, whether shown in the text or not. It would be beneficial for the understanding of the gene/protein functions. In addition, the gene/protein function could be further discussed. For example, the EDS1/PAD4, EDS1/SAG101, CNLs, NDR1, NPR1, and others.

Response. All abbreviations have now been deciphered. The role of all the proteins in the picture is also now discussed in the caption of Figure 3.

  1. The full names of the abbreviations used in Figure 5 should be described in the legend, regardless of whether they were shown in the text or not. It would be beneficial for the understanding of the gene/protein functions. For example, the nsLTPs, TLPs, RISPs, USPs, and others. In addition, the bacterial and fungal pathogens could be specified in accordance with the discussion in the text.

Response. All protein names are now deciphered in the figure caption. However, we suppose that specific bacterial and fungal pathogens should not be indicated in the figure because defensins, chitinases, TLPs, and other proteins discussed in this section are active against a very wide range of pathogens (e. g. chitinases – against most fungi), and specific representatives have been indicated in the text only because experimental studies have been conducted on them, but we expect to see the same effect on a large number of other phytopathogens.

  1. The full names of the abbreviations used in Figure 6 should be described in the legend, regardless of whether they were shown in the text or not. It would be beneficial for the understanding of the gene/protein functions. For example, the ERF40, HDAC1, and others. In addition, the necrotrophic and biotrophic pathogens could be specified in accordance with the discussion in the text.

Response. We have deciphered the names of all genes and proteins. Examples of biotrophic and necrotrophic pathogens are also given.

  1. The full names of the abbreviations used in Figure 7 should be described in the legend, regardless of whether they were shown in the text or not. It would be beneficial for the understanding of the gene/protein functions. For example, the SAUR, ARF, BRZ1, and others. In addition, the siRNAs-mediated inhibition of virulence proteins could be specified.

Response. We have decoded all the abbreviations indicated in the figure, separating them into a distinct list due to the large number of them. As for the virulence genes, we have replaced them with 95% of the pathogen genes because 16372 genes of Melampsora larici-populina, including the ribosomal genes, the G-protein genes, etc. are susceptible to suppression by small poplar RNAs, and the virulence genes constitute only one particular group (besides, the authors themselves did not highlight them in the preprint, so this can only be done by very careful analysis of supplementary files). This is now reflected both in the text of the paper and in the figure caption.

  1. The genomes of southern populations showed more resistant genes than those of northern populations, which were interesting findings. The definition should be carefully explained according to the results of different research articles. The indications of north and south populations in Figure 8 should also be described in the legend.

Response. Unfortunately, we cannot speak about it with certainty yet. This is evidenced by the data of only one study that used a genomic approach on canadian populations of P. balsamifera. Moreover, the plants were not tested experimentally for resistance, and scientists looked only at the number of genes, or copy number variations (CNVs). Nevertheless, the results seem quite logical to us, as abiotic stresses (cold, water and light deficiency, etc.) are more pronounced in the north and biotic stresses (pathogens and phytophages) are more pronounced in the south. We have added question marks to the figure to show that it is premature to draw far-reaching conclusions (along the lines of how we assumed that knockout MLOs would increase sustainability) and have reflected this in the figure caption.

  1. Figure 9 was too simplified. The poplar endophytes and pathogens could be specified in the figure or the figure legend in accordance with the discussion in the text.

Response. We have added a list of pathogens to the figure, as well as examples of endophyte-pathogen inhibition pairs that we discussed in the text.

  1. As described above, the Figure 10 was too simplified. The poplar endophytes, the endophytes from other plants, and the non-poplar plants could be specified in the figure or the figure legend in accordance with the discussion in the text.

Response. We have added examples of poplar endophytes pathogenic to other plants, as well as begomoviruses discussed in the article. The reverse situation has not yet been studied, but we assume that some endophytes of other plants may be pathogenic for poplar as well. In general, this type of ecological interactions is still poorly understood.

  1. Figure 11 was also too simplified. The poplar pathogens and carried viruses, the insect pathogens, phytophagous arthropods, and symbiotic arthropods could be specified in the figure or the figure legend in accordance with the discussion in the text.

Response. We have concretised our scheme. The mechanism of bee defence by poplar secondary metabolites is shown in more detail. Information about LdNPV and that it infects Lymantia dispar is added. Added that Lymantia dispar caterpillars preferentially eat leaves infested with Melampsora larici-populina.

  1. As described above, the Figure 12 was also too simplified. The poplar pathogens and the phytophagous arthropods could be specified in the figure or the figure legend in accordance with the discussion in the text. In addition, Cu2+, Co2+, and Ni2+in the figure were not discussed in the text. They should be further described.

Response. We specialised on those phytophages and arthropods that were mentioned in the articles devoted to the elemental protection hypothesis. Unfortunately, in the articles on poplar, the studies were conducted only with zinc and cadmium ions. Nevertheless, we expect to see a similar effect with other metals: copper, nickel, cobalt, etc. We have changed the caption on the figure to be more cautious one. Moreover, we believe that these effects are very broad and work with many plants, phytophages and pathogens.

  1. Some figures were not indicated in the text. They should be mentioned before the presentation of the figures. For example, Figures 4, 5, 6, 7, 8, 9, 10, 11, and 12.

Response. Now all the figures are referred to in the main text.

  1. The position of Section 6. Development of sustainable genetically modified poplars - prospects and challenges should be corrected.

Response. We thank the distinguished reviewer for this comment, we have supplemented section 6 in accordance with your comment.

  1. Table 1. A - Agrobacterium tumefaciensshould be in italics. * - we consider genes from other poplar species to be non-foreign to poplar and may not be suitable. It could be modified as * - genes from other poplar species to be non-foreign to poplar.

Response. Done.

  1. It is suggested that an additional section for microbiome engineering be added to reveal its importance in front of Section 7 Conclusions.

Response. In accordance with your comment, we have divided Section 6 into subsections that include a discussion of the possibility of editing the poplar genome and its microbiome, in addition, we have separately highlighted the part devoted to the challenges and prospects of these approaches.

Reviewer 3 Report

Comments and Suggestions for Authors

A very well written article summarizing the knowledge of pathogens attacking species of the genus Populus but also the ability of these plants to defend themselves. The authors described the different levels of this defense both at the molecular level and at the chemical or biological level.

I have only minor comments. Shouldn't the section names be written in italics.

How does mycorrhiza affect poplar's resistance to drought, salinity and heavy metals, as well as its susceptibility to pathogens? Is there any research available on this? Won't inoculating with the right mycorrhizal fungus increase poplar resistance?

Author Response

Dear reviewer,

Thank you for your positive feedback on our work. Your comments allowed us to improve the text and add some details. Here is our response to each point.

  1. Should we write section names in italics?

Response. Yes, you're absolutely right. We got a bit confused and wrote them not in italics, which is wrong. In the article at reference 30, all section names are italicised except ATL - it is in normal font, as we left it.

  1. Add information about mycorrhiza.

Response. Yes, mycorrhiza does help to fight biotic and abiotic stress. Regarding biotic stress, we already have information that Laccaria bicolor, which is a model ectomycorrhizal fungus, contributes to the defence of poplar against pathogens (Ref. 278). We have also added a paragraph on how mycorrhizal fungi and bacterial endophytes can contribute to poplar resistance to various types of abiotic stress such as osmotic stress, heavy metal exposure and organic pollutants.

Round 2

Reviewer 1 Report

Comments and Suggestions for Authors

Maniscript can be accepted.

Author Response

Dear reviewer!

Thank you for all your comments. Below I attach the final version of our paper.

Reviewer 2 Report

Comments and Suggestions for Authors

The manuscript broadly reviewed the findings of poplar genes and metabolites that were related to resistance to the pathogens. The relationships between the sex effects of the poplar and the resistant were also discussed. Genetical engineering, CRISPR/Cas genome editing, and microbiome engineering were proposed to create disease-resistant poplars with economic values. The revised manuscript modified the text and the figures accordingly, which is beneficial to the scientists in this field. It is suggested that essential modifications be made before recommending it to the International Journal of Molecular Sciences, as shown below.

1.     The idea of cladogram was acceptable to avoid incorrect construction of phylogenetic relationships. As described, Figures 1 and 2 were made by using the Biorender software. The fact should be mentioned in the legend because it claims certain rights.

2.     The bacterial and fungal pathogens in Figure 5 were not specified in accordance with the discussion in the text. As claimed by the authors, only specific representatives indicated in the text that conducted experimental studies need to be specified to interest the readers.

3.     The illustration of genes in Figure 7 was unsuitable. The microRNAs regulate gene expression, but the proteins were drawn in the figure. The icons that represented the protein should be replaced by showing the gene names. In addition, the gene names should be in italics.

4.     As above, it should be cautious to show the gene names in italics in the text.

Author Response

Dear reviewer,

We have have taken into account all your comments and have now corrected the article. I'm attaching the revised version.

  1. The idea of cladogram was acceptable to avoid incorrect construction of phylogenetic relationships. As described, Figures 1 and 2 were made by using the Biorender software. The fact should be mentioned in the legend because it claims certain rights.

Response. Yes, indeed. We've now added the caption "Created with BioRender.com." to all the images because they were made with BioRender.

  1. The bacterial and fungal pathogens in Figure 5 were not specified in accordance with the discussion in the text. As claimed by the authors, only specific representatives indicated in the text that conducted experimental studies need to be specified to interest the readers.

Response. We added examples of pathogens discussed in the articles. There were quite a few fungal pathogens, while the antibacterial action of defence proteins was hardly touched upon in the poplar context (e.g. in the article on defensin it in vitro inhibited the growth of E. coli and A. tumefaciens, but they are not poplar pathogens), but from molecules with a broad spectrum of action such as defensins, many AMPs and TLPs, etc. (but e.g. not chitinases), we would very likely expect also antibacterial (in addition to antifungal) actions with very high probability (also because other plants have these actions). We have marked this nuance with a question mark.

  1. The illustration of genes in Figure 7 was unsuitable. The microRNAs regulate gene expression, but the proteins were drawn in the figure. The icons that represented the protein should be replaced by showing the gene names. In addition, the gene names should be in italics.

Response. Yes, indeed, it was our serious fault. Now we replaced protein names with gene names everywhere (according to the articles, NCBI Gene and Uniprot) and wrote them in italics. In the case of too complex titles (e.g., NBS-LRRs genes), targets are listed as protein names + "genes".

  1. As above, it should be cautious to show the gene names in italics in the text.

Response. Although we missed this in Figure 7, we were quite careful to do so in the text. Where genes were mentioned, we used italics. Where we talked about proteins, we wrote in regular font. If you find places in the text where there are errors, please let us know. But it seems to me that this point has been fulfilled even before the original submission of the manuscript.
